# Gridded Transformer Neural Processes for Spatio-Temporal Data

**Matthew Ashman** [* 1]   **Cristiana Diaconu** [* 1]   **Eric Langezaal** [* 1 2]   **Adrian Weller** [1 3]   **Richard E. Turner** [1 3]

## Abstract

Effective modelling of large-scale spatio-temporal datasets is essential for many domains, yet existing approaches often impose rigid constraints on the input data, such as requiring them to lie on fixed-resolution grids. With the rise of foundation models, the ability to process diverse, heterogeneous data structures is becoming increasingly important. Neural processes (NPs), particularly transformer neural processes (TNPs), offer a promising framework for such tasks, but struggle to scale to large spatio-temporal datasets due to the lack of an efficient attention mechanism. To address this, we introduce gridded pseudo-token TNPs which employ specialised encoders and decoders to handle unstructured data and utilise a processor comprising gridded pseudo-tokens with efficient attention mechanisms. Furthermore, we develop equivariant gridded TNPs for applications where exact or approximate translation equivariance is a useful inductive bias, improving accuracy and training efficiency. Our method consistently outperforms a range of strong baselines in various synthetic and real-world regression tasks involving large-scale data, while maintaining competitive computational efficiency. Experiments with weather data highlight the potential of gridded TNPs and serve as just one example of a domain where they can have a significant impact.

## 1. Introduction

The proliferation of data from *in situ* sensors, remote observations, and scientific computing models is transforming spatio-temporal modelling. This has led to a surge of interest from the machine learning community to develop new tools and models to support these efforts. However,

---

*Equal contribution [1]University of Cambridge [2]University of Amsterdam [3]Alan Turing Institute. Correspondence to: Matthew Ashman <mca39@cam.ac.uk>, Cristiana Diaconu <cdd43@cam.ac.uk>.

*Proceedings of the 42nd International Conference on Machine Learning*, Vancouver, Canada. PMLR 267, 2025. Copyright 2025 by the author(s).

building such models is non-trivial: the data are typically heterogeneous, irregular in space and time, and multi-modal, with each modality potentially sampled at different spatio-temporal locations. A field that has greatly benefitted from these recent advancements, and will serve as the motivating real-world use case throughout this work, is medium-range weather and environmental forecasting. Here, a new generation of models have improved performance and reduced computational costs (Bodnar et al., 2025; Lam et al., 2023; Price et al., 2025; Bi et al., 2023; Nguyen et al., 2023; Chen et al., 2023b;a). These models operate on environmental variables that are regularly structured in space and time, allowing them to leverage architectures from the vision and language community, such as the Vision Transformer (ViT; Dosovitskiy et al. 2020), Swin Transformer (Liu et al., 2021), and Perceiver (Jaegle et al., 2021). They are trained and deployed on data from computationally intensive scientific simulation and analysis techniques, which integrate observational data (e.g. from weather stations, ships, buoys, etc.) with simulation data to provide the best estimate of the atmosphere's state. They are currently *not* trained on observational measurements directly.

We are now on the cusp of a second generation of such models that will integrate unstructured observational data alongside or instead of analysis data altogether, improving accuracy and reducing computational costs. Achieving this is challenging, as such models need to be able to 1) handle data at arbitrary input locations, and 2) effectively combine modalities that are not necessarily recorded at the same locations. Only a few works have attempted this more ambitious problem (Allen et al., 2025; McNally et al., 2024; Xu et al., 2025; Xiao et al., 2024), and the optimal architectures remain unclear. This presents fertile ground for impactful research, with findings easily transferable to other spatio-temporal domains. For instance, the same techniques could be used to model real-life systems governed by partial differential equations (PDEs), by pre-training on synthetic PDE simulations, followed by fine-tuning on real-world unstructured observations such as wind tunnel measurements.

What framework is most suitable for spatio-temporal tasks containing both structured and unstructured data? Here we advocate for neural processes (NPs; Garnelo et al. 2018a;b) a family of meta-learning models that map from datasets of arbitrary size and structure to predictions over outputs at

arbitrary locations. NPs support a probabilistic treatment of observations, enabling them to output uncertainty estimates that are crucial for downstream decision making. Moreover, NPs are flexible—unlike other models that only perform forecasting, they can solve more general state estimation problems, including forecasting, data fusion, data interpolation and data assimilation. Although early versions of NPs were limited, recent developments (Kim et al., 2019; Gordon et al., 2020; Nguyen & Grover, 2022; Ashman et al., 2024a; Feng et al., 2023; Bruinsma et al., 2021; Ashman et al., 2024b) have significantly improved their effectiveness, especially for small-scale spatio-temporal regression problems. NPs are now being used in a wide-range of applications, from climate downscaling, data assimilation and sensor placement (Vaughan et al., 2022; Andersson et al., 2023; Chen et al., 2024; Niu et al., 2024), to tasks as diverse as molecular property prediction (García-Ortegón et al., 2024). Among NP variants, the convolutional conditional neural process (ConvCNP; Gordon et al., 2020) is the leading approach for spatio-temporal modelling. The ConvCNP employs a set convolution operation to project unstructured data onto a regular grid through kernel-based interpolation. While effective, the success of transformers over convolutional neural networks (CNNs; LeCun et al., 1998) suggests potential for improvement. Indeed, recent developments in transformer-based NPs (TNPs; Nguyen & Grover 2022; Feng et al. 2023; Ashman et al. 2024a) have shown impressive performance on small-scale problems. However, unlike the aforementioned large-scale environmental models for gridded data, TNPs are yet to fully leverage efficient attention mechanisms. As a result of the quadratic computational complexity of full attention, they have been unable to scale to complex spatio-temporal datasets. This is because such efficient techniques require structured—more specifically, gridded—data, limiting their immediate applicability.

Motivated by the approach taken by the ConvCNP, we pursue a straightforward solution: encoding the input data onto a structured grid before passing it through a transformer-based architecture. We introduce attention-based mechanisms for encoding the unstructured data, as well as for processing the resulting structured grid, giving rise to *gridded TNPs*—a general-purpose tool for spatio-temporal state estimation. Our core contributions are:

1. We develop a novel attention-based grid encoder, inspired by the idea of 'pseudo-tokens' (Jaegle et al., 2021; Feng et al., 2023; Lee et al., 2019), which outperforms existing encoders for spatio-temporal data.

2. Equipped with the above-mentioned pseudo-token grid encoder, we enable TNPs to use efficient attention mechanisms, utilising advancements such as the ViT (Dosovitskiy et al., 2020) and Swin Transformer (Liu et al., 2021).

3. We develop an efficient $k$-nearest-neighbour attention-based grid decoder, facilitating the evaluation of predic-

tive distributions at arbitrary spatio-temporal locations. Remarkably, we find that this outperforms full attention.

4. We adapt computationally expensive methods proposed in the literature (Ashman et al., 2024a) to efficiently incorporate translation equivariance into gridded TNPs, improving learning efficiency. Additionally, to account for symmetry imperfections often present in real-world data, we build on Ashman et al. 2024b to achieve approximate translation equivariance with gridded TNPs.

5. We empirically evaluate our model on a range of synthetic and real-world spatio-temporal regression tasks, demonstrating both the ability to i) maintain strong performance on large spatio-temporal datasets and ii) handle multiple sources of unstructured data effectively, all while maintaining a low computational complexity.

## 2. Background

We consider a supervised learning setting with input and output spaces $\mathcal{X} = \mathbb{R}^{D_x}$ and $\mathcal{Y} = \mathbb{R}^{D_y}$. A dataset consists of input-output pairs $(\mathbf{x}, \mathbf{y}) \in \mathcal{X} \times \mathcal{Y}$, with context and target sets $\mathcal{D}_c, \mathcal{D}_t$ containing $|\mathcal{D}_c| = N_c$ and $|\mathcal{D}_t| = N_t$ points. Let $\mathbf{X}_c \in (\mathcal{X})^{N_c}, \mathbf{Y}_c \in (\mathcal{Y})^{N_c}$ and $\mathbf{X}_t \in (\mathcal{X})^{N_t}, \mathbf{Y}_t \in (\mathcal{Y})^{N_t}$ denote the inputs and outputs for $\mathcal{D}_c$ and $\mathcal{D}_t$. We denote a single task as $\xi = (\mathcal{D}_c, \mathcal{D}_t) = ((\mathbf{X}_c, \mathbf{Y}_c), (\mathbf{X}_t, \mathbf{Y}_t))$.

### 2.1. Neural Processes

Neural processes (NPs; Garnelo et al. 2018a;b) can be viewed as neural-network-based mappings from context sets $\mathcal{D}_c$ to predictive distributions at target locations $\mathbf{X}_t$, $p(\cdot \mid \mathbf{X}_t, \mathcal{D}_c)$. In this work, we restrict our attention to conditional NPs (CNPs; Garnelo et al. 2018a), which, unless deployed autoregressively, target marginal predictive distributions by assuming that the predictive densities factorise: $p(\mathbf{Y}_t|\mathbf{X}_t, \mathcal{D}_c) = \prod_{n=1}^{N_t} p(\mathbf{y}_{t,n}|\mathbf{x}_{t,n}, \mathcal{D}_c)$. We denote all parameters of a CNP by $\theta$. CNPs are trained in a meta-learning fashion, in which the expected predictive log-probability is maximised $\theta_{\mathrm{ML}} = \arg\max_\theta \mathcal{L}_{\mathrm{ML}}(\theta)$, where $\mathcal{L}_{\mathrm{ML}}(\theta) = \mathbb{E}_{p(\xi)}\left[\sum_{n=1}^{N_t} \log p_\theta(\mathbf{y}_{t,n}|\mathbf{x}_{t,n}, \mathcal{D}_c)\right]$. In the limit of infinite tasks and model capacity, the global maximum is achieved if and only if the model recovers the ground-truth predictive distributions (Proposition 3.26 by Bruinsma, 2022). For real-world datasets, with only a finite number of training tasks, we approximate this expectation with an average over tasks. In Appendix A we present a unifying construction for CNPs involving three components: the *encoder* $e\colon \mathcal{X} \times \mathcal{Y} \to \mathcal{Z}$, which encodes each $(\mathbf{x}_{c,n}, \mathbf{y}_{c,n}) \in \mathcal{D}_c$ into a token representation $\mathbf{z}_{c,n} \in \mathcal{Z}$, the *processor* $\rho\colon \left(\bigcup_{n=0}^{\infty} \mathcal{Z}^n\right) \times \mathcal{X} \to \mathcal{Z}$, which processes the set of context tokens and the target input $\mathbf{x}_t$ to obtain a target dependent token $\mathbf{z}_t \in \mathcal{Z}$, and the *decoder* $d\colon \mathcal{Z} \to \mathcal{P}_{\mathcal{Y}}$, which maps from the target token to the predictive distribution over the output at that target location. $\mathcal{P}_{\mathcal{Y}}$ denotes the

space of distributions over the output space $\mathcal{Y}$.

## 2.2. Transformers and Transformer Neural Processes

Transformers can be viewed as general set functions (Lee et al., 2019), making them ideal for NPs, which must ingest datasets. This section briefly overviews transformers and their integration into CNPs, leading to the TNP family.

**Self-Attention and Cross-Attention** Broadly speaking, transformer-based architectures consist of two operations: multi-head self-attention (MHSA) and multi-head cross-attention (MHCA). Informally, the MHSA operation updates a set of tokens using the same set, whereas the MHCA operation updates one set of tokens using a different set. More formally, let $\mathbf{Z} \in \mathbb{R}^{N \times D_z}$ denote a set of $N$ $D_z$-dimensional input tokens. The MHSA operation updates this set of tokens for $\forall\, n = 1, \ldots, N$ as

$$\mathbf{z}_n \leftarrow \mathrm{cat}\Big(\Big\{\sum_{m=1}^{N} \alpha_h(\mathbf{z}_n, \mathbf{z}_m)\mathbf{z}_m^T \mathbf{W}_{V,h}\Big\}_{h=1}^{H}\Big)\mathbf{W}_O. \quad (1)$$

Here, $\mathbf{W}_{V,h} \in \mathbb{R}^{D_z \times D_V}$ and $\mathbf{W}_O \in \mathbb{R}^{HD_V \times D_z}$ are the value and projection weight matrices, where $H$ denotes the number of 'heads', and $\alpha_h$ is the attention mechanism. This is most often a softmax-normalised transformed inner-product between pairs of tokens: $\alpha_h(\mathbf{z}_n, \mathbf{z}_m) = \mathrm{softmax}(\{\mathbf{z}_n^T \mathbf{W}_{Q,h}\mathbf{W}_{K,h}^T\mathbf{z}_m\}_{m=1}^{N})_m$, where $\mathbf{W}_{Q,h} \in \mathbb{R}^{D_z \times D_{QK}}$ and $\mathbf{W}_{K,h} \in \mathbb{R}^{D_z \times D_{QK}}$ are the query and key matrices. The MHCA operation updates one set of tokens, $\mathbf{Z}_1 \in \mathbb{R}^{N_1 \times D_z}$, using another set of tokens, $\mathbf{Z}_2 \in \mathbb{R}^{N_2 \times D_z}$, in a similar manner for $\forall\, n = 1, \ldots, N_1$:

$$\mathbf{z}_{1,n} \leftarrow \mathrm{cat}\Big(\Big\{\sum_{m=1}^{N_2} \alpha_h(\mathbf{z}_{1,n}, \mathbf{z}_{2,m})\mathbf{z}_{2,m}^T \mathbf{W}_{V,h}\Big\}_{h=1}^{H}\Big)\mathbf{W}_O. \quad (2)$$

MHSA / MHCA operations are combined with layer-normalisation and point-wise MLPs to obtain MHSA / MHCA blocks. Unless stated otherwise, we adopt the order used by Vaswani et al. (2017) (detailed in Appendix G).

**Pseudo-Token-Based Transformers** Transformers based on pseudo-tokens, first introduced by Jaegle et al. 2021 with the Perceiver, remedy the quadratic computational complexity induced by the standard transformer by condensing the set of $N$ tokens, $\mathbf{Z} \in \mathbb{R}^{N \times D_z}$, into a smaller set of $M \ll N$ 'pseudo-tokens', $\mathbf{U} \in \mathbb{R}^{M \times D_z}$, using the MHCA operation. The processor acts on these pseudo-tokens (instead of the original set), reducing computational complexity from $\mathcal{O}(N^2)$—the cost of MHSA on the original set—to $\mathcal{O}(NM + M^2)$—the cost of MHCA between the original set and the pseudo-tokens, followed by MHSA on the pseudo-tokens.

**Transformer Neural Processes** Transformer neural processes (TNPs) use transformer-based architectures as the processor in the CNP construction described in Section 2.1. First, each context point $(\mathbf{x}_{c,n}, \mathbf{y}_{c,n}) \in \mathcal{D}_c$ and target input $\mathbf{x}_{t,n} \in \mathbf{X}_t$ are encoded to obtain an initial set of context and target tokens, $\mathbf{Z}_c^0 \in \mathbb{R}^{N_c \times D_z}$ and $\mathbf{Z}_t^0 \in \mathbb{R}^{N_t \times D_z}$. Transformers are then used to process the union $\mathbf{Z}^0 = \mathbf{Z}_c^0 \cup \mathbf{Z}_t^0$ using a series of MHSA and MHCA operations, keeping only the output tokens corresponding to the target inputs. These processed target tokens are then mapped to predictive distributions using the decoder. The specific transformer-based architecture is unique to each TNP variant, generally consisting of MHSA operations acting on the context tokens—or pseudo-token representation of the context—and MHCA operations updating the target tokens given the context tokens. We provide a diagram of two popular TNP variants in Appendix A: the regular TNP (Nguyen & Grover, 2022) and the induced set transformer NP (ISTNP; Lee et al. 2019).

The application of TNPs to large datasets is impeded by the use of MHSA and MHCA operations acting on the entire set of context and target tokens. Even when pseudo-tokens are used in pseudo-token TNPs (PT-TNPs), the number of pseudo-tokens $M$ required for accurate predictive inference generally scales with the complexity of the dataset, and the MHCA operations between the pseudo-tokens and the context and target tokens quickly becomes prohibitive as the size of the dataset increases. This motivates the use of efficient attention mechanisms, as used in the ViT (Dosovitskiy et al., 2020) and Swin Transformer (Liu et al., 2021).

## 2.3. Translation Equivariance in TNPs

A useful inductive bias in spatio-temporal modelling, where the data are roughly stationary, is *translation equivariance* (TE): upon translating the data in space or time, the predictions of the model should translate accordingly. Previous works have already considered such symmetries in TNP architectures. Ashman et al. (2024a) introduced the translation equivariant TNP (TE-TNP), denoted by TNP ($T$), where $T$ represents the $d$-dimensional group of translations (with $d$ the dimensionality of the data). Here, translation equivariance is achieved by restricting the attention mechanism $\alpha_{h,n,m} = \alpha_h(\mathbf{z}_n, \mathbf{z}_m, \mathbf{x}_n - \mathbf{x}_m)$ to only depend on differences, rather than absolute input locations, resulting in the TE-MHSA and TE-MHCA operations. These are defined analogously to Equations 1 and 2, with $\alpha_{h,n,m}$ defined as:

$$\alpha_{h,n,m} = \frac{e^{\rho_h\big(\mathbf{z}_n^T \mathbf{W}_{Q,h}[\mathbf{W}_{K,h}]^T \mathbf{z}_m, \mathbf{x}_n - \mathbf{x}_m\big)}}{\sum_{m=1}^{N} e^{\rho_h\big(\mathbf{z}_n^T \mathbf{W}_{Q,h}[\mathbf{W}_{K,h}]^T \mathbf{z}_m, \mathbf{x}_n - \mathbf{x}_m\big)}}, \quad (3)$$

where $\rho_h : \mathbb{R} \times \mathbb{R}^{D_x} \to \mathbb{R}$ is a learnable function, parameterised by an MLP, and the initial tokens $\{\mathbf{z}_n\}_{n=1}^{N}$ only depend on output values. TE-MHCA is defined analogously. A more thorough treatment is provided in Appendix B.

Strictly equivariant architectures introduce useful inductive biases but can be overly restrictive for modelling complex real-world data that are rarely perfectly symmetric. To address this, Ashman et al. (2024b) propose the approximately equivariant NP, which also generalises beyond translation to other symmetry groups. This is achieved through a simple architecture modification, by introducing additional fixed basis functions that enable the model to learn symmetry-breaking features in a data-driven manner. Here, we focus on the approximately TE-TNP, denoted as TNP ($\widetilde{T}$), representing approximate equivariance with respect to $T(d)$.

Ashman et al. (2024b) show approximately equivariant TNPs outperform their non-equivariant counterparts, but they suffer from large time and space complexity due to the need to pass pairwise distances through an MLP when evaluating the weights of the translation equivariant attention mechanism. For this reason, their application has been limited to relatively small datasets.

## 3. Related Work

**Transformers for Point Cloud Data**  A closely related research area is point cloud data modelling (Tychola et al., 2024), with transformer-based architectures employed for a variety of tasks (Lu et al., 2022). To handle large datasets, the use of efficient attention mechanisms has been explored. Several notable approaches use voxelisation, whereby unstructured point clouds are encoded onto a structured grid (Mao et al., 2021; Zhang et al., 2022), prior to employing efficient architectures that operate on grids. Our pseudo-token grid encoder draws inspiration from the voxel-based set attention (VSA) of He et al. (2022), which also cross-attends local neighbourhoods of unstructured tokens onto a structured grid of pseudo-tokens. However, VSA uses the same initial pseudo-token values for all grid locations, as the 'cloud' in which points exist has no unobserved information. Their use of a fixed set of initial values manifests a specific implementation in which all tokens attend to the same set of pseudo-tokens. In contrast, spatio-temporal problems may involve potentially unobserved, fixed topographical information, such as elevation, land use and soil type. To enable the model to capture this, we employ different pseudo-token values for each grid location.

**Models for Structured Weather Data**  Our method is motivated by the need to develop models that can 1) scale to massive spatio-temporal datasets and 2) flexibly handle unstructured data. Here, unstructured data refers to, for example, observations that do not lie on a regular grid (e.g., off-the-grid weather station data, solutions to PDEs on unstructured meshes) or multi-modal data collected at varying spatial locations. Recent advances, particularly in weather modelling, have made significant progress on

the first goal. However, most of these approaches (as discussed in Section 1) are based on efficient transformer-based architectures that can only be applied on structured, gridded data. Moreover, they primarily focus on forecasting[1], hence constraining predictions to the same spatial locations as the inputs. Some methods partially address this limitation by handling missing data (Nguyen et al., 2023), but they still assume a gridded data structure, hence not satisfying the second desideratum. An alternative that can address both points is using graph neural networks (GNNs; Bronstein et al. 2021) as a backbone, but attempts in the weather community proved that GNNs are harder to scale than transformer-based approaches leading to worse performance (e.g. 36.7M parameters for GraphCast (Lam et al., 2023) compared to 1.3B for Aurora (Bodnar et al., 2025)).

NPs arise as a promising framework due to their ability to model stochastic processes, which can be evaluated at any target location, and to flexibly condition on unstructured data. Indeed, our work reflects a more general trend towards more flexible methods, reflected in works such as Aardvark (Allen et al., 2025) and FuXi-DA (Xu et al., 2025), both end-to-end weather prediction models that handle both unstructured and structured data, as well as AtmoRep (Lessig et al., 2023) and Prithvi WxC (Schmude et al., 2024). In Appendix C, we discuss these methods in detail and provide an extended comparison between our approach and GNNs.

## 4. Gridded Transformer Neural Processes

While TNPs have shown promising performance on small to medium-sized datasets, they are unable to scale to large spatio-temporal data. To address this, we consider the use of efficient attention mechanisms that have proved effective for gridded spatio-temporal data (Bodnar et al., 2025; Lam et al., 2023; Price et al., 2025; Nguyen et al., 2023; Bi et al., 2023). Such methods are not immediately applicable without first structuring the unstructured data. We achieve this by drawing upon methods developed in point cloud modelling—notably the VSA (He et al., 2022)—and develop the pseudo-token grid encoder: an effective attention-based method for encoding unstructured data onto a grid.

We provide an illustrative diagram of our proposed approach in Figure 1, which decomposes the processor $\rho\colon \left(\bigcup_{n=0}^{\infty} \mathcal{Z}^n\right) \times \mathcal{X} \to \mathcal{Z}$ into three parts: 1. the grid encoder, $\rho_{ge}\colon \bigcup_{n=0}^{\infty} \mathcal{Z}^n \to \mathcal{Z}^M$, which embeds the set $\{(\mathbf{x}_{c,n}, \mathbf{z}_{c,n}\}_{n=1}^{N_c}$ into tokens $\{\mathbf{u}_m\}_{m=1}^{M}$ at gridded locations $\{\mathbf{v}_m\}_{m=1}^{M}$; 2. the grid processor, $\rho_{gp}\colon \mathcal{Z}^M \to \mathcal{Z}^M$, which transforms the token values; and 3. the grid decoder, $\rho_{gd}\colon \mathcal{Z}^M \times \mathcal{X} \to \mathcal{Z}$, which maps the tokens onto the location $\mathbf{x}_t$. Here, the latent space $\mathcal{Z} = \mathcal{Z}_{\text{token}} \times \mathcal{X}$. We discuss

---

[1]Predicting the entire gridded state at time $t+1$ given a history of previous gridded state(s).

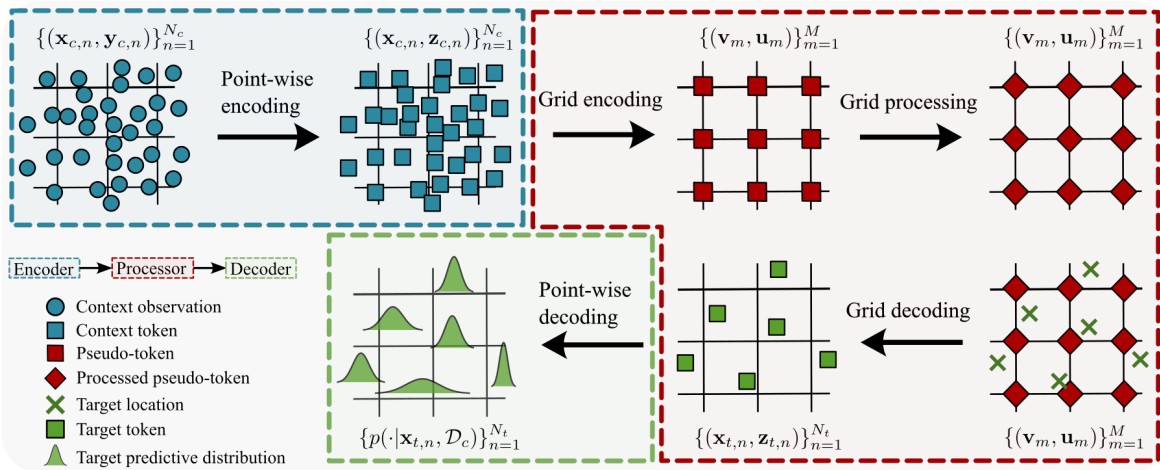

*Figure 1.* Illustration of the complete gridded TNP pipeline. We highlight the encoder (blue), processor (red) and decoder (green).

the choices for these components in this section.

**Grid Encoder: the Pseudo-Token Grid Encoder**  Let $\mathbf{z}_n \in \mathbb{R}^{D_z}$ be the token representation of the input-output pair $(\mathbf{x}_n, \mathbf{y}_n)$ after point-wise embedding. We introduce a set of $\prod_{d=1}^{D_x} M_d$ pseudo-tokens $\mathbf{U}^0 \in \mathbb{R}^{M_1 \times \cdots \times M_{D_x} \times D_z}$ at corresponding grid locations $\mathbf{V} \in \mathbb{R}^{M_1 \times \cdots M_{D_x} \times D_x}$ (i.e. pseudo-token $\mathbf{u}_{m_1,\ldots,m_{D_x}} \in \mathbb{R}^{D_z}$ is associated with location $\mathbf{v}_{m_1,\ldots,m_{D_x}} \in \mathbb{R}^{D_x}$). For ease of reading, we replace the product $\prod_{d=1}^{D_x} M_d$ with $M$ and the indexing notation $m_1, \ldots, m_{D_x}$ with $m$. The pseudo-token grid encoder obtains a pseudo-token representation $\mathbf{U} \in \mathbb{R}^{M \times D_z}$ of $\mathcal{D}_c$ on the grid $\mathbf{V}$ by cross-attending from each set of tokens $\{\mathbf{z}_{c,n}\}_{n \in \mathfrak{N}(\mathbf{v}_m; k)}$ to each initial pseudo-token $\mathbf{u}_m^0$:

$$\mathbf{u}_m \leftarrow \text{MHCA}\left(\mathbf{u}_m^0, \{\mathbf{z}_{c,n}\}_{n \in \mathfrak{N}(\mathbf{v}_m; k)}\right) \quad \forall m \in M. \quad (4)$$

Here, $\mathfrak{N}(\mathbf{v}_m; k)$ denotes the index set of input locations for which $\mathbf{v}_m$ is amongst the $k$ nearest grid locations. In practice, we found that $k = 1$ suffices. While this operation seems computationally intensive, we apply two tricks to make it computationally efficient. First, note that provided the pseudo-token grid is regularly spaced, the set of nearest neighbours for all grid locations can be found in $\mathcal{O}(N_c)$. Second, the operation described in Equation 4 can be performed in parallel for all $m \in M$ by 'padding' each set $\{\mathbf{z}_{c,n}\}_{n \in \mathfrak{N}(\mathbf{v}_m; k)}$ with $\max_i |\mathfrak{N}(\mathbf{v}_i; k)| - |\mathfrak{N}(\mathbf{v}_m; k)|$ 'dummy tokens' and applying appropriate masking, resulting in a computational complexity of $\mathcal{O}\left(M \max_i |\mathfrak{N}(\mathbf{v}_i; k)|\right)$. Restricting the neighbourhood size can further reduce this. We illustrate the pseudo-token grid encoding in Figure 2. In our experiments, we compare the performance of the pseudo-token grid encoder with simple kernel-interpolation onto a grid, a popular method used by the ConvCNP. We provide a detailed description of this approach in Appendix D.

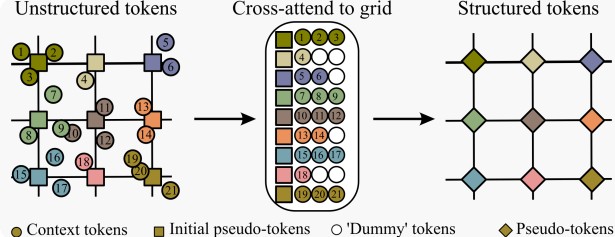

*Figure 2.* An illustration of the pseudo-token grid encoder in 2-D. For an efficient implementation of cross-attention to the pseudo-token grid, we pad sets of neighbourhood tokens with 'dummy' tokens, so that each neighbourhood has the same cardinality.

**Grid Processor: Efficient Attention Transformers**  While we can choose any architecture for processing the pseudo-token grid, including CNNs, we focus on transformer-based architectures employing efficient attention mechanisms—we consider the ViT (Dosovitskiy et al., 2020) and Swin Transformer (Liu et al., 2021). Typically, ViT uses *patch encoding* to downsample a grid of tokens to a coarser grid of tokens, upon which MHSA operations are applied. We explore both patch encoding and directly encoding to a smaller grid via the grid encoder, which, when used in gridded TNPs, we collectively refer to as ViTNPs. The Swin Transformer has the advantage of being able to operate with a finer grid of pseudo-tokens, as only local neighbourhoods of tokens attend to each other at any one time. We refer to its use in gridded TNPs as Swin-TNP.

**Grid Decoder: the Cross-Attention Grid Decoder**  In PT-TNPs, *all* pseudo-tokens $\mathbf{U}$ cross-attend to *all* target tokens $\mathbf{Z}_t$, with computational complexity of $\mathcal{O}(MN_t)$, which is prohibitive for large $N_t$ and $M$. We propose nearest-neighbour cross-attention, in which the pseudo-

tokens $\{\mathbf{u}_m\}_{m \in \tilde{\mathfrak{N}}(\mathbf{x}_{t,n};k)}$ attend to the target token $\mathbf{z}_{t,n}$:

$$\mathbf{z}_{t,n} \leftarrow \text{MHCA}\left(\mathbf{z}_{t,n}^0, \{\mathbf{u}_m\}_{m \in \tilde{\mathfrak{N}}(\mathbf{x}_{t,n};k)}\right). \quad (5)$$

Here, $\tilde{\mathfrak{N}}(\mathbf{x}_{t,n};k)$ denotes the index set of the $k$ grid locations that are closest to $\mathbf{x}_{t,n}$.[2] With equally spaced grid locations, this can be computed for all target tokens in $\mathcal{O}(N_t)$, and the operation in Equation 5 can be parallelised with a complexity of $\mathcal{O}(kN_t)$, leading to a significant computational reduction for $k \ll M$. We also found that nearest-neighbour outperformed full cross-attention in practice, likely due to its beneficial inductive bias for spatio-temporal data.

### 4.1. Handling Multiple Data Modalities

Spatio-temporal datasets often include multi-modal data. For structured data (i.e. gridded data, with modalities sharing the same input locations), this is typically addressed by concatenating the modes along the channel dimension. In contrast, unstructured data—where different modalities can be recorded at distinct locations—require a different approach. While existing frameworks mentioned in Section 3 generally lack support for this setup, our method can be extended to such scenarios. One approach, which we call the *single* pseudo-token grid encoder, is to use modality-specific encoders and apply the pseudo-token grid encoder to the union of tokens. Formally, let $\mathcal{D}_c = \bigcup_{s=1}^S \mathcal{D}_{c,s}$ denote the context dataset partitioned into $S$ source-specific smaller datasets. For each source and context datapoint, we obtain $\mathbf{z}_{c,n,s} = e_s(\mathbf{x}_{c,n,s}, \mathbf{y}_{c,n,s})$, where $e_s \colon \mathcal{X} \times \mathcal{Y}_s \to \mathcal{Z}$ denotes the source-specific point-wise encoder and $\mathcal{Y}_s$ the output space for source $s$. The set of context tokens is obtained as $\mathbf{Z}_c = \{\{\mathbf{z}_{c,n,s}\}_{n=1}^{N_{c,s}}\}_{s=1}^S$. In a second approach, called the *multi* pseudo-token grid encoder, we first form a separate pseudo-token grid for each modality. For each set of source-specific context tokens $\mathbf{Z}_{c,s} = \{e_s(\mathbf{x}_{c,n,s}, \mathbf{y}_{c,n,s})\}_{n=1}^{N_{c,s}}$, we obtain a gridded pseudo-token representation $\mathbf{U}_s \in \mathbb{R}^{M \times D_z}$. We then pass their point-wise concatenation through some function to obtain a unified gridded pseudo-token representation. Both methods can be applied analogously to kernel-interpolation grid encoding.

### 4.2. Introducing Translation Equivariance

To improve training efficiency and generalisation capabilities, we also consider incorporating translation equivariance into gridded TNPs, as a useful inductive bias for spatio-temporal modelling. As mentioned in Section 2.3, the TNP ($T$) of Ashman et al. (2024a) cannot be naively applied to large data because of the need to pass pairwise distances through an MLP when computing the attention weights, resulting in high time and space complexity. When working

with unstructured data, the need to dynamically compute this quantities cannot be overcome, as the locations of the points change at each pass. However, if we instead operated on a uniformly discretised domain, we can simply maintain a fixed matrix of attention weighting biases, closely resembling the relative positional encodings (Shaw et al., 2018) commonly used in transformers. Fortunately, this is exactly the case in the Swin-TNP processor, where the initial, unstructured data has already been projected onto a fixed, structured grid of pseudo-tokens of window size. Given that attention is only performed among these structured pseudo-tokens, we can efficiently implement the TE attention mechanism by introducing of a set of learnable relative positional encodings added to the pre-softmax-normalised attention weights. This comes with negligible computational cost.

For the pseudo-token grid encoding and grid decoding, we must compute relative positional encodings dynamically using the TE-MHCA mechanism proposed by Ashman et al. (2024a). However, due to the use of the nearest-neighbour cross-attention mechanisms, the number of pairwise computations scales only linearly with the number of context and target points. This allows for an efficient and scalable implementation of the gridded Swin-TNP ($T$) model. The encoder and decoder equations are analogous to Equations 4 and 5, but we replace the MHCA operations with TE-MHCA. Finally, we relax strict equivariance through an approximately TE model—Swin-TNP ($\widetilde{T}$). This can be implemented following Ashman et al. (2024b) by adding a few learnable basis functions to the inputs of the TE components of the architecture. We provide more details in Appendix G.

## 5. Experiments

We evaluate our gridded TNPs on synthetic and real-world regression tasks involving large datasets. We show consistent improvements over baselines, especially as task complexity and dataset size increase, while maintaining low computational complexity. Throughout, we compare gridded TNPs with two different grid encoders (GEs)—the kernel-interpolation grid encoder (KI-GE) and our pseudo-token grid encoder (PT-GE)—and two different grid processors—Swin Transformer and ViT. We also compare to the following baselines: the CNP (Garnelo et al., 2018a), the PT-TNP using the induced set-transformer architecture (Lee et al., 2019; Ashman et al., 2024a)[3], and the ConvCNP (Gordon et al., 2020). We do not compare to the regular ANP or TNP (Kim et al., 2019; Nguyen & Grover, 2022) as they are **unable to scale** to the considered context set sizes. Details on experiments, architectures, datasets and hardware can be found in Appendices F and G. We also provide a public implementation of gridded TNPs in the repository

---

[2]For dimension $d = \dim(\mathbf{x}_{t,n})$, we select the nearest $\text{ceil}(k^{1/d})$ neighbours per dimension. Appendix E provides more details.

[3]For all experiments, we use the maximum number of pseudo-tokens allowed by memory constraints.

https://github.com/cambridge-mlg/gridded-tnp.

To illustrate the generality of our framework, we include an additional study in Appendix G.6 using the large-scale EAGLE fluid-dynamics dataset (Janny et al., 2023). The results show that gridded TNPs can be effectively and directly applied—without task-specific modifications—to PDE modelling on irregular meshes.

## 5.1. Meta-Learning Gaussian Process Regression

We begin with two synthetic 2-D regression tasks with datasets drawn from a Gaussian process (GP) with a squared-exponential (SE) kernel approximated using structure kernel interpolation (SKI) (Wilson & Nickisch, 2015). Each dataset contains $1.1 \times 10^4$ datapoints ($N_c = 1 \times 10^4$ context and $N_t = 1 \times 10^3$ target points), with inputs sampled uniformly from $\mathcal{U}_{[-6,6]}$. We consider two SE kernel length-scales: $\ell = 0.1$ and $\ell = 0.5$, implying roughly $10,000$ and $500$ 'wiggles', respectively, in each dataset. While both setups are non-trivial for typical NP synthetic experiments, we expect the former to be intractable for most NPs—it requires the model to both assimilate finely grained information and modulate its predictions accordingly. In Figure 3, we plot the test log-likelihoods against the time taken for a single forward pass (FPT) for two ViT variants (with and without patch encoding), and Swin-TNP. We compare them to two strong baselines: the PT-TNP with $M = 256$ pseudo-tokens, and the ConvCNP with the same grid sizes as the Swin-TNP. Training and inference for all models is performed on one NVIDIA GeForce RTX 2080 Ti.

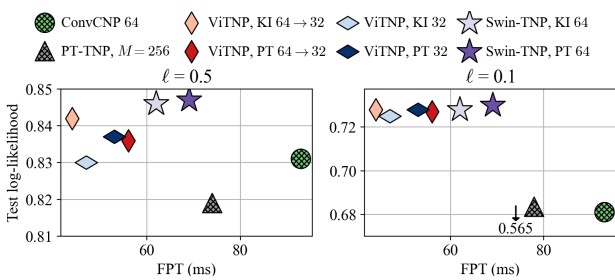

*Figure 3.* Test log-likelihood vs. forward pass time (FPT) for the GP datasets. Baselines are hatched. The considered grid sizes are $64 \times 64$ and $32 \times 32$, shown as 64 and 32. For ViTNPs, we include results with and without patch encoding, the former indicated by the $\rightarrow$ symbol in-between the pre- and post-patch-encoded grid sizes. KI / PT: kernel-interpolation / pseudo-token grid encoding.

Our results show that gridded TNPs outperform all baselines (hatched markers in Figure 3)—especially on the more complex datasets, where they significantly surpass the strongest ConvCNP baseline—while maintaining competitive computational complexity. Further, by comparing the darker to the lighter markers, we observe that the PT-GE outperforms the KI-GE. We show in Appendix G.1 a table of all results

(including the use of mean aggregation grid encoder, as used by FuXi-DA (Xu et al., 2025)), an ablation of the grid size, and the predictive means of selected models on an example dataset with $\ell = 0.1$, illustrating gridded TNPs' superior ability to capture complex data. We also provide a smaller-scale experiment comparing gridded TNPs to the standard TNP (which cannot scale to the larger-scale version due to the lack of efficient attention). We show that in regimes where the standard TNP is applicable, gridded TNPs are able to recover its performance. These additional results demonstrate that the improvements in efficiency of gridded TNPs do not come at the expense of predictive performance.

## 5.2. Real-World Weather Data

We perform two experiments on data from the ERA5 reanalysis by the European Centre for Medium-Range Weather Forecasts (ECMWF; Hersbach et al. 2020). We train on data between 2009-2017, validate on 2018 and test on 2019. After outlining the experimental setups, we present key conclusions from aggregated results, highlighting the consistency across experiments. Training and inference are performed using a single NVIDIA A100 80GB with 32 CPU cores.

**Combining Weather Station Observations with Structured Reanalysis** This experiment tests our models' ability to combine sparse, unstructured observations with structured, on-the-grid data. For this, we use skin temperature (skt) and 2m temperature (t2m).[4] We construct each context dataset by combining the t2m at a random subset of $9,957$ weather station locations (proportion sampled from $\mathcal{U}_{[0,0.3]}$) with a coarsened $180 \times 360$ grid—corresponding to a grid spacing of $1°$—of skt values. The targets are the t2m values at all $9,957$ weather station locations.

**Combining Multiple Sources of Unstructured Wind Speed Observations** We also evaluate the ability of gridded TNPs to model sparsely sampled eastward (u) and northward (v) components of wind at three different pressure levels: 700hPa, 850hPa and 1000hPa (surface level). This dataset is considered *unstructured*, with input locations that differ across the six variables. We perform spatio-temporal interpolation over a latitude / longitude range of $[25°, 49°]$ / $[-125°, -66°]$, which corresponds to the contiguous US, spanning four hours. The proportion of the $543,744$ observations used as the context dataset is sampled from $\mathcal{U}_{[0.05,0.25]}$. The target set size is fixed at $135,936$.

The conclusions of the two weather experiments, empirically supported by the results in Table 1 (where we bold the results that support each conclusion), are as follows:

1. Gridded TNPs outperform all baselines while maintain-

---

[4] skt is the Earth's surface temperature, directly output by atmospheric models, while t2m is found by interpolating between the lowest model temperature and skt (Owens & Hewson, 2018).

*Table 1.* Test log-likelihood (↑) for the t2m station prediction (left) / multi-modal wind speed (right) experiments. The standard errors of the log-likelihood are all below 0.010/0.020. For the station experiment, N̸N̸ indicates that NN-CA is not employed. For the wind speed experiment, m- and s- stand for multi and single GE. FPT: forward pass time for a batch size of eight in ms.

**(a)** t2m station prediction experiment.

| | Model | GE | Grid size | Log-lik.↑ | FPT |
|---|---|---|---|---|---|
| 1) | CNP | - | - | 0.636 | 32 |
| | PT-TNP | - | $M = 256$ | 1.344 | 230 |
| | ConvCNP N̸N̸ | SetConv | $64 \times 128$ | 1.535 | 96 |
| | Swin-TNP | PT-GE | $64 \times 128$ | **1.819** | 127 |
| 2) | ViTNP | KI-GE | $48 \times 96$ | 1.628 | 167 |
| | ViTNP | KI-GE | $144 \times 288 \rightarrow 48 \times 96$ | **1.734** | 171 |
| 3) | ViTNP | KI-GE | $144 \times 288 \rightarrow 48 \times 96$ | 1.734 | 171 |
| | ViTNP | PT-GE | $144 \times 288 \rightarrow 48 \times 96$ | **1.808** | 215 |
| | Swin-TNP | KI-GE | $64 \times 128$ | 1.683 | 121 |
| | Swin-TNP | PT-GE | $64 \times 128$ | **1.819** | 127 |
| 4) | ViTNP | PT-GE | $144 \times 288 \rightarrow 48 \times 96$ | 1.808 | 215 |
| | Swin-TNP | PT-GE | $64 \times 128$ | **1.819** | 127 |
| 5) | Swin-TNP | PT-GE | $64 \times 128$ | **1.819** | 127 |
| | Swin-TNP N̸N̸ | PT-GE | $64 \times 128$ | 1.636 | 144 |
| 7) | ConvCNP N̸N̸ | SetConv | $64 \times 128$ | 1.535 | 96 |
| | ConvCNP | SetConv | $192 \times 384$ | 1.689 | 74 |
| | Swin-TNP | PT-GE | $64 \times 128$ | 1.819 | 127 |
| | Swin-TNP | PT-GE | $192 \times 384$ | **2.053** | 306 |

**(b)** Multi-modal wind speed experiment.

| | Model | GE | Grid size | Log-lik.↑ | FPT |
|---|---|---|---|---|---|
| 1) | CNP | - | - | $-1.593$ | 33 |
| | PT-TNP | - | $M = 64$ | 3.988 | 166 |
| | ConvCNP N̸N̸ | SetConv | $24 \times 60$ | 6.143 | 210 |
| | Swin-TNP | m-PT-GE | $24 \times 60$ | **8.603** | 375 |
| 2) | ViTNP | m-KI-GE | $12 \times 30$ | 5.371 | 349 |
| | ViTNP | m-PT-GE | $24 \times 60 \rightarrow 12 \times 30$ | **7.288** | 392 |
| 3) | ViTNP | m-KI-GE | $24 \times 60 \rightarrow 12 \times 30$ | 6.906 | 372 |
| | ViTNP | m-PT-GE | $24 \times 60 \rightarrow 12 \times 30$ | **7.288** | 392 |
| | Swin-TNP | m-KI-GE | $24 \times 60$ | 7.603 | 355 |
| | Swin-TNP | m-PT-GE | $24 \times 60$ | **8.603** | 375 |
| 4) | ViTNP | m-PT-GE | $24 \times 60 \rightarrow 12 \times 30$ | 7.288 | 392 |
| | Swin-TNP | m-PT-GE | $24 \times 60$ | **8.603** | 375 |
| 6) | Swin-TNP | s-PT-GE | $24 \times 60$ | 8.073 | 364 |
| | Swin-TNP | m-PT-GE | $24 \times 60$ | **8.603** | 375 |
| 7) | ConvCNP N̸N̸ | SetConv | $24 \times 60$ | 6.143 | 210 |
| | ConvCNP | SetConv | $48 \times 120$ | 7.841 | 64 |
| | Swin-TNP | m-PT-GE | $24 \times 60$ | 8.603 | 375 |
| | Swin-TNP | m-PT-GE | $48 \times 120$ | **9.383** | 369 |

ing relatively low computational complexity.

2. Performing patch encoding prior to projecting onto the latent grid gives better performance in ViTNPs.

3. Pseudo-token approaches outperform kernel interpolation in the grid encoder, especially for coarse grids.

4. Swin-TNP achieves performance comparable to or better than that of ViTNP at a lower computational cost.

5. In the grid decoder, nearest-neighbour cross-attention both reduces the computational cost and improves predictive performance relative to full cross-attention, as highlighted in the t2m station experiment.

6. For the multi-modal experiment, the use of the multi (m-) grid encoders outperform the use of single (s-) ones.

7. When varying the grid size, even the smaller Swin-TNP outperforms the larger ConvCNP baseline.

In Appendices G.2 and G.3 we show per-experiment results, as well as predictive errors for an example dataset for the Swin-TNP, ConvCNP, and PT-TNP, indicating that Swin-TNP also has the most accurate predictive uncertainties. For the t2m station prediction experiment, we provide results for the Swin-TNP using either solely t2m or skt measurements, showing that the model leverages both information sources. Additionally, we experiment with different numbers of nearest neighbours in the encoder and decoder, as well as with richer context sizes, to evaluate the robustness of the model under varying configurations.. Finally, in Ap-

pendix G.5 we conduct further experiments using larger grid sizes for the strongest baseline—ConvCNP—and show that even at its largest configuration, ConvCNP underperforms relative to our best Swin-TNP variant and typically requires more parameters.

**Incorporating Translation Equivariance** Finally, we study the effect of translation equivariance with the efficient Swin-TNP $(T)$ and the approximately equivariant Swin-TNP $(\widetilde{T})$. The architectures are analogous to the non-equivariant Swin-TNP. For the wind speed experiment, we also show the ability of Swin-TNP $(T)$ to generalise spatially, by testing on a region unseen during training (i.e. we train on the US, and test on Europe). The results in Table 2 show that 1) equivariant models outperform their non-equivariant counterpart, 2) they generalise spatially, and 3) relaxing strict equivariance further boosts performance.

# 6. Conclusion

This paper introduces gridded TNPs, an extension to the family of TNPs which enables the use of efficient attention-based transformer architectures—such as the ViT and Swin Transformer—and thus facilitates the application of TNPs on large-scale spatio-temporal data. We also propose a method to leverage gridded TNPs to address the scalability issues of translation equivariant TNPs, and incorporate a

*Table 2.* Test log-likelihood ($\uparrow$) for Swin-TNP ($T$) and ($\widetilde{T}$), compared to the non-equivariant Swin-TNP. For the t2m station experiment we report results for Swin-TNP PT-GE $64 \times 128$, while for the wind speed experiment, we use Swin-TNP m-PT-GE $24 \times 60$. FPT: forward pass time for a batch size of eight in ms.

| | **(a)** t2m stations. | | **(b)** Wind speed. | | |
| | | | US | Europe | |
| Model | Log-lik. ($\uparrow$) | FPT | Log-lik. ($\uparrow$) | Log-lik. ($\uparrow$) | FPT |
|---|---|---|---|---|---|
| Swin-TNP ($T$) | 1.895 | 125 | 9.972 | 10.087 | 373 |
| Swin-TNP ($\widetilde{T}$) | **1.926** | 142 | **10.086** | **10.429** | 467 |
| Swin-TNP | 1.819 | 127 | 8.603 | $-13.265$ | 375 |

useful inductive bias for spatio-temporal tasks. Gridded TNPs consist of three components: the grid encoder, the grid processor, and the grid decoder. Through a comprehensive study, we identify the optimal design choices for each component. The best-performing gridded TNP variant incorporates: 1. our novel *pseudo-token grid encoder* to move unstructured data onto a structured grid of pseudo-tokens, 2. the *Swin Transformer* to process this grid, 3. the *pseudo-token grid decoder* equipped with the nearest-neighbour cross-attention to evaluate at arbitrary locations, 4. and *approximate translation equivariance* for efficient training and generalisation capabilities. We show that gridded TNPs outperform strong baselines on large-scale synthetic and real-world regression tasks, with context sets with over 100,000 datapoints. This work marks a step toward building flexible architectures for large, unstructured spatio-temporal data, with applications including, but not limited to, weather and environmental, as well as PDE modelling.

## Acknowledgements

CD is supported by the Cambridge Trust Scholarship. EL acknowledges financial support for this conference from the Amsterdam ELLIS Unit, Qualcomm and Ørpheus AI. AW acknowledges support from a Turing AI fellowship under grant EP/V025279/1 and the Leverhulme Trust via CFI. RET is supported by Google, Amazon, ARM, Improbable, EPSRC grant EP/T005386/1, and the EPSRC Probabilistic AI Hub (ProbAI, EP/Y028783/1). We thank Anna Vaughaun, Stratis Markou and Wessel Bruinsma for their valuable feedback.

## Impact Statement

This paper presents work whose goal is to advance the field of Machine Learning. There are many potential societal consequences of our work, none which we feel must be specifically highlighted here.

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

# A. A Unifying Construction of Conditional Neural Processes

The many variants of CNPs differ in their construction of the predictive distribution $p(\cdot|\mathbf{x}_t, \mathcal{D}_c)$. Here, we introduce a construction of CNPs that generalises all variants. CNPs are formed from three components: the *encoder*, the *processor*, and the *decoder*. The encoder, $e\colon \mathcal{X} \times \mathcal{Y} \to \mathcal{Z}$, first encodes each $(\mathbf{x}_{c,n}, \mathbf{y}_{c,n}) \in \mathcal{D}_c$ into some latent representation, or *token*, $\mathbf{z}_{c,n} \in \mathcal{Z}$. The processor, $\rho\colon \left(\bigcup_{n=0}^{\infty} \mathcal{Z}^n\right) \times \mathcal{X} \to \mathcal{Z}$, processes the set of context tokens $e(\mathcal{D}_c) = \{e(\mathbf{x}_{c,n}, \mathbf{y}_{c,n})\}_n$ together with the target input $\mathbf{x}_t$ to obtain a target dependent token, $\mathbf{z}_t \in \mathcal{Z}$.[5] Finally, the decoder, $d\colon \mathcal{Z} \to \mathcal{P}_\mathcal{Y}$, maps from the target token to the predictive distribution over the output at that target location. Here, $\mathcal{P}_\mathcal{Y}$ denotes the set of distributions over $\mathcal{Y}$. We illustrate this decomposition in Figure 4.

$$\{\mathbf{x}_{c,n}, \mathbf{y}_{c,n}\}_n \xrightarrow{\text{Encode, } e(\cdot)} \{\mathbf{z}_{c,n}\}_n \xrightarrow{\text{Process, } \rho(\cdot, \mathbf{x}_t)} \mathbf{z}_t \xrightarrow{\text{Decode, } d(\cdot)} p(\cdot \mid \mathcal{D}_c, \mathbf{x}_t)$$

*Figure 4.* A unifying construction of CNPs, with $\mathcal{D}_c = \{(\mathbf{x}_{c,n}, \mathbf{y}_{c,n})\}_n$ and $\mathbf{z}_{c,n} = e(\mathbf{x}_{c,n}, \mathbf{y}_{c,n})$.

We present below several schematics showing the architectures of different members of the CNP family, and detail how they can be constructed following this universal construction.

**Original CNP** The first architecture is based on (Garnelo et al., 2018a) and is the least complex of the CNP variants, using a summation as the permutation-invariant aggregation. The diagram is shown in Figure 5. The encoder of a CNP is an MLP, which maps from each concatenated pair $(\mathbf{x}_{c,n}, \mathbf{y}_{c,n}) \in \mathcal{D}_c$ to some representation $\mathbf{z}_{c,n} \in \mathbb{R}^{D_z}$. The processor sums together these representations, and combines the aggregated representation with the target input using $\tau$: $\rho(\{\mathbf{z}_{c,n}\}_n, \mathbf{x}_t) = \tau(\sum_n \mathbf{z}_{c,n}, \mathbf{x}_t)$. $\tau$ is often just the concatenation operation. Finally, the decoder consists of another MLP which maps from $\mathbf{z}_t = \tau(\sum_n \mathbf{z}_{c,n}, \mathbf{x}_t)$ to the parameter space of some distribution over the output space (e.g. Gaussian).

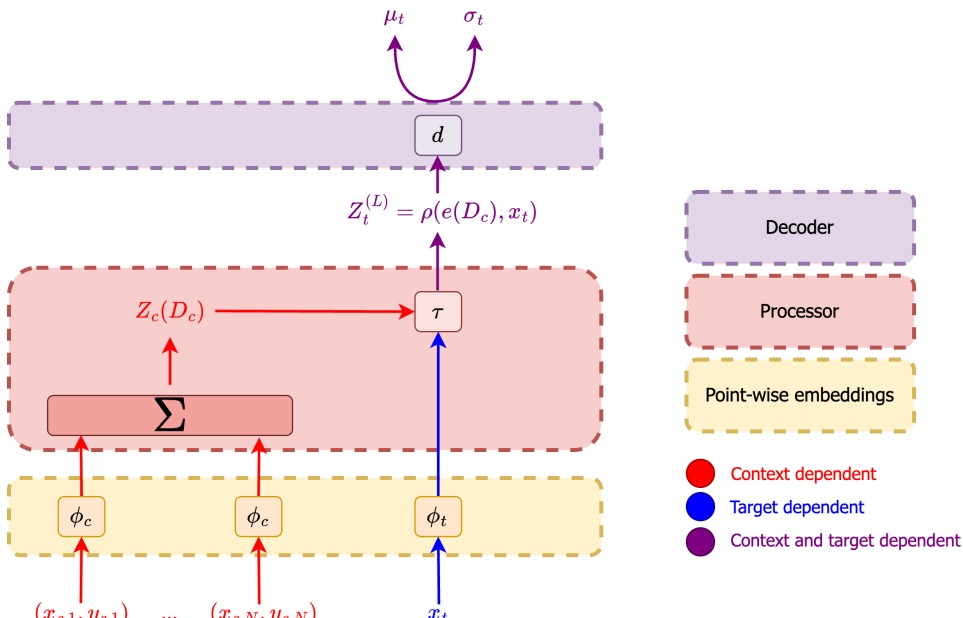

*Figure 5.* A diagram illustrating the architecture of the plain CNP (Garnelo et al., 2018a). First, the context set $(\mathbf{x}_{c,n}, \mathbf{y}_{c,n})$ and the target tokens $\mathbf{x}_{t,n}$ are encoded using point-wise embeddings. These are fed into the CNP processor, which performs a simple permutation-invariant aggregation of the context tokens. These are then concatenated with the target tokens and fed into the decoder, which outputs the parameters of the specified NP distribution based on the target representation (in this case, mean and variance of a Gaussian).

---

[5]The space of target dependent tokens does **not** need to be the same as that of context tokens—we have used $\mathcal{Z}$ in both cases for simplicity. It is also possible for $\mathcal{Z}$ to be the product of multiple spaces, e.g. $\mathcal{Z} = \mathcal{Z}_{token} \times \mathcal{X}$ where we retain information about the input locations.

**ConvCNP**   Another member of the CNP family is the ConvCNP (Gordon et al., 2020), that embeds sets into function space in order to achieve translation equivariance. This can lead to more efficient training in applications where such an inductive bias is appropriate, such as stationary time-series or spatio-temporal regression tasks. We show in Figure 6 a schematic of the architecture. The encoder of a ConvCNP is simply the identity function. The processor then encodes the discrete function represented by input-output pairs $\{(\mathbf{x}_{c,n}, \mathbf{y}_{c,n})\}_n$ onto a regular grid using the kernel-interpolation grid encoder (KI-GE). It then processes this grid using a CNN, which is afterwards combined with the target location using a kernel-interpolation grid decoder (KI-GD). Letting $\mathbf{U} = \{\mathbf{u}_m\}_m$ denote the set of $M$ values on gridded locations $\mathbf{V} = \{\mathbf{v}_m\}_m$, we can decompose the ConvCNP processor as

$$\mathbf{u}_m \leftarrow \sum_n [1, \ \mathbf{y}_{c,n}]^T \psi_{ge}(\mathbf{v}_m - \mathbf{x}_{c,n}) \tag{6}$$

$$\mathbf{U} \leftarrow \mathrm{CNN}(\mathbf{U}, \mathbf{V}) \tag{7}$$

$$\mathbf{z}_t \leftarrow \sum_m \mathbf{u}_m \psi_{gd}(\mathbf{x}_t - \mathbf{v}_m). \tag{8}$$

Here, $\psi_{ge}, \ \psi_{gd} \colon \mathbb{R}^{D_x} \times \mathbb{R}^{D_x} \to \mathbb{R}$ denote the KI-GE kernel and KI-GD kernel. As with the CNP, the decoder is an MLP mapping to the parameter space of some distribution over the output space.

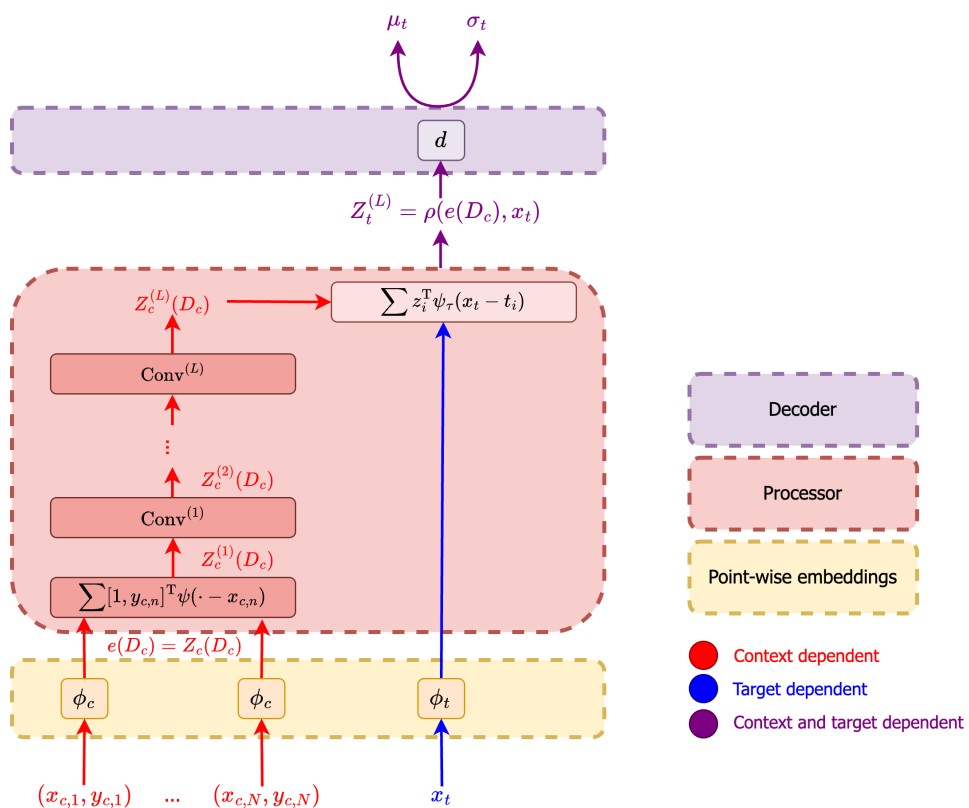

*Figure 6.* A diagram illustrating the architecture of the ConvCNP (Gordon et al., 2020). First, the context set $(\mathbf{x}_{c,n}, \mathbf{y}_{c,n})$ and the target tokens $\mathbf{x}_{t,n}$ are encoded using the identity encoder. These are fed into the ConvCNP processor, which uses a KI-GE to project the tokens into function space. These are then evaluated at discrete locations using a pre-specified resolution, followed by multiple layers of a CNN-based architecture acting upon the discretised signal. To decode at arbitrary locations, a KI-GD is used, giving rise to the target token representation. This is fed into the decoder, which outputs the parameters of the specified NP distribution (in this case, mean and variance of a Gaussian).

**TNPs**   There are a number of different architectures used for the different members of the TNP family. We provide below diagrams for two members mentioned in the main paper, namely the TNP of (Nguyen & Grover, 2022) and the induced

set transformer (ISTNP) of (Lee et al., 2019). For the standard TNP, the encoder consists of an MLP mapping from each $(\mathbf{x}_{c,n}, \mathbf{y}_{c,n}) \in \mathcal{D}_c$ to some representation (token) $\mathbf{z}_{c,n} \in \mathbb{R}^{D_z}$. The processor begins by embedding the target location in the same space as the context tokens, giving $\mathbf{z}_t \in \mathbb{R}^{D_z}$. It then iterates between applying MHSA operations on the set of context tokens, and MHCA operations from the set of context tokens into the target token:

$$\left.\begin{array}{l} \mathbf{Z}_c \leftarrow \mathrm{MHSA}(\mathbf{Z}_c) \\ \mathbf{z}_t \leftarrow \mathrm{MHCA}(\mathbf{z}_t;\ \mathbf{Z}_c) \end{array}\right\} \times L. \tag{9}$$

Again, the decoder consists of an MLP mapping from $\mathbf{z}_t$ to the parameter space of some distribution over outputs.

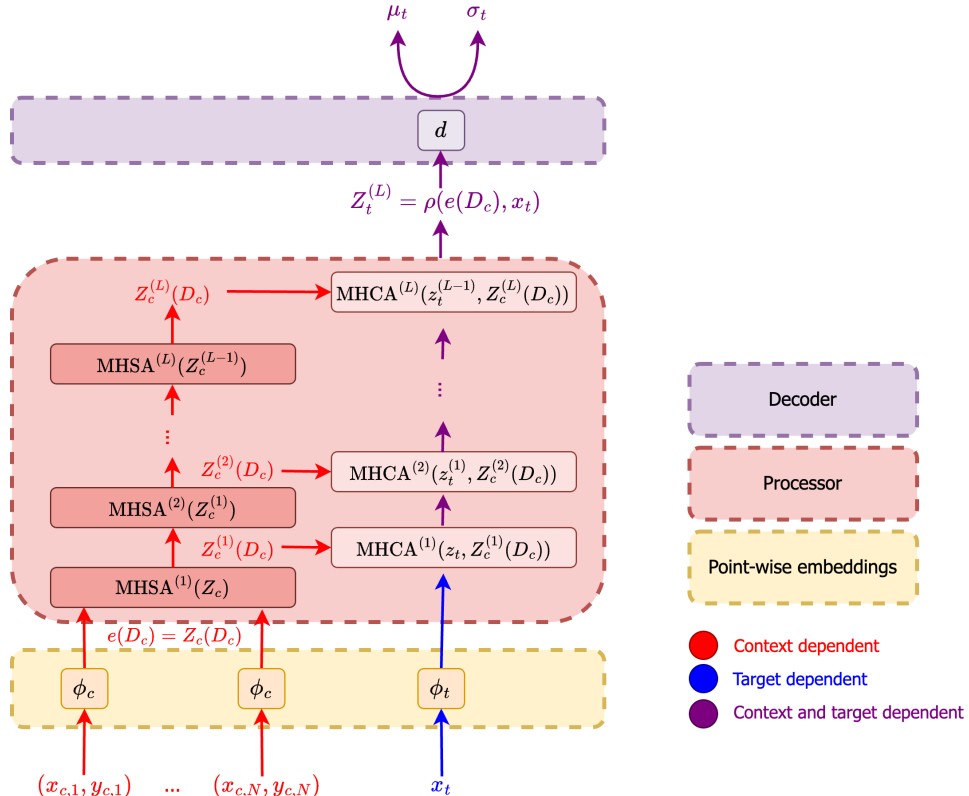

*Figure 7.* A diagram illustrating the architecture of the TNP (Nguyen & Grover, 2022). First, the context set $(\mathbf{x}_{c,n}, \mathbf{y}_{c,n})$ and the target tokens $\mathbf{x}_{t,n}$ are encoded using point-wise embeddings to obtain the context set representation $\mathbf{Z}_c$ and target representation $\mathbf{Z}_t$. These are fed into the TNP processor, which takes in the union of $[\mathbf{Z}_c, \mathbf{Z}_t]$ and outputs the token corresponding to the target inputs $\mathbf{Z}_t^{(L)}$. At each layer of the processor, the context set representation is first updated through an MHSA layer, which is then used to modulate the target set representation through a MHCA layer between the target set representation from the previous layer and the updated context representation. Finally, the decoder outputs the parameters of the specified NP distribution based on the target representation from the final layer (in this case, mean and variance of a Gaussian).

One of the main limitations of TNPs is the cost of the attention mechanism, which scales quadratically with the number of input tokens. Several works (Feng et al., 2023; Lee et al., 2019) addressed this shortcoming by incorporating ideas from the Perceiver-style architecture (Jaegle et al., 2021) into NPs. The strategy is to introduce a set of $M$ 'pseudo-tokens' which act as an information bottleneck between the context and target sets. Provided that $M << N_c$, where $N_c$ is the number of context points, this leads to a significant reduction in computational complexity. The architecture we consider in this work is called the induced set transformer NP (ISTNP), and differs from the plain TNP in the calculations performed in the processor. At each layer, the pseudo-token representation is first updated through an MHCA operation from the context set to the pseudo-tokens. The updated representation is then used to modulate the context and target sets separately, through separate MHCA operations:

$$\left.\begin{array}{l} \mathbf{U} \leftarrow \mathrm{MHCA}(\mathbf{U};\ \mathbf{Z}_c) \\ \mathbf{z}_t \leftarrow \mathrm{MHCA}(\mathbf{z}_t;\ \mathbf{U}) \\ \mathbf{Z}_c \leftarrow \mathrm{MHCA}(\mathbf{Z}_c;\ \mathbf{U}) \end{array}\right\} \times L. \tag{10}$$

Thus, as apparent in Figure 8, the context and target sets never interact directly, but only through the 'pseudo-tokens'. The computational cost at each layer reduces from $\mathcal{O}(N_c^2 + N_c N_t)$ in the plain TNP, where $N_c$ and $N_t$ represent the number of context and target points, respectively, to $\mathcal{O}(M(2N_c + N_t))$. This is a significant reduction provided that $M << N_c$, resulting in an apparent linear dependency on $N_c$. However, in practice, $M$ is not independent of $N_c$, with more pseudo-tokens needed as the size of context set increases.

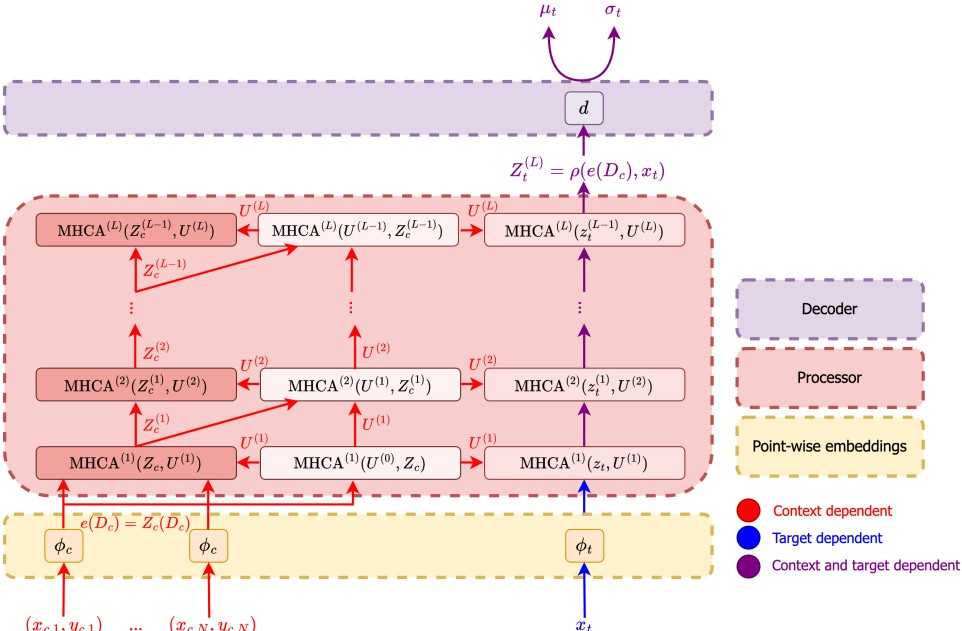

*Figure 8.* A diagram illustrating the architecture of the ISTNP (Lee et al., 2019). As opposed to the regular TNP of Nguyen & Grover 2022, the ISTNP uses a 'summarised' representation of the context set through the use of pseudo-tokens. They are first randomly initialised ($\mathbf{U}^{(0)}$). Then, their representation is updated through cross attention with the context set representation from the previous layer (i.e. at layer $l$: $\mathbf{U}^{(l)} = \mathrm{MHCA}^{(l)}(\mathbf{U}^{(l-1)}, \mathbf{Z}_c^{(l)})$). This updated set of pseudo-tokens $\mathbf{U}^{(l)}$ is then used to modulate both the context set representation at the current layer through cross-attention (i.e. $\mathbf{Z}_c^{(l)} = \mathrm{MHCA}(\mathbf{Z}_c^{(l-1)}, \mathbf{U}^{(l)})$), as well as the target set representation (i.e. $\mathbf{Z}_t^{(l)} = \mathrm{MHCA}(\mathbf{Z}_t^{(l-1)}, \mathbf{U}^{(l)})$). Thus, the context and target set representations do not interact directly, but only through the pseudo-tokens, which act as an information flow bottleneck between the two in order to decrease the computational demands of the TNP.

## B. Translation and Approximate Equivariance

We aim to give a brief introduction to Translation Equivariant TNPs (TE-TNPs) (Ashman et al., 2024a) and Approximately Equivariant NPs (Ashman et al., 2024b), and refer the reader to the aforementioned papers for a more thorough treatment. Pseudo-code for a forward pass through these models is provided in Algorithms 3 and 4 in Ashman et al. (2024b).

### B.1. Translation Equivariant TNPs

As mentioned in Ashman et al. (2024a), if the ground-truth stochastic process we are interested in modelling is stationary, building translation equivariance into the model leads to more efficient training by reducing the model search space. Moreover, as formally indicated in Theorem 2.2 in Ashman et al. (2024a), this inductive bias also allows NPs to generalise spatially.

In order to achieve translation equivariance, we must ensure that each token $\mathbf{z_n}$ is translation invariant with respect to its

inputs—this can be achieved by only allowing relative, rather than absolute dependencies on the inputs. More specifically, the attention mechanism only depends on pairwise distances $\mathbf{x_i} - \mathbf{x_j}$, resulting in the translation equivariant MHSA and MHCA operations, which we denote as TE-MHSA and TE-MHCA. In addition, unlike the non-equivariant counterpart, we force the token encoder to only depend on output values $\mathbf{y}_n$.

The TE-MHSA operation is constructed analogously to Equation 1, but the inputs to the attention mechanism only depend on pairwise differences between the inputs, rather than their absolute value

$$\mathbf{z}_n \leftarrow \text{cat}\left(\left\{\sum_{m=1}^{N} \alpha_h(\mathbf{z}_n, \mathbf{z}_m, \mathbf{x}_n - \mathbf{x}_m)\mathbf{z}_m{}^T\mathbf{W}_{V,h}\right\}_{h=1}^{H}\right)\mathbf{W}_O. \tag{11}$$

There are various ways in which the translation equivariant attention weighting can be implemented, but we choose to follow Ashman et al. (2024a) and set $\alpha_{h,n,m} = \alpha_h(\mathbf{z}_n, \mathbf{z}_m, \mathbf{x}_n - \mathbf{x}_m)$, as indicated in Equation 3, as

$$\alpha_{h,n,m} = \frac{e^{\rho_h\left(\mathbf{z}_n{}^T\mathbf{W}_{Q,h}[\mathbf{W}_{K,h}]^T\mathbf{z}_m, \mathbf{x}_n - \mathbf{x}_m\right)}}{\sum_{m=1}^{N} e^{\rho_h\left(\mathbf{z}_n{}^T\mathbf{W}_{Q,h}[\mathbf{W}_{K,h}]^T\mathbf{z}_m, \mathbf{x}_n - \mathbf{x}_m\right)}}, \tag{12}$$

where $\rho_h \colon \mathbb{R} \times \mathbb{R}^{D_x} \to \mathbb{R}$ is a learnable function, parameterised by an MLP, and $\{\mathbf{z}_n\}_{n=1}^{N}$ only depend on output values. TE-MHCA is defined analogously

$$\mathbf{z}_{1,n} \leftarrow \text{cat}\left(\left\{\sum_{m=1}^{N_2} \alpha_h(\mathbf{z}_{1,n}, \mathbf{z}_{2,m}, \mathbf{x}_{1,n} - \mathbf{x}_{2,m})\mathbf{z}_{2,m}{}^T\mathbf{W}_{V,h}\right\}_{h=1}^{H}\right)\mathbf{W}_O. \tag{13}$$

In Ashman et al. (2024a), the authors also propose optional translation equivariant updates to the inputs, but we do not use them in our implementation. Together with layer normalisation and pointwise MLPs, the TE-MHSA and TE-MHCA operations form the TE-MHSA and TE-MHCA blocks. The TE-TNP (TNP ($T$)) shares an identical architecture to the regular TNP, with the MHSA and MHCA blocks replaced by TE-MHSA and TE-MHCA blocks.

**Translation Equivariant Gridded TNPs**   The architecture of Swin-TNP ($T$) is similar to that of Swin-TNP, with MHSA / MHCA blocks replaced by TE-MHSA / TE-MHCA blocks:

1. Pseudo-token grid encoder: we follow the definitions from Equation 4, but the pseudo-tokens are shared for each grid point, and $\{\mathbf{z}_{c,n}\}_{n=1}^{N_c}$ only depend on output values $\{\mathbf{y}_{c,n}\}_{n=1}^{N_c}$.

$$\mathbf{u}_m \leftarrow \text{TE-MHCA}\left(\mathbf{u}_m^0, \{\mathbf{z}_{c,n}\}_{n \in \mathfrak{N}(\mathbf{v}_m;k)}\right) \quad \forall m \in M. \tag{14}$$

2. Grid processor: We use a Swin Transformer augmented with a fixed set of learnable relative positional encodings. The number of these encodings is equal to the number of grid points multiplied by the number of heads in the attention mechanism. Their introduction comes with negligible computational cost.

3. Pseudo-token grid decoder: similarly to Equation 5 we perform nearest-neighbour cross-attention, but replace the MHCA blocks with TE-MHCA blocks.

$$\mathbf{z}_{t,n} \leftarrow \text{TE-MHCA}\left(\mathbf{z}_{t,n}^0, \{\mathbf{u}_m\}_{m \in \tilde{\mathfrak{N}}(\mathbf{x}_{t,n};k)}\right). \tag{15}$$

### B.2. Approximately Equivariant NPs

Although introducing inductive biases such as translation equivariance has been shown to improve sample efficiency and the ability to generalise, real-world data is rarely perfectly symmetric. To address this, Ashman et al. (2024b) propose Approximately Equivariant Neural Processes, which relax strict equivariance assumptions within the training domain in a data-driven manner, while maintaining equivariance out-of-distribution to allow for good generalisation capabilities.

The main insight towards achieving approximate equivariance is outlined in Theorems 2 and 3 in Ashman et al. (2024b), indicating that any non-equivariant operator between function spaces can be constructed as, or approximated by, an equivariant mapping with additional, fixed functions as input. They use this result to construct the approximately equivariant neural process, which, following the unifying construction in Appendix A, can be constructed as a G-equivariant CNP with encoder $e \colon \mathcal{X} \times \mathcal{Y} \to \mathcal{Z}$, a G-invariant processor $\rho \colon \left(\bigcup_{n=0}^{\infty} \mathcal{Z}^n\right) \times \mathcal{X} \to \mathcal{Z}$, and a decoder $d \colon \mathcal{Z} \to \mathcal{P}_\mathcal{Y}$. Approximate

equivariance is achieved by inserting $B$ fixed inputs $\{t_i\}_{i=1}^B$ into the processor, which become additional model parameters that break the G-invariance of the processor:

$$p(\mathbf{Y}_t|\mathbf{X}_t, D_c) = \prod_{n=1}^{N_t} p(\mathbf{y}_{t,n}|\mathbf{x}_{t,n}, D_c) = \prod_{n=1}^{N_t} p(\mathbf{y}_{t,n}|d(\rho(e(D_c), \mathbf{x}_{t,n}, t_1, \ldots, t_B))). \tag{16}$$

The authors argue that the number of additional inputs determines the degree to which the model breaks equivariance, with more inputs resulting in a more non-equivariant processor. In the limit $B \to \infty$ we eventually recover any non-equivariant processor.

There are various ways in which these additional inputs can be incorporated into G-equivariant architectures. Here, we only outline how they are included into the translation equivariant TNP ($T$). The authors consider fixed functions on $\{\mathbf{x}_{c,n}\}_{n=1}^{N_c}$ which are summed to the initial context tokens $\{\mathbf{z}_{c,n}\}_{n=1}^{N_c}$. These are obtained by performing a Fourier expansion on the context token locations, followed by passing the result through an MLP with two hidden layers of dimension $D_z$.

The authors introduce two additional techniques to ensure generalisation beyond the training distribution. These are motivated by the idea that we only have information about local symmetry-breaking features inside the training domain; outside, where no data are available, it is more reliable to assume strict equivariance. To determine the most suitable equivariant mapping that should be applied outside the training regime, the authors propose the following view: Any approximately equivariant mapping can be decomposed into an equivariant component, which is, in some sense, the best "equivariant approximation" of the original mapping, and a residual non-equivariant component. The latter can then be approximated by an equivariant mapping with additional fixed inputs (according to Theorems 2 and 3). Thus, they propose the following:

1. During training, they set the additional basis functions to zero with some fixed probability. This allows the models to learn a good approximation of the equivariant component of the underlying system when the additional inputs are zero.
2. Forcefully set the additional fixed inputs to zero outside the training domain.

In our implementation of the Swin-TNP ($\widetilde{T}$), we only add additional fixed inputs to the initial context set embeddings—before feeding $\{\mathbf{z}_{c,n}\}_{n=1}^{N_c}$ into the pseudo-token grid encoder, we perform:

$$\mathbf{z}_{c,n} = \phi(\mathbf{y}_{c,n}) + t(\mathbf{x}_{c,n}), \tag{17}$$

where $\phi$ is implemented with an MLP, and $t(\mathbf{x}_{c,n})$ is obtained by performing the Fourier expansion, as mentioned above.

## C. Related Work

**Models for Structured Weather Data**   As mentioned in Section 3, recently there has been significant interest from the research community in developing models that are able to scale to massive spatio-temporal datasets—some of the most successful of which can be seen in weather modelling. The majority operate on structures, gridded data, and are only trained to perform forecasting. Models such as Aurora (Bodnar et al., 2025), GraphCast (Lam et al., 2023), GenCast (Price et al., 2025), Pangu (Bi et al., 2023), FuXi (Chen et al., 2023b) and FengWu (Chen et al., 2023a) share a similar encoder-processor-decoder to our construction of CNPs presented in Section 2.1, in which the input grid is projected onto some latent space which is then transformed using an efficient form of information propagation (e.g. sparse attention with Swin Transformer (Bodnar et al., 2025; Bi et al., 2023), message passing with GNNs (Price et al., 2025)) before being projected back onto the grid at the output.

At the same time, there has been a line of work aiming to develop more flexible methods for spatio-temporal data, either by enabling models to deal with unstructured data, or by adjusting the training task. Perhaps the most relevant works to ours are Aardvark (Allen et al., 2025) and FuXi-DA (Xu et al., 2025), both end-to-end weather prediction models that handle both unstructured and structured data. Aardvark employs kernel interpolation, also known as a SetConv (Gordon et al., 2020), to move unstructured data onto a grid, followed by a ViT which processes this grid and outputs gridded predictions, whereas FuXi-DA simply averages observations within each grid cell. We provide an extensive comparison between the kernel interpolation approach to structuring data and our pseudo-token grid encoder in Section 5, demonstrating significantly better performance. Additionally, in the GP experiment, we evaluate our encoder against FuXi-DA's encoder, with the results in Table 3 showing superior performance.

Another relevant example is Lessig et al. (2023), a task-agnostic stochastic model of atmospheric dynamics, trained to predict randomly masked or distorted tokens. While the task agnosticism of this approach shares similarities with that of

NPs, it is unclear how this approach extends to settings in which data can exist at arbitrary spatio-temporal locations, and they are more limited in their approach to model multiple sources of the input data.

Finally, Schmude et al. (2024) propose Prithvi WxC, a foundation model for weather and climate that can be effectively tuned for multiple distinct use cases. To this end, they employ a training objective that combines ideas from masked reconstruction with forecasting, whereby they predict a future state with a variable lead time, based on masked current and past states. While able to adapt to various downstream tasks through fine-tuning (which generally requires additional architectural components), the backbone transformer is only able to operate on gridded data. The architecture is based on a 2D vision transformer that combines approaches from Hiera (Ryali et al., 2023) and MaxVit (Tu et al., 2022). This implies that, as it is, the model is not able to handle data at arbitrary spatio-temporal locations, and would need architectural adaptations to do so. Indeed, our grid encoder could represent a solution to move the initial unstructured data onto a structured grid, compatible with the foundation model's backbone.

**Connections between our method and GNNs** It is well known that regular transformers can be interpreted as GNNs (Bronstein et al., 2021); in particular, fully-connected graph attention networks (GAT; Veličković et al. 2018). This interpretation extends to our construction of the transformer processor. In particular, we can interpret the grid encoder as dynamically constructing a graph connecting the tokenised observations to their nearest neighbours on some latent grid of pseudo-tokens with a directed edge. The operation of our pseudo-token grid encoder is then a single GAT update. The grid processor can be interpreted as a fully-connected GAT when a ViT is used, or a sparsely-connected GAT when a Swin Transformer is used. Finally, similar to the interpretation of the grid encoder, the grid decoder can be interpreted as dynamically constructing a graph connecting the tokenised target locations to their nearest neighbours on the same latent grid of pseudo-tokens with a (reversed) directed edge, and performing a single GAT update.

Given these connections, it is reasonable to question why we chose to use attention-based updates rather than other forms of GNN updates, such as message passing. Our response is straightforward: empirical evidence suggests that transformer-based implementations achieve superior computational efficiency compared to GNN counterparts for which operations cannot be performed in parallel (Veličković et al., 2018). Furthermore, the widespread adoption of transformers has led to transformer-specific operations, such as the attention mechanism, being highly optimised for modern GPU architectures (Dao et al., 2022; Dao, 2024; Pagliardini et al., 2023).

**Universal Physics Transformers** The recently proposed universal physics transformers (UPTs; **?**) share similar motivations and methodologies as our work. UPTs consider a different, but related problem of learning approximate solutions to partial differential equations (PDEs), which can be formulated as learning mappings between function spaces. The broader family of models which achieve this are known as neural operators (Kovachki et al., 2023). In practice, neural operators work with a finite number of function evaluations according to some discretisation scheme. Dependent on the choice of discretisation scheme, the set of function evaluations may not lie on a regularly spaced grid or mesh, and when the number of function evaluations is large we arrive at a similar problem to that considered in this work. UPTs use a single message passing layer (Gilmer et al., 2017) to local neighbourhoods of tokenised representations of the discretised function evaluations into a set of pseudo-tokens. The primary difference between our pseudo-token grid encoder and this approach is that in UPTs the pseudo-tokens are not constrained to lie on a regular grid; rather, the pseudo-locations are randomly sampled from the original set of discretised locations. Their approach is motivated by the desiderata for the original discretisation structure to be preserved; however, it prevents application of efficient grid-based transformer architectures such as Swin Transformer.

# D. Kernel-Interpolation Grid Encoder

Let $\mathbf{z}_n \in \mathbb{R}^{D_z}$ denote the token representation of input-output pair $(\mathbf{x}_n, \mathbf{y}_n)$ after point-wise embedding. We introduce the set of grid locations $\mathbf{V} \in \mathbb{R}^{M_1 \times \cdots M_{D_x} \times D_x}$. For ease of reading, we shall replace the product $\prod_{d=1}^{D_x} M_d$ with $M$ and the indexing notation $m_1, \ldots, m_{D_x}$ with $m$.

The kernel-interpolation grid encoder obtains a pseudo-token representation $\mathbf{U} \in \mathbb{R}^{M \times D_z}$ of $\mathcal{D}_c$ on the grid $\mathbf{V}$ by interpolating from *all* tokens $\{\mathbf{z}_{c,n}\}_n$ at corresponding locations $\{\mathbf{x}_{c,n}\}_n$ to *all* pseudo-token locations:

$$\mathbf{u}_m \leftarrow \sum_n \mathbf{z}_{c,n} \psi(\mathbf{v}_m, \mathbf{x}_{c,n}) \quad \forall m \in M. \tag{18}$$

Here, $\psi\colon \mathcal{X} \times \mathcal{X} \to \mathbb{R}$ is the kernel used for interpolation, which we take to be the squared-exponential (SE) kernel when

$\mathcal{X} = \mathbb{R}^{D_x}$:

$$\psi_{SE}(\mathbf{v}_m, \mathbf{x}_{c,n}) = \exp\left(-\sum_{d=1}^{D_x} \frac{(x_{c,n,d} - v_{m,d})^2}{\ell_d^2}\right) \tag{19}$$

where $\ell_d$ denotes the 'lengthscale' for dimension $d$. Similar to the pseudo-token grid encoder, we can restrict the kernel-interpolation grid encoder to interpolate only from sets of token $\{\mathbf{z}_{c,n}\}_{n \in \mathfrak{N}(\mathbf{v}_m;k)}$ for which $\mathbf{v}_m$ is amongst the $k$ nearest grid locations. The computational complexity of the kernel-interpolation and pseudo-token grid encoders differ only by a scale factor when this approach is used.

## E. Nearest-Neighbour Cross-Attention in the Grid Decoder

In this section we provide more details, alongside schematics, of our nearest-neighbour cross-attention scheme. As mentioned in the main text, in the grid decoder we only allow a subset of the gridded pseudo-tokens to attend to the target token (those in the vicinity of it). Finding the nearest neighbours is not done in the standard k-nearest neighbours fashion, because we use the same number of nearest neighbours along each dimension of the original data (i.e. latitude and longitude for spatial interpolation; latitude, longitude and time for spatio-temporal interpolation). This is to ensure that we do not introduce specific preferences for any dimension. In the case the data lies on a grid with the same spacing in all its dimensions, the procedure becomes equivalent to k-nearest neighbours.

In practice, we specify the total number of nearest-neighbours we want to use $k$. We then compute the number of nearest-neighbours in each dimension by $k_{\text{dim}} = \text{ceil}(k^{\frac{1}{\dim(x)}})$, where $\dim(x)$ represents the dimensionality of the input. For efficient batching purposes, we tend to choose $k = (2n-1)^{\dim(x)}$, where $n \in \mathbb{N}$ (i.e. 9 for experiments with latitude-longitude grids, 27 for experiments with latitude-longitude-time grids). We then find the indices in each dimension of these nearest-neighbours by performing an efficient search that leverages the gridded nature of the data, leading to a computational complexity of $\mathcal{O}(kN_t)$, with $N_t$ the number of target points. When the neighbours go off the grid (i.e. for targets very close to the edges of the grid), we only consider the number of viable (i.e. within the bounds of the grid) neighbours.

**Example in 2D** This procedure is visualised in Figure 9, where we consider both grids with the same spacing along each dimension, as well as grids with different spacings. We cover both the case of a central target point, as well as a target point closer to the edges of the grid.

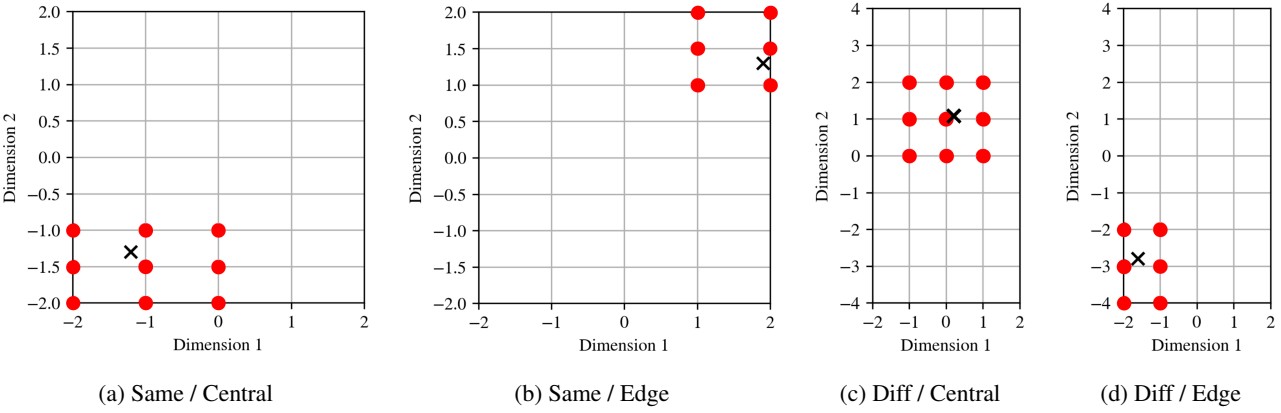

|  |  |  |  |
|---|---|---|---|
| (a) Same / Central | (b) Same / Edge | (c) Diff / Central | (d) Diff / Edge |

*Figure 9.* Example of our nearest-neighbours procedure in 2D on a $5 \times 9$ grid for 9 nearest neighbours. We consider four different cases. Same / Diff refers to whether the grid spacing is the same in the two dimensions or different. Central / Edge refers to the position of the target. In the case of an edge target, we do not consider invalid neighbours (i.e. those that are outside the grid bounds).

**Accounting for non-Euclidean geometry** A lot of our experiments are performed on environmental data, distributed across the Earth. For our purposes, we assume the Earth shows cylindrical geometry, whereby there is no such thing as a grid edge along the longitudinal direction (i.e. a longitude of -180° is the same as 180°). This is not the same for latitude, where one extreme corresponds to the North Pole, and the other one to the South Pole. Thus, we would like to allow for the grid to 'roll' around the longitudinal direction when computing the nearest neighbours. In this case, there should be no edge

target points along the longitudinal dimension. This procedure is graphically depicted in Figure 10 and we use it in our experiments on environmental data.

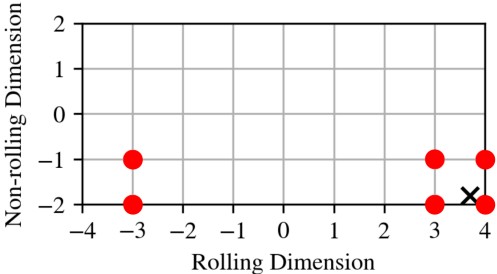

*Figure 10.* Example of nearest-neighbours procedure in 2D on a $5 \times 9$ grid for 9 nearest neighbours. We allow rolling along the horizontal dimension (e.g. longitude), but do not allow rolling along the vertical one (i.e. latitude). Hence, the neighbours extend on the other side of the grid horizontally, but not vertically. This example corresponds to cylindrical symmetry.

**Example for 3D data** We also provide examples of the nearest-neighbours procedure on 3D spaces. These dimensions could represent, for example, latitude, longitude and time, or latitude, longitude and height/pressure levels. In Figure 11 we consider a case where we do not allow for rolling along any dimension, while in Figure 12 we allow for rolling along one of the dimensions.

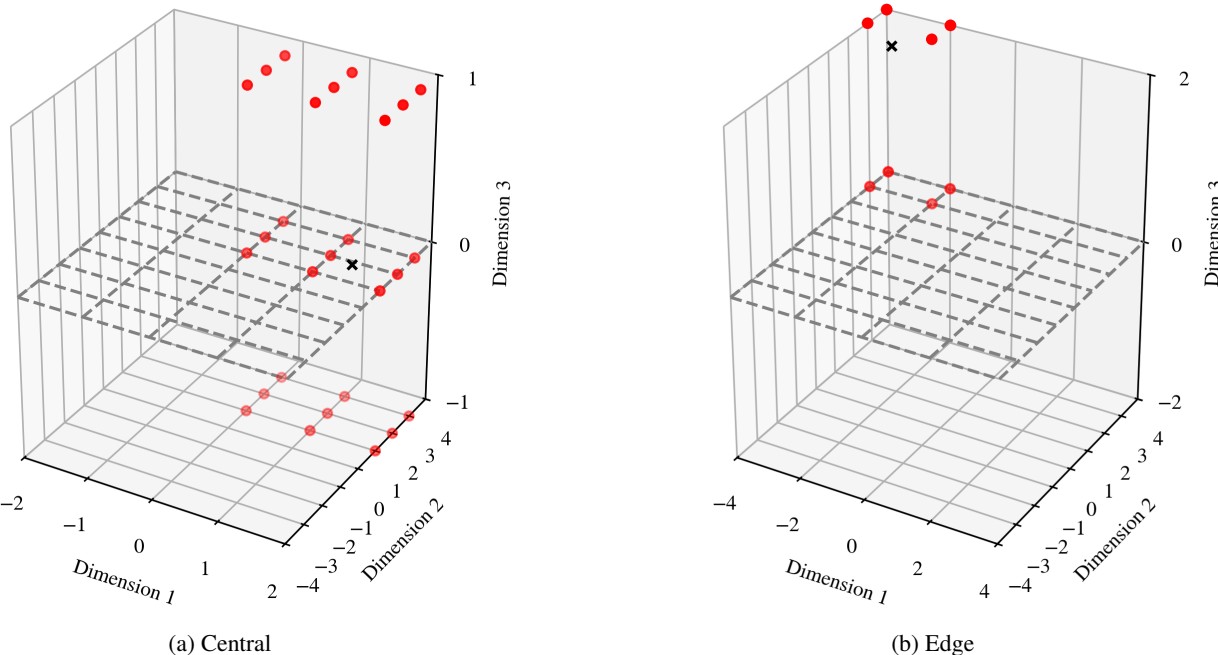

(a) Central

(b) Edge

*Figure 11.* Example of our nearest-neighbours procedure in 3D on a $5 \times 9 \times 3$ grid for 27 nearest neighbours. We consider two different cases—whether the target is central or near the edge of the grid. In the case of an edge target (right), we do not consider invalid neighbours (that are outside the grid bounds).

## F. Hardware specifications

For the smaller synthetic GP regression experiment, we perform training and inference for all models on a single NVIDIA GeForce RTX 2080 Ti GPU with 20 CPU cores. For the other two, larger experiments, we perform training and inference for all models on a single NVIDIA A100 80GB GPU with 32 CPU cores.

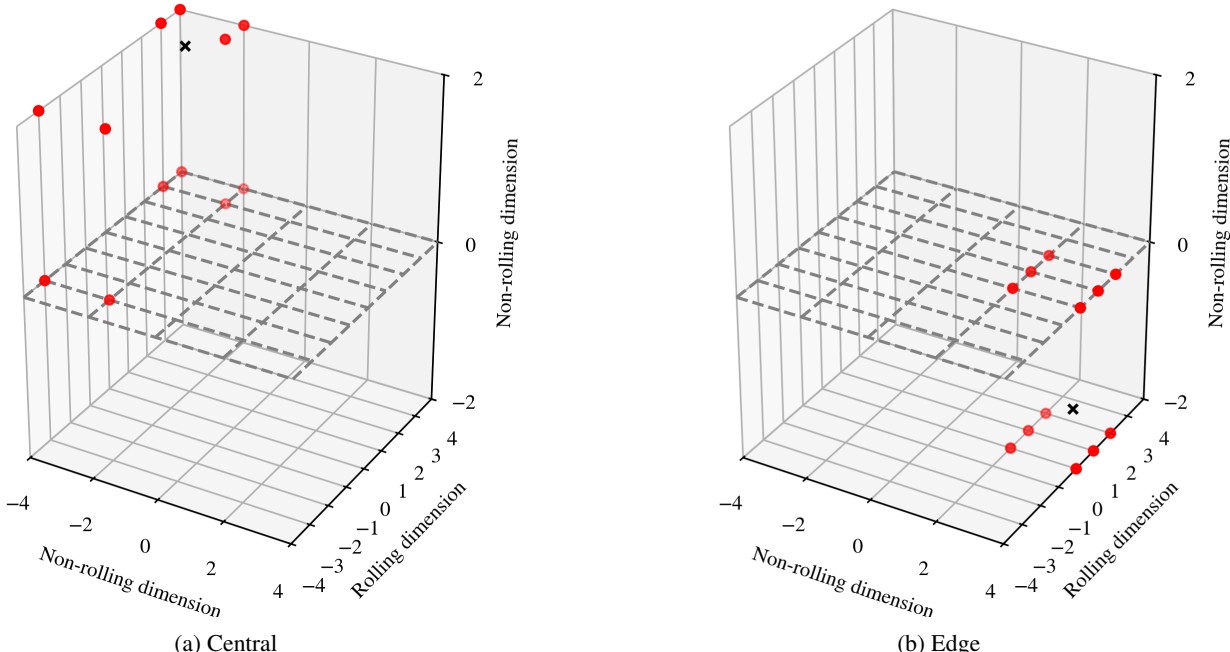

(a) Central                                             (b) Edge

*Figure 12.* Example of our nearest-neighbours procedure in 3D on a $5 \times 9 \times 3$ grid for 27 nearest neighbours. We allow for rolling along the second dimension, but not along any of the other ones.

## G. Experiment Details

**Common optimiser details**   For all experiments and all models, we use the AdamW optimiser (Loshchilov & Hutter, 2019) with a fixed learning rate of $5 \times 10^{-4}$ and apply gradient clipping to gradients with magnitude greater than 0.5.

**Common likelihood details**   For all experiments and all models, we employ a Gaussian likelihood parameterised by a mean and inverse-softplus variance, i.e. the decoder of each model outputs

$$\boldsymbol{\mu}_t, \ \log(\exp \boldsymbol{\sigma}_t^2 - 1) = d(\mathbf{z}_t), \quad p(\cdot \mid \mathcal{D}_c, \mathbf{x}_t) = \mathcal{N}(\cdot; \boldsymbol{\mu}_t, \boldsymbol{\sigma}_t^2). \tag{20}$$

For the experiment modelling skin and 2m temperature (`skt` and `t2m`) with a richer context, we set a minimum noise level of $\boldsymbol{\sigma}_{\min}^2 = 0.01$ by parameterising

$$\boldsymbol{\mu}_t, \ \log(\exp (\boldsymbol{\sigma}_t - \boldsymbol{\sigma}_{\min})^2 - 1) = d(\mathbf{z}_t), \quad p(\cdot \mid \mathcal{D}_c, \mathbf{x}_t) = \mathcal{N}(\cdot; \boldsymbol{\mu}_t, \boldsymbol{\sigma}_t^2). \tag{21}$$

**CNP details**   For the CNPs, we encode each $(\mathbf{x}_{c,n}, \mathbf{y}_{c,n}) \in \mathcal{D}_c$ in $\mathbb{R}^{D_z}$ using an MLP with two-hidden layers of dimension $D_z$. We obtain a representation for the entire context set by summing these representations together, $\mathbf{z}_c = \sum_n \mathbf{z}_{c,n}$, which is then concatenated with the target input $\mathbf{x}_t$. The concatenation $[\mathbf{z}_c, \mathbf{x}_t]$ is decoded using an MLP with two-hidden layers of dimension $D_z$. We use $D_z = 128$ in all experiments.

**ConvCNP details**   For the ConvCNP model, we use a U-Net architecture (Ronneberger et al., 2015) for the CNN consisting of 11 layers with input size $C$. Between the five downward layers we apply pooling with size two. For the five upward layers, we use $2C$ input channels and $C$ output channels, as the input channels are formed from the output of the previous layer concatenated with the output of the corresponding downward layer. Between the upward layers we apply linear up-sampling to match the grid size of the downward layer. In all experiments, we use $C = 128$, a kernel size of five or nine, and a stride of one. We use SE kernels for the SetConv encoder and SetConv decoder with learnable lengthscales for each input dimension. The grid encoding is modified similarly to the pseudo-token grid encoder, whereby we only interpolate from the set of observations for which each grid point is the closest grid point. Unless otherwise specified, we also modify the grid decoding similarly to the pseudo-token grid decoder, whereby we only interpolate from the $k = 3^{D_x}$ nearest points on a

distance-normalised grid to each target location. We resize the output of the SetConv encoder to dimension $C$ using an MLP with two hidden layers of dimension $C$. We resize the output of the SetConv decoder using an MLP with two hidden layers of dimension $C$.

**Common transformer details**   For each MHSA / MHCA operation, we construct a layer consisting of two residual connections, two layer norm operations, one MLP, together with the MHSA / MHCA operation as follows:

$$\widetilde{\mathbf{Z}} \leftarrow \mathbf{Z} + \text{MHSA/MHCA}(\text{layer-norm}_1(\mathbf{Z}))$$
$$\mathbf{Z} \leftarrow \widetilde{\mathbf{Z}} + \text{MLP}(\text{layer-norm}_2(\widetilde{\mathbf{Z}})). \tag{22}$$

All MHSA / MHCA operations use $H = 8$ heads, each with $D_V = 16$ dimensions. We use a $D_z = D_{QK} = 128$ throughout.

**PT-TNP details**   We use an induced set transformer (IST) architecture for the PT-TNPs, with each layer consisting of the following set of operations:

$$\mathbf{U} \leftarrow \text{MHCA-layer}(\mathbf{U}; \mathbf{Z}_c)$$
$$\mathbf{z}_t \leftarrow \text{MHCA-layer}(\mathbf{z}_t; \mathbf{U}) \tag{23}$$
$$\mathbf{Z}_c \leftarrow \text{MHCA-layer}(\mathbf{Z}_c; \mathbf{U}).$$

In all experiments we use five layers, and encoder / decoder MLPs consisting of two hidden layers of dimension $D_z$.

**ViT details**   The ViT architecture consists of optional patch encoding, followed by five MHSA layers. The patch encoding is implemented using a single linear layer. In all experiments we use five layers, and encoder / decoder MLPs consisting of two hidden layers of dimension $D_z$.

**Swin Transformer details**   Each layer of the Swin Transformer consists of two MHSA layers applied to each window, and a shifting operation between them. Unless otherwise specified, we use a window size of four and shift size of two for all dimensions (except for the time dimension in the final experiment, as the original grid only has four elements in the time dimension). For the second experiment in which the grid covers the entire globe, we allow the Swin attention masks to 'roll' over the longitudinal dimension, allowing the pseudo-tokens near $180°$ longitude to attend to those near $-180°$ longitude. In all experiments, we use five Swin Transformer layers (10 MHSA layers in total), and encoder / decoder MLPs consisting of two hidden layers of dimension $D_z$. We found that the use of a hierarchical Swin Transformer—as used in the original Swin Transformer and models such as Aurora (Bodnar et al., 2025)—did not lead to any improvement in performance.

**Spherical harmonic embeddings**   When modelling input data on the sphere (i.e. the final two experiments), the CNP, PT-TNP, and gridded TNP models first encode the latitude / longitude coordinates using spherical harmonic embeddings following (Rußwurm et al., 2024) using 10 Legendre polynomials. We found this to improve performance in all cases.

**Temporal Fourier embeddings**   When modelling input data through time (i.e. the final experiment), the CNP, PT-TNP, and gridded TNP models first encode the temporal coordinates using a Fourier embedding. The time value is originally provided in hours since 1st January 1970. Following (Bodnar et al., 2025), we embed this using a Fourier embedding of the following form:

$$\text{Emb}(t) = \left[ \cos \frac{2\pi t}{\lambda_i},\ \sin \frac{2\pi t}{\lambda_i} \right] \text{ for } 0 \le i < L/2. \tag{24}$$

where the $\lambda_i$ are log-spaced values between the minimum and maximum wavelength. We set $\lambda_{min} = 1$ and $\lambda_{max} = 8760$, the number of hours in a year. We use $L = 10$.

**Great-circle distance**   For methods using the kernel-interpolation grid encoder (i.e. the ConvCNP and some gridded TNPs), we use the great-circle distance, rather than Euclidean distance, as the input into the kernel when modelling input data on the sphere. The haversine formula determines the great-circle distance between two points $\mathbf{x}_1 = (\varphi_1, \lambda_2)$ and $\mathbf{x}_2 = (\varphi_2, \lambda_2)$, where $\lambda$ and $\varphi$ denote the latitude and longitude, and is given by:

$$d(\mathbf{x}_1, \mathbf{x}_2) = 2r \arcsin \left( \sqrt{\frac{1 - \cos(\Delta\varphi) + \cos\varphi_1 \cdot \cos\varphi_2 \cdot (1 - \cos(\Delta\lambda))}{2}} \right) \tag{25}$$

where $\Delta\varphi = \varphi_2 - \varphi_1$, $\Delta\lambda = \lambda_2 - \lambda_1$ and $r$ is taken to be 1.

## G.1. Meta-Learning Gaussian Process Regression

We utilise the `GPyTorch` software package (Gardner et al., 2018) for generating synthetic samples from a GP. As the number of datapoints in each sampled dataset is very large by GP standards ($1.1 \times 10^4$), we approximate the SE kernel using structured kernel interpolation (SKI) (Wilson & Nickisch, 2015) with 100 grid points in each dimension. We use an observation noise of $\sigma_n = 0.1$ for the smaller and larger lengthscale tasks. In Figure 13, we show an example dataset generated using a lengthscale of $0.1$ to demonstrate the complexity of these datasets. We were unable to compute ground truth log-likelihood values for these datasets without running into numerical issues.

In addition, we also plot the predictive means (Figure 14) and predictive errors (Figure 15) in the form of heatmaps for a number of CNP models on a different example dataset.

We train all models for $500,000$ iterations on $160,000$ pre-generated datasets using a batch size of eight. For all models, excluding the ConvCNP, we apply Fourier embeddings to each input dimension with $L = 64$ wavelengths with $\lambda_{min} = 0.01$ and $\lambda_{max} = 12$. We found this to significantly improve the performance of all models.

We provide a plot of the test log-likelihood against the forward pass time in Figure 16, where we include both the larger models shown in Figure 3, as well as smaller versions. In Table 3, we provide test log-likelihood values for a number of gridded TNPs and baselines for both tasks. We observe that even when increasing the size of the baseline models they still underperform the smaller gridded TNPs. We include results for the Swin-TNP with full attention grid decoding (no nearest-neighbour cross-attention indicated as N̶N̶), which fail to model the more complex dataset when using the PT-GE.

*Table 3.* Test log-likelihood ($\uparrow$) for the synthetic GP regression dataset. N̶N̶ indicates that NN-CA is not employed. FPT: forward pass time for a batch size of eight in ms. Params: number of model parameters in units of M.

| Model | Grid encoder | Grid size | $\ell = 0.5$ ($\uparrow$) | $\ell = 0.1$ ($\uparrow$) | FPT | Params |
|---|---|---|---|---|---|---|
| CNP | - | - | $-0.406$ | 0.112 | 9 | 0.21 |
| PT-TNP | - | $M = 128$ | 0.819 | 0.558 | 53 | 1.50 |
| PT-TNP | - | $M = 256$ | 0.819 | 0.565 | 74 | 1.52 |
| ConvCNP | SetConv | $32 \times 32$ | 0.801 | 0.536 | 13 | 2.11 |
| ConvCNP | SetConv | $64 \times 64$ | 0.830 | 0.681 | 93 | 6.70 |
| ViTNP | KI-GE | $32 \times 32 \to 16 \times 16$ | 0.841 | 0.722 | 30 | 1.16 |
| ViTNP | PT-GE | $32 \times 32 \to 16 \times 16$ | 0.841 | 0.721 | 32 | 1.39 |
| ViTNP | KI-GE | $16 \times 16$ | 0.833 | 0.711 | 28 | 1.09 |
| ViTNP | PT-GE | $16 \times 16$ | 0.840 | 0.712 | 29 | 1.22 |
| ViTNP | KI-GE | $64 \times 64 \to 32 \times 32$ | 0.842 | 0.728 | 44 | 1.16 |
| ViTNP | PT-GE | $64 \times 64 \to 32 \times 32$ | 0.836 | 0.727 | 56 | 1.78 |
| ViTNP | KI-GE | $32 \times 32$ | 0.830 | 0.725 | 47 | 1.09 |
| ViTNP | PT-GE | $32 \times 32$ | 0.837 | 0.728 | 53 | 1.32 |
| Swin-TNP | KI-GE | $32 \times 32$ | 0.844 | 0.723 | 39 | 1.09 |
| Swin-TNP | PT-GE | $32 \times 32$ | 0.844 | 0.723 | 42 | 1.32 |
| Swin-TNP | Avg-GE | $32 \times 32$ | 0.840 | 0.723 | 34 | 1.22 |
| Swin-TNP | KI-GE | $64 \times 64$ | 0.846 | 0.728 | 62 | 1.09 |
| Swin-TNP | PT-GE | $64 \times 64$ | **0.847** | **0.730** | 69 | 1.72 |
| Swin-TNP | Avg-GE | $64 \times 64$ | 0.845 | 0.725 | 58 | 1.62 |
| Swin-TNP N̶N̶ | KI-GE | $32 \times 32$ | 0.834 | 0.716 | 45 | 1.09 |
| Swin-TNP N̶N̶ | PT-GE | $32 \times 32$ | 0.837 | 0.709 | 48 | 1.32 |

**ConvCNP** For the ConvCNP models, we use a regular CNN architecture with $C = 128$ channels and five layers. We use a kernel size of five for the smaller ConvCNP ($32 \times 32$ grid) and a kernel size of nine for the larger ConvCNP ($64 \times 64$).

**Swin-TNP** For the Swin-TNP models, we use a window size of $4 \times 4$ for the smaller model ($32 \times 32$ grid) and a window size of $8 \times 8$ for the larger model ($64 \times 64$ grid). The shift size is half the window size in each case. We also provide results when a simple average pooling is used for the grid encoder, which is similar to Xu et al. 2025 except that the pooling is performed in token space rather than on raw observations.

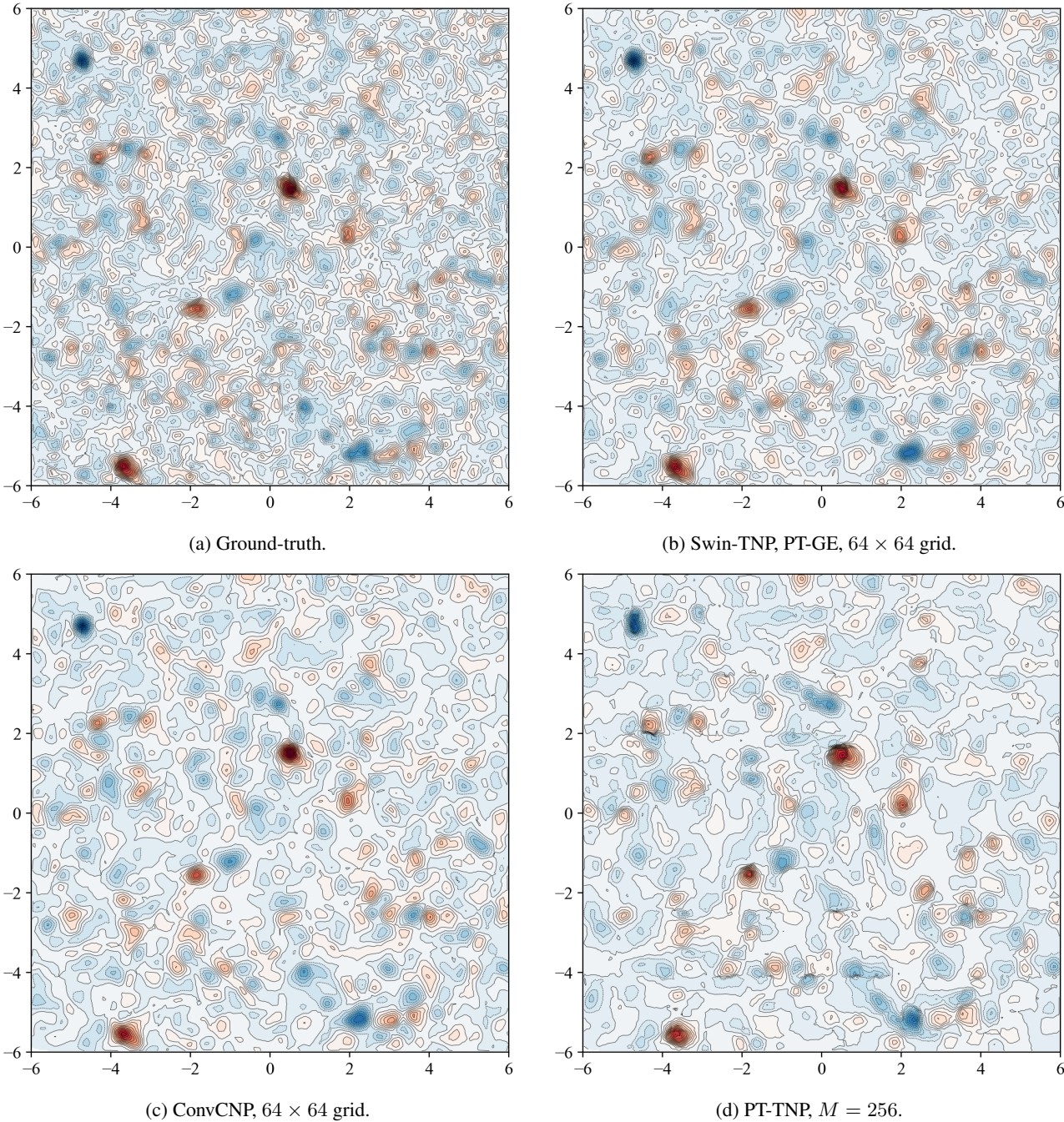

(a) Ground-truth.

(b) Swin-TNP, PT-GE, $64 \times 64$ grid.

(c) ConvCNP, $64 \times 64$ grid.

(d) PT-TNP, $M = 256$.

*Figure 13.* A comparison between the predictive means of a selection of CNP models on a synthetic GP dataset with $\ell = 0.1$. The noiseless ground-truth dataset is shown in Figure 13a, and the context set is a randomly sampled set of $N_c = 1 \times 10^4$ noisy observations of this. The colour corresponds to the output value, with the same scale used in each plot. Observe the complexity of the ground-truth dataset, which the Swin-TNP's predictive mean resembles. The ConvCNP and PT-TNP's predictive means are notably smoother.

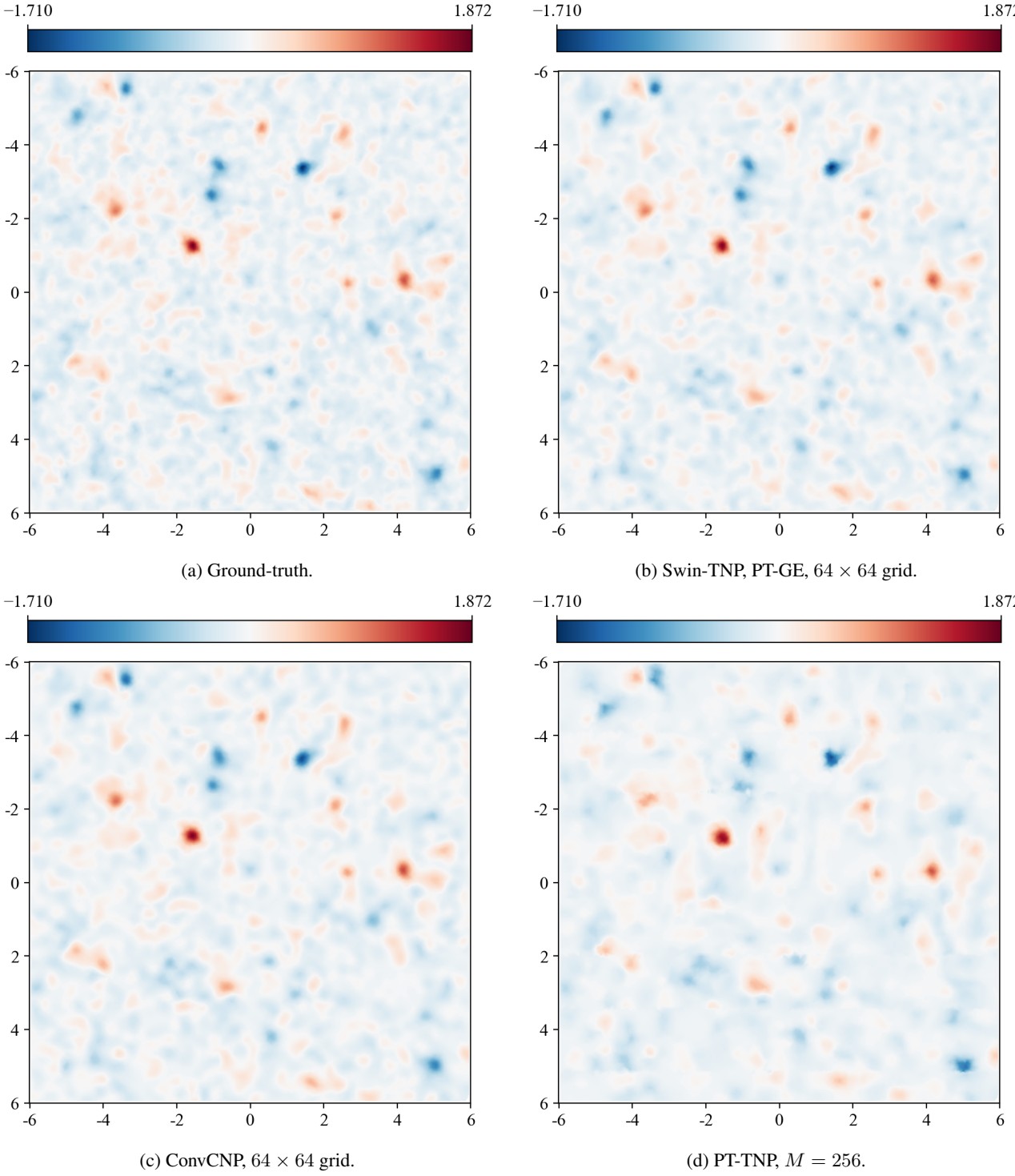

(a) Ground-truth.

(b) Swin-TNP, PT-GE, $64 \times 64$ grid.

(c) ConvCNP, $64 \times 64$ grid.

(d) PT-TNP, $M = 256$.

*Figure 14.* A comparison between the predictive means of a selection of CNP models on a synthetic GP dataset with $\ell = 0.1$. The noiseless ground-truth dataset is shown in Figure 14a, and the context set is a randomly sampled set of $N_c = 1 \times 10^4$ noisy observations of this. The colour corresponds to the output value, with the same scale used in each plot.

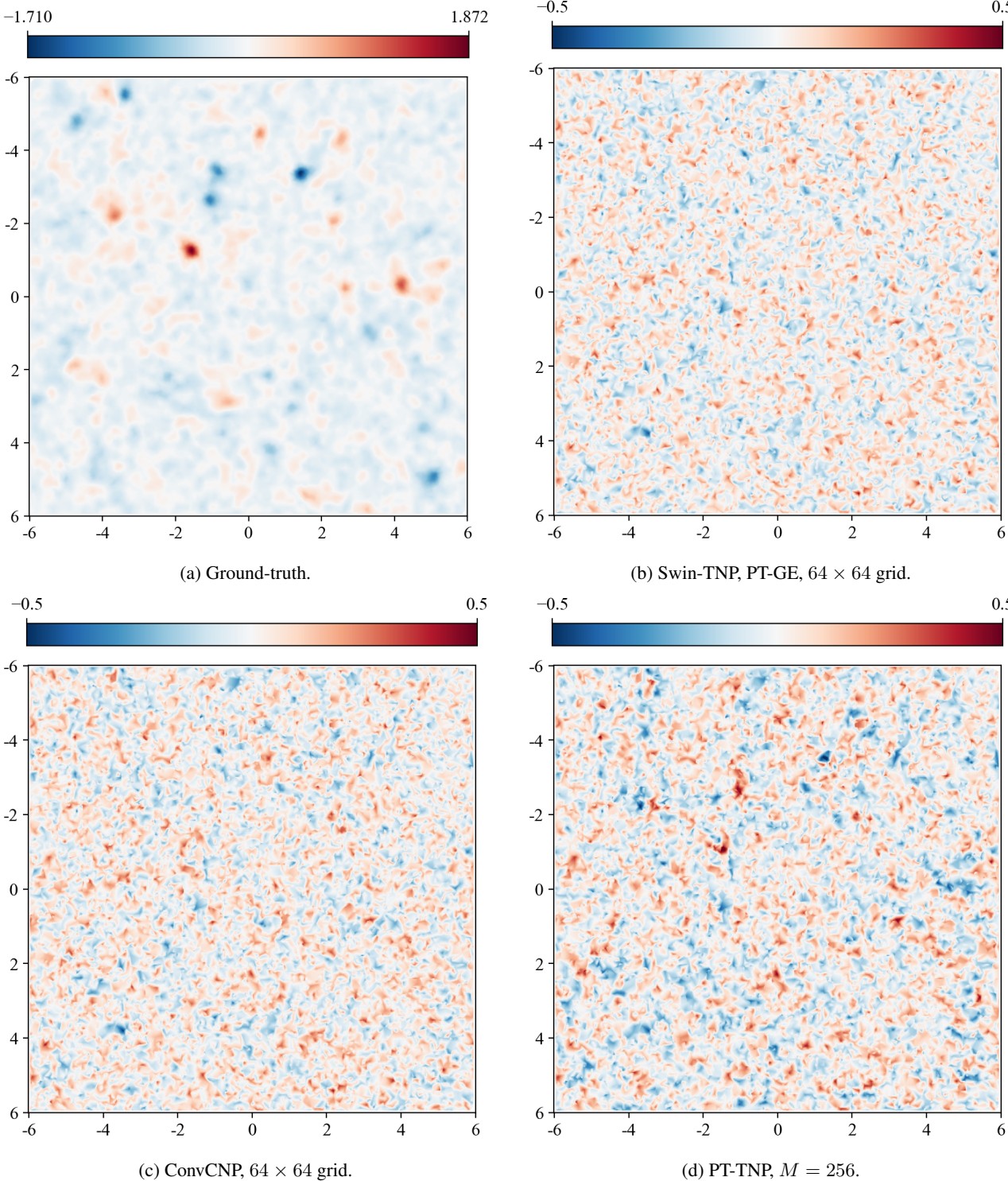

(a) Ground-truth.

(b) Swin-TNP, PT-GE, $64 \times 64$ grid.

(c) ConvCNP, $64 \times 64$ grid.

(d) PT-TNP, $M = 256$.

*Figure 15.* A comparison between the difference between the predictive mean and ground-truth for a selection of CNP models on a synthetic GP dataset with $\ell = 0.1$. The noiseless ground-truth dataset is shown in Figure 15a, and the context set is a randomly sampled set of $N_c = 1 \times 10^4$ noisy observations of this. The colour corresponds to the prediction error, with the same scale used in each plot.

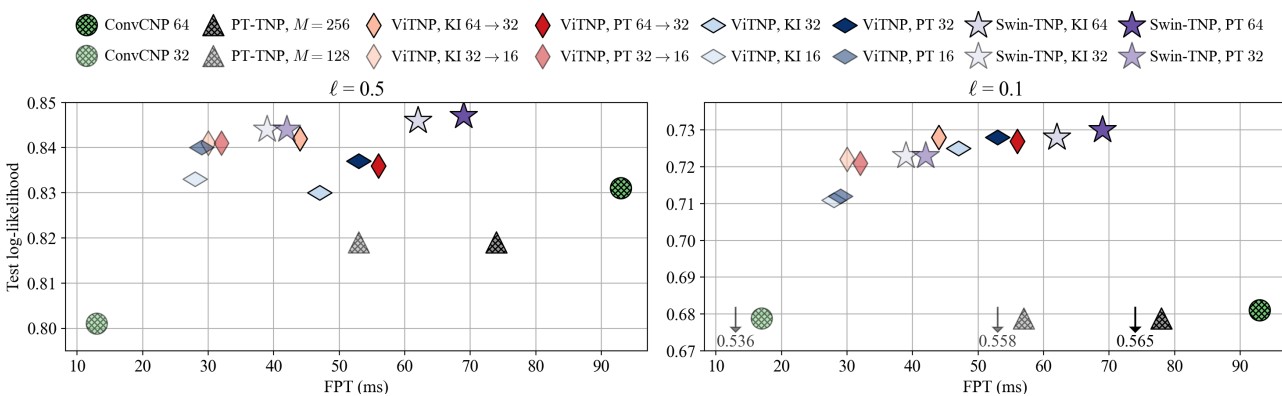

*Figure 16.* Test log-likelihood vs. forward pass time (FPT) for the GP datasets. For each model, we show the results for a large and small (transparent) version. Baselines are hatched. The considered grid sizes are $64 \times 64$ and $32 \times 32$, shown as 64 and 32. For ViTNPs, we include results with and without patch encoding, the former indicated by the $\rightarrow$ symbol in-between the pre- and post-patch-encoded grid sizes. KI / PT: kernel-interpolation / pseudo-token grid encoding.

### G.1.1. SMALL-SCALE META-LEARNING GAUSSIAN PROCESS REGRESSION

We also consider a smaller GP regression task with datasets drawn from a GP with SE kernel with lengthscale $\ell = 0.1$. Each dataset in this smaller task has a randomly sized context set, $N_c \sim \mathcal{U}\{1, 1000\}$, and a fixed sized target set $N_t = 100$. The inputs are sampled uniformly in the range $[-2, 2]$ in each dimension. The use of a smaller dataset allows us to make comparisons with the regular TNP. In Table 4, we compare the performance of the best performing smaller ViTNP and Swin-TNP gridded TNPs from the paper with the regular TNP, implemented with five MHSA layers, token dimension $D_z = 128$, $H = 8$ heads and $D_Q = D_{KV} = 16$. We include the standard error of the mean test log-likelihood, which demonstrate that there is no significant difference in performance between the regular TNP and Swin-TNP. It should be noted, however, that the regular TNP is more computationally efficient than both gridded TNPs for this small-scale dataset. This reflects the suitability of gridded TNPs for large-scale datasets, as there is little difference in forward pass time for the small-scale datasets here and the large-scale datasets considered in the paper for the gridded TNPs. In contrast, the TNP cannot be implemented on the large-scale datasets considered in the paper due to the quadratic computational and memory complexity associated with full attention.

*Table 4.* Test log-likelihood ($\uparrow$) for the synthetic GP regression dataset. FPT: forward pass time for a batch size of eight in ms. Params: number of model parameters in units of M.

| Model | Grid encoder | Grid size | Test log-likelihood ($\uparrow$) | FPT | Params |
|---|---|---|---|---|---|
| TNP | - | - | $\mathbf{-0.596 \pm 0.02}$ | 17 | 0.60 |
| ViTNP | PT-GE | $32 \times 32 \rightarrow 16 \times 16$ | $-0.657 \pm 0.02$ | 22 | 1.39 |
| Swin-TNP | PT-GE | $32 \times 32$ | $\mathbf{-0.616 \pm 0.02}$ | 33 | 1.32 |

### G.2. Combining Weather Station Observations with Structured Reanalysis

Inspired by the real-life assimilation of 2m temperature (t2m), we use the ERA5 reanalysis dataset to extract skin temperature (skt) and 2m temperature (t2m) at a $0.25°$ resolution (corresponding to a $721 \times 1440$ grid). We then coarsen the skt grid to a $180 \times 360$ grid, corresponding to $1°$ in both the latitudinal and longitudinal directions. This implies that, because t2m lies on a finer grid, it essentially becomes an off-the-grid variable with respect to the coarsened grid on which skt lies. In order for the experimental setup to better reflect real-life assimilation conditions, we assume to only observe off-the-grid t2m values at real weather station locations[6]. In total, there are $9,957$ such weather station locations, extracted from the HadISD dataset (Dunn et al., 2012). We show their geographical location in Figure 17.

For each task, we first randomly sample a time point, and then use the entire coarsened skt grid as the on-the-grid context

---

[6]More specifically, because the 2m temperature values come from the gridded ERA5 data, we only consider the nearest grid points to the true station locations as valid off-the-grid locations (i.e. if a station has coordinates at $(44.19°, 115.43°)$ latitude-longitude, we consider the grid point at $(44.25°, 115.5°)$)

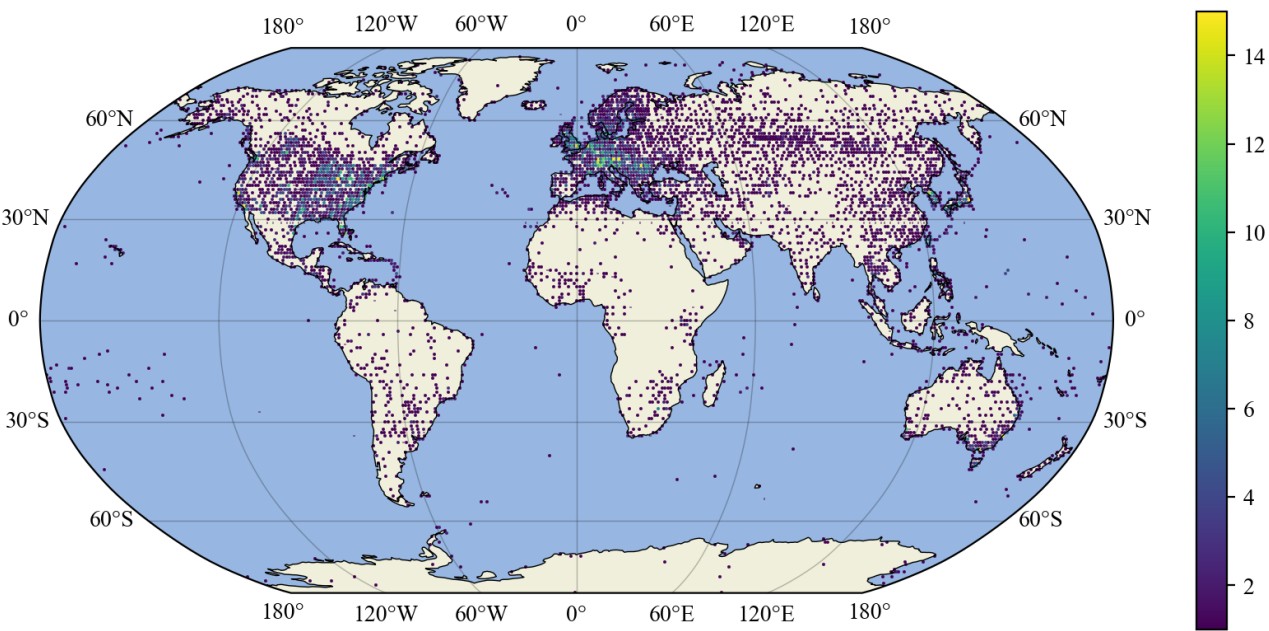

*Figure 17.* Station distribution within $1° \times 1°$ patches. The colour indicates the number of stations within each patch, clipped from a maximum value of 22 to 15. The distribution is far from uniform, with dense station areas in continents such as North America, Europe and parts of Asia, and a sparser distribution in Africa, South America, and in the oceans.

data (64, 800 points), as well as $N_{\text{off},c}$ off-the-grid t2m context points randomly sampled from the station locations. In the experiment from the main paper, $N_{\text{off},c} \sim \mathcal{U}_{[0,0.3]}$, but we also consider the case of richer off-the-grid context sets with $N_{\text{off},c} \sim \mathcal{U}_{[0.25,0.5]}$ in Table 9. The target locations are all the 9, 957 station locations.

We train all models for 300, 000 iterations on the hourly data between $2009 - 2017$ with a batch size of eight. Validation is performed on 2018 and testing on 2019. The test metrics are reported for 16, 000 data samples. Experiment specific architecture choices are described below.

**Input embedding**   We use spherical harmonic embeddings for the latitude / longitude values. These are not used in the ConvCNP model as the ConvCNP does not modify the inputs in order to maintain translation equivariance (in this case, with respect to the great-circle distance).

**Grid sizes**   For the main experiment (with results reported in Table 1), we chose a grid size of $64 \times 128$ for the ConvCNP and Swin-TNP models, corresponding to a grid spacing of $2.8125°$ in both the latitudinal and longitudinal directions.

In Table 9 we report results for a richer context set using a grid size of $128 \times 256$ for the Swin-TNP models, corresponding to a grid spacing of $\approx 1.41°$ in both the latitudinal and longitudinal directions. The results for the ConvCNP are for a grid size of $64 \times 128$, to maintain a smaller gap in parameter count between models.

**CNP**   We use a different deepset for the on- and the off-the-grid data, and the mean as the permutation-invariant function to aggregate the context tokens, i.e. $\mathbf{z}_c = \frac{1}{N_{\text{off},c}} \sum_{n=1}^{N_{\text{off},c}} e_{\text{off}}(\mathbf{x}_{\text{off},c,n}, \mathbf{y}_{\text{off},c,n}) + \frac{1}{N_{\text{on},c}} \sum_{n=1}^{N_{\text{on},c}} e_{\text{on}}(\mathbf{x}_{\text{on},c,n}, \mathbf{y}_{\text{on},c,n})$

**PT-TNP**   We managed to use up to $M = 256$ pseudo-tokens without running into memory issues. This shows that even if we only use two variables (one on- and one off-the-grid), PT-TNPs do not scale well to large data. We use a different encoder for the on- and off-the-grid data, before aggregating the two sets of tokens into a single context set.

**ConvCNP**   For the ConvCNP we use a grid of size $64 \times 128$ for all experiments. We first separately encode both the on- and the off-the-grid to the specified grid size using the SetConv. We then concatenate the two and project them to a

dimension of $C = 128$ before passing through the U-Net (Ronneberger et al., 2015). The U-Net uses a kernel size of $k = 9$ with a stride of one.

**Swin-TNP**  For the Swin-TNP models in the main experiment, we use a grid size of $64 \times 128$, a window size of $4 \times 4$ and a shift size of $2 \times 2$. For the experiment with richer off-the-grid context sets (i.e. between $0.25$ and $0.5$ of the off-the-grid data), we use a grid size of $128 \times 256$, a window size of $8 \times 8$ and a shift size of $4 \times 4$.

**Swin-TNP ($T$)**  For the Swin-TNP ($T$) models we use analogous architectures to Swin-TNP. For the pseudo-token grid encoder and decoder, we parameterise $\{\rho_h\}_{h=1}^{H}$ by an MLP with two hidden layers of dimension $H$ and output dimension $H$, where $H$ is the number of heads used in the multi-head attention mechanism. For the translation equivariant Swin processor, we add a fixed set of learnable relative positional encodings.

**Swin-TNP ($\widetilde{T}$)**  For the Swin-TNP ($\widetilde{T}$) models we use analogous architectures to Swin-TNP ($T$). To break equivariance, we first perform a Fourier expansion on the context token location, and pass the result through an MLP with two hidden layers of dimension $D_z$. We use 16 Fourier coefficients, and drop the fixed inputs out during training with a probability of $0.5$. We do not zero them out as during training we observe the entire $\mathcal{X}$ domain (i.e., the entire world).

### G.2.1. Additional Results for the Main Experiment

**Aggregated experiment results**  We provide in Table 5 the results for the t2m station prediction experiment. The figures correspond to those presented in Table 1, but are aggregated per experiment, and additionally include the root mean squared error (RMSE) metric and the number of parameters associated with each model. Moreover, we also provide results for the Swin-TNP using just the t2m station data (Swin-TNP (t2m)), or just the skt gridded data (Swin-TNP (skt)). The superior performance of the Swin-TNP using both sources of information proves that the model manages to capture meaningful relationships between the two variables.

Table 5. Test log-likelihood ($\uparrow$) and RMSE ($\downarrow$) for the t2m station prediction experiment. The standard errors of the log-likelihood are all below 0.010, and of the RMSE below 0.013. N̶N̶ indicates that NN-CA is not employed. FPT: forward pass time for a batch size of eight in ms. Params: number of model parameters in units of M.

| Model | GE | Grid size | Log-lik. $\uparrow$ | RMSE $\downarrow$ | FPT | Params |
|---|---|---|---|---|---|---|
| CNP | - | - | 0.636 | 3.266 | 32 | 0.34 |
| PT-TNP | - | $M = 256$ | 1.344 | 1.659 | 230 | 1.57 |
| ConvCNP (N̶N̶) | SetConv | $64 \times 128$ | 1.535 | 1.252 | 96 | 9.36 |
| ViTNP | KI-GE | $48 \times 96$ | 1.628 | 1.197 | 167 | 1.14 |
| ViTNP | PT-GE | $48 \times 96$ | 1.704 | 1.118 | 181 | 1.83 |
| ViTNP | KI-GE | $144 \times 288 \to 48 \times 96$ | 1.734 | 1.073 | 171 | 1.29 |
| ViTNP | PT-GE | $144 \times 288 \to 48 \times 96$ | 1.808 | 1.021 | 215 | 6.69 |
| Swin-TNP | KI-GE | $64 \times 128$ | 1.683 | 1.157 | 121 | 1.14 |
| Swin-TNP | PT-GE | $64 \times 128$ | **1.819** | **1.006** | 127 | 2.29 |
| Swin-TNP (N̶N̶) | KI-GE | $64 \times 128$ | 1.544 | 1.436 | 137 | 1.14 |
| Swin-TNP (N̶N̶) | PT-GE | $64 \times 128$ | 1.636 | 1.273 | 144 | 2.29 |
| Swin-TNP (skt) | PT-GE | $64 \times 128$ | 1.427 | 1.330 | 123 | 2.29 |
| Swin-TNP (t2m) | PT-GE | $64 \times 128$ | 1.585 | 1.599 | 107 | 2.24 |
| ConvCNP | SetConv | $192 \times 384$ | 1.689 | 1.166 | 74 | 9.36 |
| Swin-TNP | PT-GE | $192 \times 384$ | **2.053** | **0.873** | 306 | 10.67 |

**Example predictions**  We provide in Figure 18 a comparison for an example dataset between the predictive errors (i.e. difference between predicted mean and ground truth) produced by three models: Swin-TNP with PT-GE, ConvCNP, and PT-TNP. The predictions are performed at all station locations. The stations included in the context set are indicated with a black dot. The figures show how Swin-TNP usually produces lower errors in comparison to the baselines, indicated through paler colours. Examples of regions where this is most prominent include central US, as well as southern Australia and southern Europe. We also provide in Figure 19 a zoomed-in version of Figure 18 focusing on the US region.

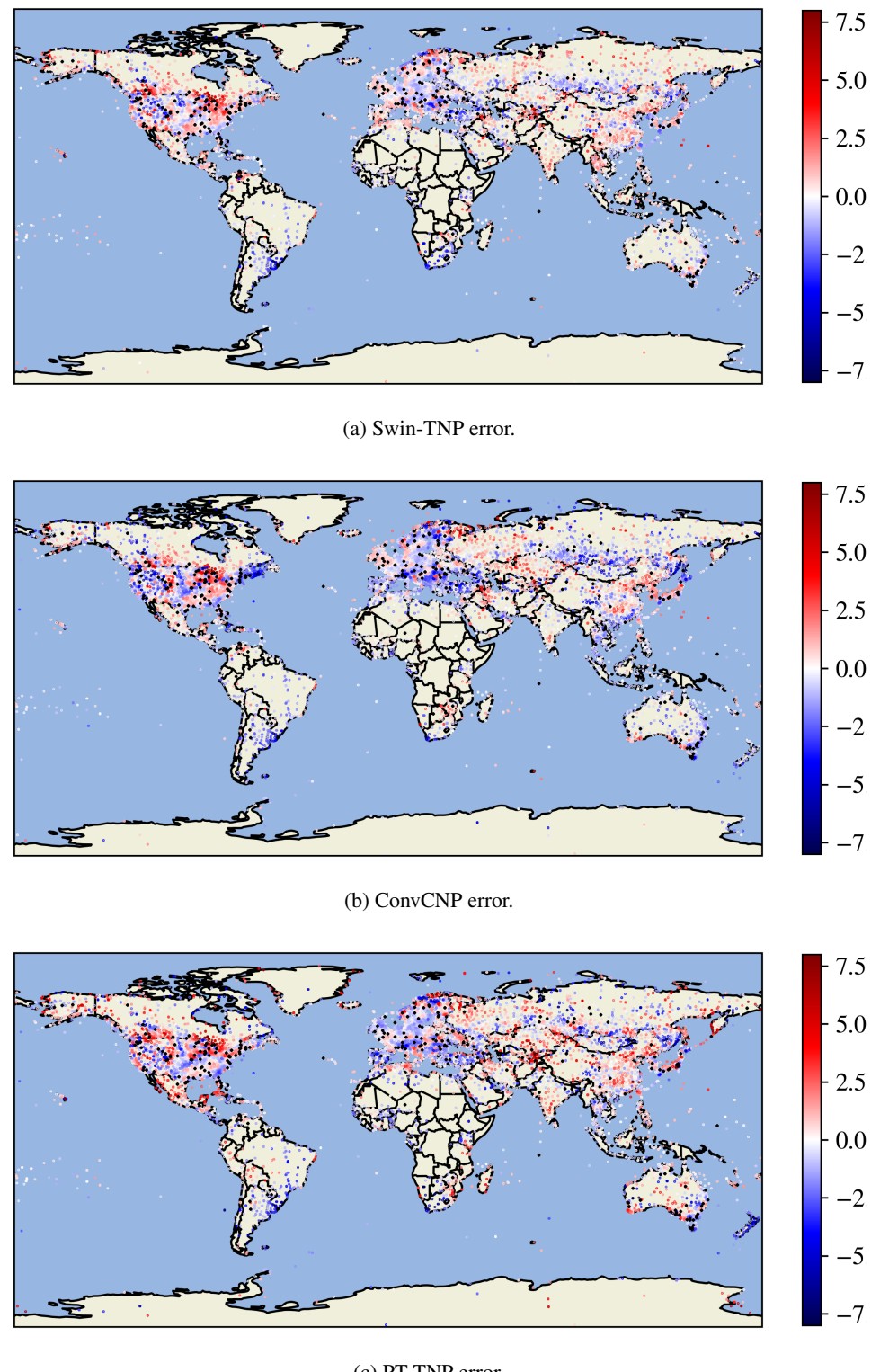

(a) Swin-TNP error.

(b) ConvCNP error.

(c) PT-TNP error.

*Figure 18.* A comparison between the predictive error—the difference between predictive mean and ground truth—of the 2m temperature at all weather station locations at 15:00, 28-01-2019. Stations included in the context dataset are shown as black dots (3% of all station locations). The mean predictive log-likelihoods (averaged across the globe) for these samples are 1.611 (Swin-TNP, PT-GE, grid size of $64 \times 128$), 1.351 (ConvCNP, grid size of $64 \times 128$), and 1.271 (PT-TNP, $M = 256$).

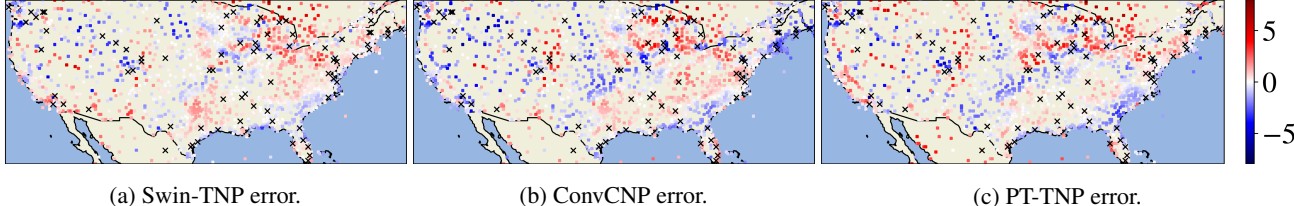

(a) Swin-TNP error.        (b) ConvCNP error.        (c) PT-TNP error.

*Figure 19.* A comparison between the predictive error of the 2m temperature at the US weather station locations at 15:00, 28-01-2019. Stations included in the context dataset are shown as black crosses ($\approx 3\%$ of station locations). The Swin-TNP uses the PT-GE. Both the Swin-TNP and ConvCNP use a grid size of $64 \times 128$. The PT-TNP uses $M = 256$ pseudo-tokens. The mean log-likelihoods of this sample for the three models are $1.611$, $1.351$, and $1.271$, respectively.

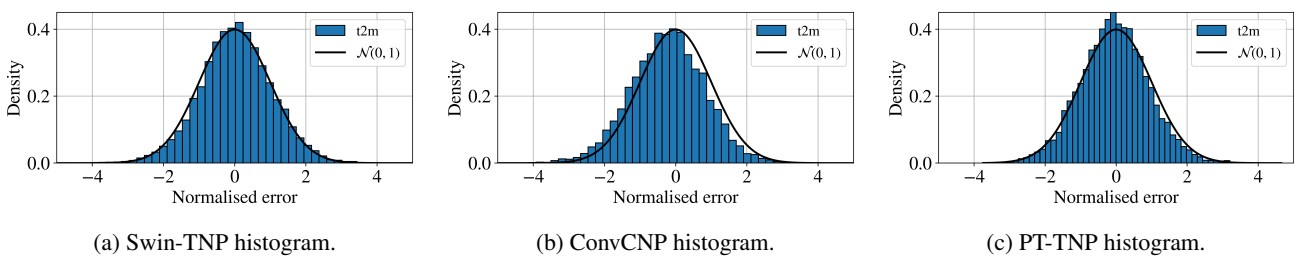

(a) Swin-TNP histogram.        (b) ConvCNP histogram.        (c) PT-TNP histogram.

*Figure 20.* A comparison between the normalised predictive error—the predictive error divided by the predicted standard deviation—of the t2m at the US weather station locations at 15:00, 28-01-2019. The context set contains observations at $3\%$ of station locations. Each plot shows a histogram of the normalised errors based on the predictions at all station locations, alongside an overlaid standard normal distribution that perfect predictive uncertainties should follow. The mean log-likelihoods of the normalised predictive errors under a standard normal distribution for the Swin-TNP (PT-GE, grid size of $64 \times 128$), ConvCNP (grid size $64 \times 128$), and PT-TNP ($M = 256$) are $-1.439$, $-1.490$, and $-1.380$, respectively. For reference, a standard normal distributionn has a negative entropy of $-1.419$, indicating that the Swin-TNP has the most accurate predictive uncertainties.

**Analysis of the predictive uncertainties**    For the example dataset considered above, Figure 20 shows histograms of the normalised predictive errors, defined as the predictive errors divided by the predictive standard deviations. We compute the mean log-likelihoods under a standard normal distribution, and compare it to the reference negative entropy of the standard normal distribution of $-1.419$. This acts as an indicator of the accuracy of the predictive uncertainties produced by the three models we consider: Swin-TNP (PT-GE, grid size $64 \times 128$), ConvCNP (grid size $64 \times 128$), and PT-TNP ($M = 256$). For the dataset considered in Figure 20, we obtain $-1.439$ for Swin-TNP, $-1.490$ for ConvCNP, and $-1.380$ for PT-TNP, indicating that, out of the three models, Swin-TNP outputs the most accurate uncertainties.

**Analysis of grid size influence**    We study to what extent increasing the grid size of the models, and hence their capacity, improves their predictive performance. We repeat the experiment for two models: Swin-TNP (with PT-GE) and ConvCNP with a grid size of $192 \times 384$, corresponding to $0.9375°$ in both latitudinal and longitudinal directions. For the Swin-TNP we use a window size of $8 \times 8$, and a shift size of $4 \times 4$. For the U-Net architecture within the ConvCNP we use a kernel size of nine. In the decoder of the bigger ConvCNP model, attention is performed over the nearest 9 neighbours, whereas for the smaller ConvCNP we use full attention. This makes the FPT of the bigger model smaller that that of the $64 \times 128$ model.

The results are shown in Table 6 (and represent a subset of the results shown in Table 5). In comparison to the Swin-TNP and ConvCNP models which use a grid size of $64 \times 128$, both models improve significantly. However, the bigger ConvCNP still underperforms both the small and big variant of Swin-TNP, with a significant gap in both log-likelihood and RMSE.

**Analysis of influence of nearest-neighbour encoding and decoding**    A final ablation we perform in this experiment studies the influence of the number of nearest neighbours considered for the encoder and decoder on the performance of the model. Initially, we focus on the effect of nearest neighbour decoding, investigating two models—with and without full attention at decoding time. More specifically, we compare Swin-TNP with PT-GE and KI-GE with a grid size of $64 \times 128$, and either perform full attention in the grid decoder, or cross-attention over the 9 nearest neighbours (NN-CA). For the variants with full attention, we evaluate the log-likelihood at $25\%$ randomly sampled station locations instead of all of them because of memory constraints. The results are shown in Table 7, and indicate that, not only does NN-CA offer a more

*Table 6.* Test log-likelihood (↑) and RMSE (↓) for the `t2m` station prediction experiment when varying grid size for two models. The standard errors of the log-likelihood are all below 0.004, and of the RMSE below 0.005. NN̸ signifies that full attention is applied in the decoder. FPT: forward pass time for a batch size of eight in ms. Params: number of model parameters in units of M. Best results for each configuration (Swin-TNP / ConvCNP) are bolded.

| Model | GE | Grid size | Log-lik. ↑ | RMSE ↓ | FPT | Params |
|---|---|---|---|---|---|---|
| ConvCNP NN̸ | SetConv | $64 \times 128$ | 1.535 | 1.252 | 96 | 9.36 |
| ConvCNP | SetConv | $192 \times 384$ | **1.689** | **1.166** | 74 | 9.36 |
| Swin-TNP | PT-GE | $64 \times 128$ | 1.819 | 1.006 | 127 | 2.29 |
| Swin-TNP | PT-GE | $192 \times 384$ | **2.053** | **0.873** | 306 | 10.67 |

scalable decoder attention mechanism, but it also leads to improved predictive performance when applied to spatio-temporal data. We hypothesise this is due to the inductive biases it introduces, which are appropriate for the strong spatio-temporal correlations present in the data we used.

*Table 7.* Test log-likelihood (↑) and RMSE (↓) for the `t2m` station prediction experiment when varying the decoder attention mechanism—nearest-neighbour cross-attention and full attention (NN̸, indicating that NN-CA is not employed). The standard errors of the log-likelihood and RMSE are all below 0.003. FPT: forward pass time for a batch size of eight in ms. Params: number of model parameters in units of M. Best results for each configuration (PT-GE / KI-GE) are bolded.

| Model | GE | Grid size | Log-lik. ↑ | RMSE ↓ | FPT | Params |
|---|---|---|---|---|---|---|
| Swin-TNP | KI-GE | $64 \times 128$ | **1.683** | **1.157** | 121 | 1.14 |
| Swin-TNP (NN̸) | KI-GE | $64 \times 128$ | 1.544 | 1.436 | 137 | 1.14 |
| Swin-TNP | PT-GE | $64 \times 128$ | **1.819** | **1.006** | 127 | 2.29 |
| Swin-TNP (NN̸) | PT-GE | $64 \times 128$ | 1.636 | 1.273 | 144 | 2.29 |

Focusing on just the models using the PT-GE, we also study intermediate regimes for the nearest neighbour decoding mechanism with $k_{\text{dec}} = 25$, and $k_{\text{dec}} = 49$. Moreover, we also investigate how the models perform with an increased number of nearest neighbours considered during encoding ($k_{\text{enc}} = 9$). The full results are presented in Table 8, where for each model we specify the number of nearest neighbours considered in the encoder ($k_{\text{enc}}$) and decoder ($k_{\text{dec}}$).

*Table 8.* Test log-likelihood (↑) and RMSE (↓) for the `t2m` station prediction experiment when varying the number of nearest neighbours considered for the encoder ($k_{\text{enc}}$) and decoder ($k_{\text{dec}}$). NN̸ signifies that full attention is applied in the decoder. The standard errors of the log-likelihood and RMSE are all below 0.003. FPT: forward pass time for a batch size of eight in ms. Params: number of model parameters in units of M. Best results for each configuration (PT-GE / KI-GE) are bolded.

| Model | GE | $k_{\text{enc}}$ | $k_{\text{dec}}$ | Grid size | Log-lik. ↑ | RMSE ↓ | FPT | Params |
|---|---|---|---|---|---|---|---|---|
| Swin-TNP | PT-GE | 1 | 9 | $64 \times 128$ | **1.819** | **1.006** | 127 | 2.29 |
| Swin-TNP (NN̸) | PT-GE | 1 | - | $64 \times 128$ | 1.636 | 1.273 | 144 | 2.29 |
| Swin-TNP | PT-GE | 1 | 25 | $64 \times 128$ | 1.787 | 1.038 | 207 | 2.29 |
| Swin-TNP | PT-GE | 1 | 49 | $64 \times 128$ | 1.810 | 1.028 | 327 | 2.29 |
| Swin-TNP | PT-GE | 9 | 9 | $64 \times 128$ | **1.857** | **0.995** | 264 | 2.29 |

We observe that the performance of the models tends to:

- Be optimal for lower values of $k_{\text{dec}}$ (9)—we believe this is because locality represents a good inductive bias in the task we consider. We also suspect that with sufficient training, the models with different $k_{\text{dec}}$ would eventually reach similar performance, but a lower value encourages more efficient training and has a lower computational cost.
- Slightly improve with increasing $k_{\text{enc}}$—the gridded pseudo-tokens are, on average, modulated by more context points, hence increasing predictive performance. However, this comes at an increased computational cost.

G.2.2. ADDITIONAL RESULTS FOR RICHER CONTEXT SET

In the previous experiment, the model was provided with relatively little context information about the off-the-grid `t2m` variable, forcing it to learn meaningful relationships between the relatively sparse off-the-grid information and rich on-the-grid data (`skt` values). In order to investigate whether the model still manages to learn these relationships between the off-

and on-the-grid data (t2m and skt) and to exploit the on-the-grid information even in the presence of more off-the-grid data[7], we also perform an experiment with richer context sets. More specifically, the number of off-the-grid context points is sampled according to $N_{\text{off},c} \sim \mathcal{U}_{[0.25,0.5]}$. The results are given in Table 9.

For the ConvCNP we evaluated two models—one with full decoder attention and one with nearest-neighbour cross-attention (NN-CA). Similarly to the previous section, we found that the latter has a better performance with a log-likelihood of 1.705 (NN-CA) as opposed to 1.635 (full attention). As such, Table 9 shows the results for the NN-CA ConvCNP model.

*Table 9.* Test log-likelihood (↑) and RMSE (↓) for the t2m station prediction experiment with richer off-the-grid context information. The standard errors of both the log-likelihood and the RMSE are all below 0.003. FPT: forward pass time for a batch size of eight in ms. Params: number of model parameters in units of M.

| Model | GE | Grid size | Log-lik. (↑) | RMSE (↓) | FPT | Params |
|---|---|---|---|---|---|---|
| CNP | - | - | 0.715 | 3.056 | 27 | 0.34 |
| PT-TNP | - | $M = 256$ | 1.593 | 1.403 | 219 | 1.57 |
| ConvCNP | SetConv | $64 \times 128$ | 1.705 | 1.100 | 21 | 9.36 |
| ViTNP | KI-GE | $48 \times 96$ | 1.754 | 1.112 | 175 | 1.14 |
| ViTNP | PT-GE | $48 \times 96$ | 1.988 | 0.932 | 188 | 1.83 |
| ViTNP | KI-GE | $144 \times 288 \rightarrow 48 \times 96$ | 1.842 | 1.046 | 179 | 1.29 |
| ViTNP | PT-GE | $144 \times 288 \rightarrow 48 \times 96$ | 2.242 | 0.798 | 221 | 6.69 |
| Swin-TNP | KI-GE | $128 \times 256$ | 2.362 | 0.758 | 174 | 1.14 |
| Swin-TNP | PT-GE | $128 \times 256$ | **2.446** | **0.697** | 208 | 5.43 |
| Swin-TNP (skt) | PT-GE | $128 \times 256$ | 1.501 | 1.266 | 178 | 5.43 |
| Swin-TNP (t2m) | PT-GE | $128 \times 256$ | 2.331 | 0.909 | 186 | 5.38 |

The results are consistent with the findings from the main experiment:

- The performances of the baselines (CNP, PT-TNP, and ConvCNP) are significantly worse than the gridded TNP variants considered.
- Among the ViT variants, the ones that employ patch encoding before projecting to a $48 \times 96$ grid outperform the ones that directly encode to a $48 \times 96$ grid.
- For each gridded TNP variant, the pseudo-token grid encoder (PT-GE) performs better than the kernel-interpolation one (KI-GE).
- The variants with a Swin-transformer backbone outperform the ones with a ViT-backbone, even when the latter has more parameters and a higher FPT.
- Performing nearest-neighbour cross-attention in the decoder as opposed to full attention leads to both computational speed-ups, as well as enhanced predictive performance.

In comparison to the main experiment, the gap in performance between Swin-TNP and Swin-TNP (t2m) is smaller—this is expected, given that the context already includes between 25% and 50% of the off-the-grid station locations. However, the gap is still significant, implying that Swin-TNP manages to leverage the on-the-grid data (skt) and exploits its relationship with the target t2m to improve its predictive performance.

### G.3. Combining Multiple Sources of Unstructured Wind Speed Observations

In this experiment, we consider modelling the eastward (u) and northward (v) components of wind speed at 700hPa, 850hPa and 1000hPa (surface level). These quantities are essential for understanding and simulating large-scale circulation in the atmosphere, for wind energy integration into power plants, or for private citizens and public administrations for safety planning in the case of hazardous situations (Lagomarsino-Oneto et al., 2023). We obtain each of the six modalities from the ERA5 reanalysis dataset (Hersbach et al., 2020), and construct datasets over a latitude / longitude range of $[25°, 49°]$ / $[-125°, -66°]$, which corresponds to the contiguous US, spanning four hours. We show plots of wind speeds at the three pressure levels for a single time step in Figure 21.

For each task, we first sample a series of four consecutive time points. From this $4 \times 96 \times 236$ grid, we sample a proportion $p_c$ and $p_t$ of total points to form the context and target datasets, where $p_c \sim \mathcal{U}_{[0.05,0.25]}$ and $p_t = 0.25$. The context and

---

[7]This is achieved by comparing the performance of a model that is only given off-the-grid context information, with a similar model that is provided with both off- as well as on-the-grid data.

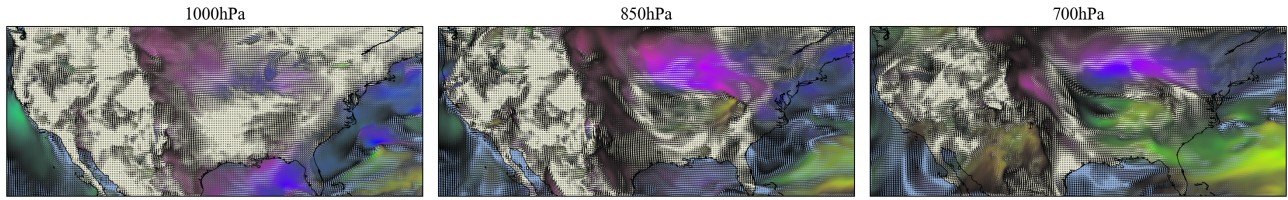

*Figure 21.* Wind-speed and direction at each of the three pressure levels, 700hPa, 850hPa and 1000hPa, over the contiguous US at 15:00 GMT, 08-06-1997. The colours correspond to the magnitude and direction of the wind speed.

|     |     |     |
| --- | --- | --- |
| (a) Swin-TNP error. | (b) ConvCNP error. | (c) PT-TNP error. |

*Figure 22.* A comparison between the predictive error—the difference between predictive mean and ground truth—of normalised wind speeds for a selection of CNP models on a small region of the US at 04:00, 01-01-2019. Each plot consists of 2,400 arrows with length and orientation corresponding to the direction and magnitude of the wind-speed error at the corresponding pressure level. The colour of each arrow is given by the HSV values with hue dictated by orientation, and saturation and value dictated by length (i.e. the brighter the colour, the larger the error). For this dataset, the context dataset consists of $5\%$ of the total available observations, and the corresponding mean predictive log-likelihoods for the Swin-TNP (PT-GE, grid size $4 \times 24 \times 60$), ConvCNP (grid size $4 \times 24 \times 60$) and PT-TNP ($M = 64$) are 5.84, 3.23 and 2.41.

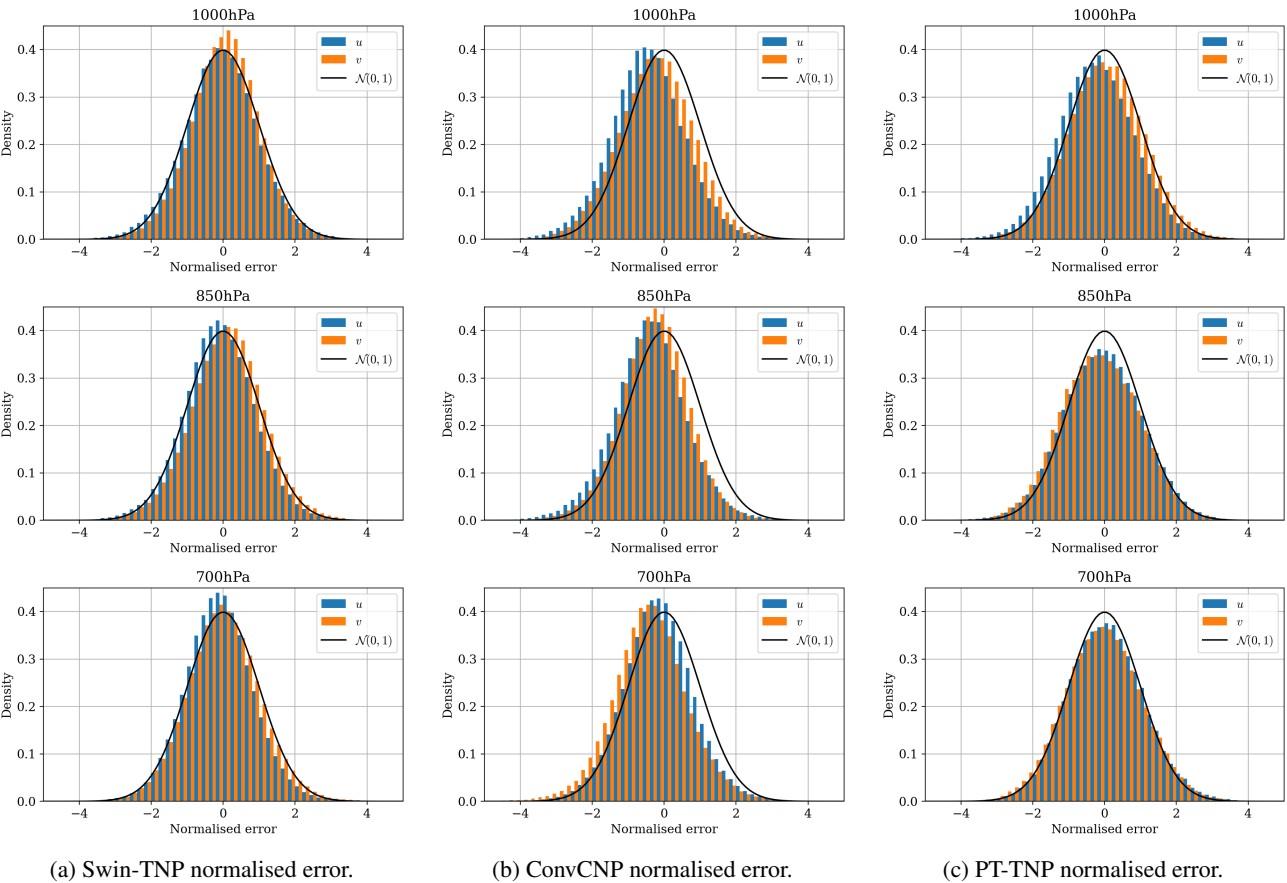

(a) Swin-TNP normalised error.    (b) ConvCNP normalised error.    (c) PT-TNP normalised error.

*Figure 23.* A comparison between the normalised predictive error—the predictive error divided by the predictive standard deviation—of wind speeds for a selection of CNP models on a small region of the US at 04:00, 01-01-2019. Each plot compares a histogram of values for both the $u$ and $v$ components with a standard normal distribution, which perfect predictive uncertainties follow. For this dataset, the context dataset consists of $5\%$ of the total available observations. The mean log-likelihoods (averaged over pressure levels and the 4 time points) of the normalised predictive errors under a standard normal distribution for the Swin-TNP (PT-GE, grid size $4 \times 24 \times 60$), ConvCNP (grid size $4 \times 24 \times 60$) and PT-TNP ($M = 64$) are -1.425, -1.501 and -1.514. For reference, a standard normal distribution has a negative entropy of -1.419. This indicates that the Swin-TNP has the most accurate predictive uncertainties.

target locations are sampled independently for each of the six modalities. All models are trained for $300,000$ iterations on hourly data between $2009 - 2017$ with a batch size of eight. Validation is performed on 2018 and testing on 2019. Experiment specific architecture choices are described below, and a full set of results is provided in Table 10.

**Input embedding**    As the input contains both temporal and latitude / longitude information, we use both Fourier embeddings for time and spherical harmonic embeddings for the latitude / longitude values. These are not used in the ConvCNP as the ConvCNP does not modify the inputs to maintain translation equivariance (in this case, with respect to time and the great-circle distance).

**Grid sizes**    We chose a grid size of $4 \times 24 \times 60$ for the ConvCNP and Swin-TNP models, as this corresponds to a grid spacing of $1°$ in the latitudinal direction and around $1°$ in the longitudinal direction. For the ViTNP, we chose a grid size of $4 \times 12 \times 30$, corresponding to a grid spacing of $2°$.

**CNP**    A different deepset is used for each modality, with the aggregated context token for each modality then summed together to form a single aggregated context token, i.e. $\mathbf{z}_c = \sum_{s=1}^{S} \frac{1}{N_{c,s}} \sum_{n=1}^{N_{c,s}} e_s(\mathbf{x}_{c,n,s}, \mathbf{y}_{c,n,s})$.

**PT-TNP**    For this experiment, we could only use $M = 64$ pseudo-tokens for the PT-TNP without running into out-of-memory issues. This highlights a limitation in scaling PT-TNPs to large datasets. We note that there does exist work that remedies the poor memory scaling of PT-TNPs (Feng et al., 2024); however, this trades off against time complexity which itself is a bottleneck given the size of datasets we consider. A different encoder is used for each modality, before aggregating the tokens into a single context set of tokens.

**ConvCNP**    For the ConvCNP, we use a grid size of $4 \times 24 \times 60$. Each modality is first grid encoded separately using the SetConv, concatenated together and then to $C = 128$ dimensions before passing through the U-Net.

**Swin-TNP**    For the Swin-TNP models, we use a grid size of $4 \times 24 \times 60$, a window size of $4 \times 4 \times 4$ and a shift size of $0 \times 2 \times 2$.

**Swin-TNP ($T$)**    For the Swin-TNP ($T$) models we use analogous architectures to Swin-TNP. For the pseudo-token grid encoder and decoder, we parameterise $\{\rho_h\}_{h=1}^{H}$ by an MLP with two hidden layers of dimension $H$ and output dimension $H$, where $H$ is the number of heads used in the multi-head attention mechanism. For the translation equivariant Swin processor, we add a fixed set of learnable relative positional encodings.

**Swin-TNP ($\widetilde{T}$)**    For the Swin-TNP ($\widetilde{T}$) models we use analogous architectures to Swin-TNP ($T$). To break equivariance, we first perform a Fourier expansion on the context token locations, and pass the result through an MLP with two hidden layers of dimension $D_z$. We use 8 Fourier coefficients, and drop the fixed inputs out during training with a probability of 0.5. We zero them out outside the training region (the US, corresponding to a latitude / longitude range of $[25°, 49°]$ / $[-125°, -66°]$). Although minor, the additional memory overhead from the extra basis functions required us to slightly reduce the proportion of target points during training, validation, and testing to $p_t = 0.235$.

### G.4. Incorporating translation equivariance

We provide the full set of results (including RMSE and parameter count) in Table 11. For the `t2m` station experiment, the standard errors of the log-likelihoods are all below $0.004$ and those of the RMSE below $0.003$. For the wind speed experiment when tested on the US region, the standard errors of the log-likelihoods are below $0.010$ for the Swin-TNP and Swin-TNP ($T$), and below $0.030$ for Swin-TNP ($\widetilde{T}$). The same holds true when tested on Europe, with the exception of the Swin-TNP which has a standard error of $0.065$. The standard errors of the RMSE are below $0.013$.

### G.5. Additional comparisons to ConvCNP

The ConvCNP is a strong baseline: its FPT is fairly low when we incorporate the NN-CA encoding and decoding mechanisms, it incorporates translation equivariance, and its number of parameters does not increase with the grid size. To perform a comprehensive comparison between Swin-TNP and ConvCNP, we provide additional results for larger grid sizes for both real-world experiments. The results are illustrated in Table 12 and show that:

*Table 10.* Test log-likelihood (↑) and RMSE (↓) for the the multi-modal wind speed dataset. All grids have a grid size of 4 in the time dimension. The standard errors of the log-likelihoods are all below 0.02, and of the RMSE below 0.005. NN signifies that full attention is applied in the decoder. FPT: forward pass time for a batch size of eight in ms. Params: number of model parameters in units of M.

| Model | GE | Grid size | Log-lik. ↑ | RMSE ↓ | FPT | Params |
|---|---|---|---|---|---|---|
| CNP | - | - | −1.593 | 2.536 | 33 | 0.66 |
| PT-TNP | - | $M = 64$ | 3.988 | 1.185 | 166 | 1.79 |
| ConvCNP NN | SetConv | $24 \times 60$ | 6.143 | 0.784 | 210 | 14.41 |
| ViTNP | multi KI-GE | $12 \times 30$ | 5.371 | 0.908 | 349 | 1.49 |
| ViTNP | multi PT-GE | $12 \times 30$ | 7.754 | 0.651 | 374 | 2.36 |
| ViTNP | multi KI-GE | $24 \times 60 \rightarrow 12 \times 30$ | 6.906 | 0.718 | 372 | 1.55 |
| ViTNP | multi PT-GE | $24 \times 60 \rightarrow 12 \times 30$ | 7.288 | 0.681 | 392 | 3.25 |
| Swin-TNP | multi KI-GE | $24 \times 60$ | 7.603 | 0.651 | 355 | 1.49 |
| Swin-TNP | multi PT-GE | $24 \times 60$ | **8.603** | **0.577** | 375 | 3.19 |
| Swin-TNP | single KI-GE | $24 \times 60$ | 7.794 | 0.642 | 366 | 1.39 |
| Swin-TNP | single PT-GE | $24 \times 60$ | 8.073 | 0.614 | 364 | 1.67 |
| ConvCNP | SetConv | $48 \times 120$ | 7.841 | 0.615 | 64 | 14.41 |
| Swin-TNP | multi PT-GE | $48 \times 120$ | **9.383** | **0.509** | 369 | 6.51 |

*Table 11.* Test log-likelihood (↑) for Swin-TNP ($T$) and ($\widetilde{T}$), compared to the non-equivariant Swin-TNP. FPT: forward pass time for a batch size of eight in ms. Params: number of model parameters in units of M.

| | t2m stations | | | | Wind speed | | | | | |
|---|---|---|---|---|---|---|---|---|---|---|
| | | | | | **US** | | **Europe** | | | |
| Model | Log-lik. (↑) | RMSE (↓) | FPT | Params | Log-lik. (↑) | RMSE (↓) | Log-lik. (↑) | RMSE (↓) | FPT | Params |
| Swin-TNP ($T$) | 1.895 | 0.981 | 125 | 1.20 | 9.972 | 0.485 | 10.087 | 0.483 | 373 | 2.09 |
| Swin-TNP ($\widetilde{T}$) | **1.926** | **0.965** | 142 | 1.24 | **10.086** | **0.469** | **10.429** | **0.477** | 467 | 2.12 |
| Swin-TNP | 1.819 | 1.006 | 127 | 2.29 | 8.603 | 0.577 | −13.265 | 3.000 | 375 | 3.19 |

- In the station experiment, the ConvCNP with the largest grid size ($384 \times 768$) still lags behind the smaller Swin-TNP ($64 \times 128$), with twice the FPT, and 4x more parameters.

- In the multi-modal wind speed experiment, ConvCNP at the native resolution of the grid ($96 \times 236$) still lags behind the translation equivariant version of Swin-TNP ($T$) with $24 \times 60$ grid size. Efficiency-wise, the ConvCNP has a lower FPT, but it requires 7x more parameters. ConvCNP ($96 \times 236$) also falls behind the bigger Swin-TNP ($48 \times 120$), and, although has a lower FPT, it requires 2x more parameters.

*Table 12.* Test log-likelihood (↑) for the t2m station prediction (left) / multi-modal wind speed (right) experiments. The standard errors of the log-likelihood are all below 0.010/0.030. For the wind speed experiment, m- stands for multi GE. FPT: forward pass time for a batch size of eight in ms. Params: number of model parameters in M.

**(a)** t2m station prediction experiment.

| Model | GE | Grid size | TE | Log-lik. ↑ | FPT | Params |
|---|---|---|---|---|---|---|
| ConvCNP | SetConv | $192 \times 384$ | ✓ | 1.689 | 74 | 9.36 |
| ConvCNP | SetConv | $288 \times 576$ | ✓ | 1.784 | 149 | 9.36 |
| ConvCNP | SetConv | $384 \times 768$ | ✓ | 1.808 | 259 | 9.36 |
| Swin-TNP | PT-GE | $64 \times 128$ | × | 1.819 | 127 | 2.29 |
| Swin-TNP ($T$) | PT-GE | $64 \times 128$ | ✓ | 1.895 | 125 | 1.20 |
| Swin-TNP | PT-GE | $192 \times 384$ | × | **2.053** | 306 | 10.67 |

**(b)** Multi-modal wind speed experiment.

| Model | GE | Grid size | TE | Log-lik. ↑ | FPT | Params |
|---|---|---|---|---|---|---|
| ConvCNP | SetConv | $48 \times 120$ | ✓ | 7.841 | 64 | 14.41 |
| ConvCNP | SetConv | $72 \times 180$ | ✓ | 8.478 | 100 | 14.41 |
| ConvCNP | SetConv | $96 \times 236$ | ✓ | 9.253 | 142 | 14.41 |
| Swin-TNP | m-PT-GE | $24 \times 60$ | × | 8.603 | 375 | 3.19 |
| Swin-TNP ($T$) | m-PT-GE | $24 \times 60$ | ✓ | **9.972** | 373 | 2.09 |
| Swin-TNP | m-PT-GE | $48 \times 120$ | × | 9.383 | 369 | 6.51 |

## G.6. The EAGLE dataset

To show that our approach can be used as a general-purpose tool for spatio-temporal state estimation, we additionally apply it to a large-scale fluid dynamics dataset called the EAGLE (Janny et al., 2023). For a thorough description of the data, we refer the reader to Janny et al. (2023). The dataset is composed of $\sim 1.1$ million simulations on an irregular 2D

mesh, modelling unsteady fluid dynamics caused by a moving flow source interacting with nonlinear scene structure. More specifically, EAGLE models the airflow generated by a 2D drone moving within a 2D scene with varying floor profile. The dataset comprises of three types of floor geometries (step, triangular, and spline), and for each geometry it contains roughly 200 different scenes, giving a total of approximately $1,200$ simulations (two simulations per geometry depending on whether the drone goes to the left or right of the scene). Each simulation contains 990 time steps. The simulated quantities are the velocity field and the pressure field over the entire domain, sampled on an irregular 2D mesh.

The task we address with Swin-TNP is spatial interpolation, modelling the pressure and velocity at randomly sampled points in space at a randomly sampled time point. The inputs to the Swin-TNP are 1) the location of the drone, 2) the locations of the domain boundaries (open or closed), as well as 3) a proportion $p_c \sim \mathcal{U}_{[0.05, 0.25]}$ of randomly sampled context points containing the velocity and pressure values at the context locations. Swin-TNP predicts the velocity and pressure at the remaining target locations. To indicate the different types of inputs (i.e. node associated with the drone / with a boundary / within the fluid), we apply a similar strategy as the one for handling multi-modal data, whereby we use input-specific point-wise encoders.

We train the models for $1,000,000$ iterations with a batch size of eight. Testing is performed on $80,000$ samples, where a sample is defined as a randomly-chosen time step from one of the trajectories included in the test set. We report the results in Table 13 for two different grid sizes—$32 \times 12$ and $60 \times 36$—and using both the PT-GE and the KI-GE. All models use NN-CA in the encoder and decoder with $k_{\text{enc}} = 1$ and $k_{\text{dec}} = 9$, a window size of $4 \times 4$ and a shift size of $2 \times 2$. Moreover, we provide some example predictions of the velocity field in Figure 24. These results show that our approach can be easily applied to challenging spatio-temporal datasets, and that, consistent with the findings from the other experiments, our proposed pseudo-token grid encoder outperforms the kernel-interpolation grid encoder. Future work includes extending the analysis to spatio-temporal forecasting, and comparing to established approaches that are able to deal with irregular meshes.

*Table 13.* Test log-likelihood ($\uparrow$) for the Eagle experiment. The results are based on $80,000$ samples, where a sample is one randomly-chosen time step from one of the trajectories included in the test set (which are also randomly sampled). FPT: forward pass time for a batch size of eight in ms. Params: number of model parameters in units of M. Best results are bolded.

| Model | GE | Grid size | Log-lik. $\uparrow$ | Log-lik std. | FPT | Params |
|---|---|---|---|---|---|---|
| Swin-TNP | PT-GE | $32 \times 12$ | 8.153 | 0.018 | 41 | 1.16 |
| Swin-TNP | KI-GE | $32 \times 12$ | 7.748 | 0.018 | 40 | 1.41 |
| Swin-TNP | PT-GE | $60 \times 36$ | **8.319** | 0.018 | 40 | 1.79 |
| Swin-TNP | KI-GE | $60 \times 36$ | 8.059 | 0.017 | 40 | 1.41 |

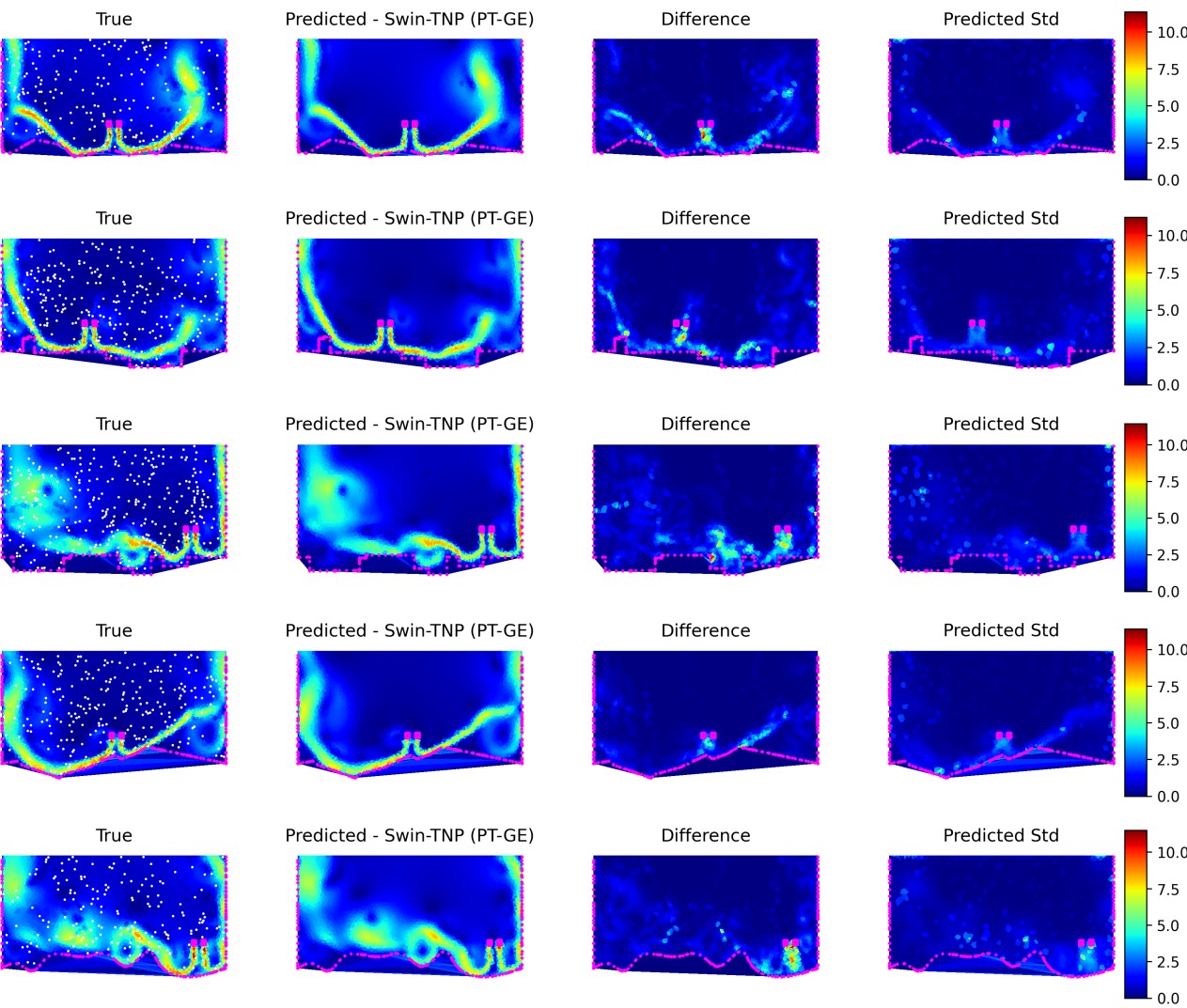

*Figure 24.* Five example predictions for the EAGLE dataset. The first column shows the true PDE state, and the coloured dots (white and magenta) represent the context set that is fed to the Swin-TNP. The magenta points denote boundaries, while the white dots are points within the fluid mesh. The second column shows the predictions of the Swin-TNP with PT-GE ($32 \times 12$) in terms of predicted velocity (i.e. the model predicts the horizontal and vertical velocities and pressures, and we are plotting $v = \sqrt{v_x^2 + v_y^2}$). The third column shows the difference between the true and predicted velocities, and the last column shows the predicted standard deviation of the velocity $v$.

