# OpenReview forum: "Gridded Transformer Neural Processes for Spatio-Temporal Data"
_ICML.cc/2025/Conference — ICML 2025 spotlightposter_

### Official Review · Reviewer_tnKQ · 2025-03-12

**Overall Recommendation:** 4

**Summary:**

The authors present an approach for spatio-temporal modelling. This involves encoding point observations into grid based “pseudo-tokens” using an attention mechanism and processing these pseudo tokens using a Transformer Neural Process approach. A grid decoder also using an attention mechanism allows prediction at arbitrary spatio-temporal locations.

The authors evaluate this approach on synthetic data and weather prediction tasks and find that it performs favourably compared to baseline approaches. Ablations of their approach are performed allowing us to see the impact of various elements of it, and many details are provided in the extensive appendices.

The author’s approach combines recent advancements in order to scale effectively to allow spatio-temporal modelling from large datasets. This approach can handle both unstructured “off the grid” and structured “on the grid” data, and can combine multiple modalities of data.

**Claims And Evidence:**

Claims made are supported by clear and convincing evidence.

**Essential References Not Discussed:**

All relevant references seem to be included.

**Experimental Designs Or Analyses:**

Both the synthetic data and weather predicting tasks seem sound. Perhaps the weather predicting tasks could be expanded as this seems to be the focus of the paper from the introduction section.

**Methods And Evaluation Criteria:**

Yes, the synthetic dataset and the weather prediction dataset make sense. The evaluation criteria seem sensible.

**Other Comments Or Suggestions:**

I believe a crossed out NN should probably be included in row 3 of the multimodal results of Table 1 to indicate that the kernel here is not using the k nearest neighbours approach.

**Other Strengths And Weaknesses:**

The paper is quite well written and provided many helpful background details. The investigation into their approach / ablations is extensive. The approach seems more generally applicable than many of the spatio-temporal forecasting papers that are cited.

The baseline grid encoder approach and other TNP approaches seems to be a strong baselines, and it appears that thought has been put into optimizing hyperparameters etc of these baselines.

Perhaps a comparison to a weather predicting approach from the literature would be useful given that the introduction seems to focus on this area quite heavily, however I can understand this is difficult to do. For example some modification of the Vaughan et al. (2024) “Aardvark Weather: End-to-end data-driven weather prediction” approach.

I wonder if the originality compared to Aardvark weather is significant enough. The differences mostly seem to relate to using a local attention based mechanism instead of a "kernel interpolation" approach.

**Questions For Authors:**

1. Looking at Table 6, given that:
The number of parameters of the ConvCNP approach remains fixed as you change the grid size; The FPT for the fine grid size used by this approach is still much lower than for your approach (and might not change too much if you are using the knn KI); Performance for the ConvCNP approach seems to improve with increasing number of grid points,
do you think that using a finer grid could lead to convCNP outperforming your approach while still being competitive on parameters and computation time? Are there practical reasons why this is not feasible?

A response could allow me to evaluate the effectiveness of the proposed method compared to existing approaches.

**Relation To Broader Scientific Literature:**

The approach used does seem quite similar to that developed by Vaughan et al. (2024) “Aardvark Weather: End-to-end data-driven weather prediction”. In their work, a similar grid encoder and decoder is used that can also deal with off the grid and multimodal data, and a Swin transformer is used to process these grid tokens, and so should be able to efficiently deal with large numbers of grid tokens.

Aardvark weather does seem to include many of the key elements of this work such as enabling TNPs to use efficient attention mechanisms, and the use of gridded “pseudo tokens” which aggregate information from the local area and are used as inputs to the transformer component, and similarly a decoder which aggregates information from local pseudo tokens and allows predictions at arbitrary locations.

A difference is that this work uses an attention mechanism to aggregate this local information rather than a kernel interpolation approach. This work also seems to add “translational equivariance” and “approximate translational equivariance”, which is an incorporation of previous work seen in Ashman et al. “Approximately Equivariant Neural Processes” (NeurIPS 2024) and Ashamn et al. “Translation Equivariant Transformer Neural Processes” (ICML 2024).

The authors have provided significant details of their approach and experiments investigating the impact of different components of the approach, which will be helpful to the broader scientific community.

**Theoretical Claims:**

No proofs or theoretical claims.

---

> ### Author Rebuttal · Authors · 2025-03-31
>
> Thank you for your review. We address your concerns below.
>
> **Similarity to Aardvark** - While our method shares similarities with Aardvark, we highlight key differences:
> - **Methodological advancement, rather than weather-specific**: Aardwark is specifically designed for weather modelling, while our method provides a general framework for spatio-temporal tasks. For example, during the rebuttal we conducted experiments on a different spatio-temporal task - PDE modelling with the Eagle dataset [1] - to show the generality of our approach. Example predictions can be seen [here](https://anonymous.4open.science/r/ICML_2025-06DE/eagle_figures.png) and a comparison between the PT-GE and KI-GE for this dataset [here](https://anonymous.4open.science/r/ICML_2025-06DE/eagle_table.png).
> - **Grid Encoding**: Aardvark uses a kernel-interpolation grid encoder (KI-GE), while we propose a pseudo-token grid encoder (PT-GE), which outperforms KI-GE in our experiments.
> - **Transformer Choice**: Aardvark uses a Vision Transformer (ViT) in the encoder, while we use a Swin Transformer, proven to perform better in high-resolution and dense prediction tasks. This is also supported by our experiments.
> - **Training Objective**: Aardvark optimises a latitude-weighted RMSE, while we optimise the predictive log-likelihood, enabling probabilistic modelling, crucial for chaotic systems.
> - **Decoder Design**: Aardvark uses a lightweight convolutional decoder for station forecasts, while we introduce a k-nearest-neighbour attention-based grid decoder.
>
> Both approaches handle observations at arbitrary locations, but our contributions/modifications (PT-GE, replacing ViT with Swin, and k-nearest-neighbour decoder) improve upon Aardvark’s approach, and, unlike Aardvark, our method supports a probabilistic treatment.
>
> Moreover, as the reviewer noted, our method allows for efficient incorporation of (strict and approximate) **translation equivariance**, further enhancing performance and generalisability.
>
> **Other Strengths and Weaknesses**
>
> Aardvark is a modular system for end-to-end weather forecasting, while our work focuses on fundamental research to identify the best approach for spatio-temporal tasks. Rather than directly comparing to Aardvark, which was trained at a much larger scale and in a modular fashion with a focus on forecasting, we propose architectural improvements that could enhance its performance---replacing Aardvark’s encoder (ViT with KI-GE) with our Swin transformer with PT-GE, and its decoder with our k-nearest-neighbour attention-based grid decoder.
>
> "comparison to a weather predicting approach" - Please see response to reviewer t5r6 regarding comparisons to other weather models (GraphCast, FourCastNet, Pangu-Weather). Moreover, please note that we compare to the fundamental methods (ViT + KI-GE) used in Aardvark in all experiments and to the grid encoder used in FuXi-DA (Avg-GE) in the GP experiment.
>
> **Other Comments or Suggestions**
>
> Thank you, we will correct this in a revised version.
>
> **Questions**
>
> The FPT for the ConvCNP approach is indeed lower, but this is due to the fact that we incorporated the k-nearest-neighbours into the kernel-interpolation method (in both the encoder and decoder). If we wanted to run the ConvCNP with full attention in the decoder for the $192 \times 384$ (station experiment) or $48 \times 120$ (multimodal experiment) grids, we would require more than 80 GB of memory, which is more than what we used for the TNP experiments. Moreover, while FPT is an important efficiency metric, ConvCNP requires significantly more parameters than Swin-TNP (up to 7x).
>
> To address your question, we ran a set of [additional experiments](https://anonymous.4open.science/r/ICML_2025-06DE/swintnp_convcnp_comparison.png) with larger grid sizes for the ConvCNP.
> - In the station experiment, the ConvCNP with the largest grid ($384 \times 768$) still lags behind the smaller Swin-TNP ($64 \times 128$), with twice the FPT, and $4$x more parameters.
> - In the wind experiment, ConvCNP at the native resolution ($96 \times 236$) still lags behind the TE Swin-TNP with $24\times 60$ grid size. Efficiency-wise the ConvCNP has a lower FPT, but it requires $7$ times more parameters. ConvCNP ($96 \times 236$) also falls behind Swin-TNP ($48 \times 120$), and although has a lower FPT, it requires $2$x more parameters.
>
> We hope these additional results help you better compare our approach to the strong ConvCNP benchmark, showing that the latter either lags behind in terms of performance and efficiency (as in the t2m experiment), or struggles to match the performance of our strongest TE models even when we increase the grid size to the native data resolution (as in the wind experiment).
>
> [1] Janny, S., B'eneteau, A., Thome, N., Wolf, M.N., Digne, J., & Wolf, C. (2023). Eagle: Large-Scale Learning of Turbulent Fluid Dynamics with Mesh Transformers.

---

> > ### Comment · Reviewer_tnKQ · 2025-04-07
> >
> > Thank you very much for your thoughtful response.
> >
> > The new experiment and discussion on larger grid sized ConvCNP is very interesting. Thank you for this.
> >
> > I feel the authors prime the reader to expect comparisons to weather models due to those approaches being the focus of the start of the introduction, but I understand that you are focusing on a more general approach. Perhaps some changes to this section could minimize the number of questions about this.
> >
> > Your response has increased my opinion of the paper, but I still lean slightly more towards a 3 than a 4.

---

> > > ### Author Response · Authors · 2025-04-07
> > >
> > > Thank you for your thoughtful reply and for suggesting the experiment with increased grid sizes for the ConvCNP—we agree that this would be a valuable addition to the paper.
> > >
> > > It is certainly true that the narrative of the paper mostly focuses on the weather setting and this can somehow obscure the broader applicability of our approach to general spatiotemporal problems. Our main motivation for this focus was to provide a **clear motivating real-world use case**---one where existing architectures often struggle due to the complex nature of the data (i.e. unstructured / off-the-grid, multi-modal but not observed at the same locations, etc.), whereas our model can be successfully applied. Given computational (and space) constraints, we opted to conduct a thorough empirical study in this domain (by (1) combining structured and unstructured data, and (2) combining multi-modal data), rather than spreading our efforts across multiple settings potentially at the cost of experimental depth and conclusiveness.
> > >
> > > That said, we fully appreciate the concern that this emphasis may make the method appear more domain-specific than it actually is. To mitigate this, we had already noted in the introduction that: "This presents fertile ground for impactful research, with findings easily transferable to other spatiotemporal domains. For instance, the same techniques could be used to model real-life systems governed by partial differential equations (PDEs)....". During the rebuttal phase, we took this claim one step further and applied our technique to a PDE dataset, with qualitative results shown in [here](https://anonymous.4open.science/r/ICML_2025-06DE/eagle_figures.png) and a quantitative ones shown in [here](https://anonymous.4open.science/r/ICML_2025-06DE/eagle_table.png).
> > >
> > > In a revised version of the manuscript, we would be happy to incorporate these results, and reframe the introduction to emphasise the general-purpose nature of our method. We can start with this broader perspective, positioning our approach as a flexible framework for spatiotemporal modelling, and then present weather modelling as one motivating example of a domain that benefits from our model's flexibility. Following that, we can include the PDE results as a second example that demonstrates its broader relevance and effectiveness.
> > >
> > > We again thank the reviewer for their constructive suggestions. We believe they will help strengthen the framing of our contributions and better support our claims empirically.

---

### Official Review · Reviewer_t5r6 · 2025-03-19

**Overall Recommendation:** 4

**Summary:**

This paper introduces Gridded Transformer Neural Processes (TNPs), a new framework for modeling large-scale spatio-temporal data. Existing approaches, such as Conditional Neural Processes (CNPs), Convolutional CNPs (ConvCNPs), and Transformer Neural Processes (TNPs), struggle with either scalability or handling unstructured data efficiently. The authors propose a novel solution that incorporates pseudo-token grids and efficient attention mechanisms to improve performance on structured and unstructured datasets.

Main algorithmic and conceptual contributions:
---

1. Gridded Pseudo-Token TNPs:
- Introduces pseudo-tokens on a structured grid to efficiently process unstructured spatio-temporal data.
- Uses cross-attention to encode data into a grid before applying transformers.

2. Efficient Attention Mechanisms for Scalability
- Uses Vision Transformers (ViT) and Swin Transformers to efficiently model spatio-temporal dependencies.
- Nearest-neighbor cross-attention further reduces computational complexity.

3. Handling Multi-Modal Data
- Compares single pseudo-token grid encoding (s-PT-GE) vs. multi pseudo-token grid encoding (m-PT-GE) for handling datasets with multiple sources (e.g., satellite data, ground observations).

4. Incorporating Translation Equivariance
- Introduces translation equivariant TNPs (TE-TNPs) to enforce shift-invariance in spatial patterns.
- Proposes approximate TE-TNPs, which improve generalization while allowing minor symmetry-breaking adaptations.

Main results and findings:
---
1. Synthetic Gaussian Process Regression
- Gridded TNPs outperform ConvCNPs and PT-TNPs, especially on high-resolution data.
- Pseudo-token grid encoding (PT-GE) is superior to kernel interpolation (KI-GE).

2. Real-World weather forecasting
- Temperature Prediction at Weather Stations: Gridded TNPs outperform ConvCNPs by better fusing gridded and unstructured data.
- Multi-Modal Wind Speed Forecasting: Multi pseudo-token encoding (m-PT-GE) improves accuracy in datasets with multiple data sources.
- Translation Equivariance: Equivariant Swin-TNP (T) generalizes better, especially when tested on unseen geographic regions.

**Claims And Evidence:**

The claims made in the paper are generally well-supported by experimental results and conceptual arguments. The authors provide both synthetic and real-world experiments to validate their contributions, and the benchmarks against strong baselines (ConvCNP, PT-TNP, etc.) help substantiate their findings.

Claims that need stronger justification:
---
1. Gridded TNPs generalize better to unseen spatial regions (translation equivariance claim).
The paper shows some evidence that equivariant models generalize better, but:
- Additional experiments on out-of-distribution (OOD) regions could strengthen this claim.
- More direct comparisons with standard TNPs in OOD settings would be useful.

2. Approximate translation equivariance improves generalization while allowing flexibility.
The paper claims that relaxing strict TE improves performance, but:
- The comparison between strict TE-TNP (T) and approximate TE-TNP (T̃) is not fully explored beyond weather data.
- More systematic analysis of when and why approximate TE works better would clarify this finding.

3. Nearest-neighbor cross-attention (NN-CA) improves both computational efficiency and accuracy.
The experiments support the efficiency claim but do not fully isolate the accuracy benefit of NN-CA.
- A direct ablation study (comparing full cross-attention vs. NN-CA) on a fixed dataset would clarify whether NN-CA improves accuracy beyond just efficiency.

**Essential References Not Discussed:**

-

**Experimental Designs Or Analyses:**

The experimental design and analysis in the paper are generally well-structured and appropriate for evaluating the proposed Gridded Transformer Neural Processes (TNPs). The authors conduct both synthetic and real-world experiments, use strong baselines, and report multiple performance metrics. However, there are areas where further validation or alternative setups could strengthen the findings.
For example, lack of comparison with state-of-the-art Transformer forecasting models GraphCast (Google DeepMind), FourCastNet (NVIDIA), Pangu-Weather (Huawei).

**Methods And Evaluation Criteria:**

The proposed methods and benchmarks are generally well-chosen for large-scale spatio-temporal modeling.
ERA5 and Gaussian Process regression are strong datasets for testing model performance.

**Other Comments Or Suggestions:**

-

**Other Strengths And Weaknesses:**

Strengths:
---
- Scalability: Efficient pseudo-token encoding and nearest-neighbor cross-attention (NN-CA) improve computational efficiency.
- Multi-Modal Data Handling: Supports both structured (gridded) and unstructured (sensor-based) data.
- Translation Equivariance: Introduces efficient TE-TNPs and approximate TE (T̃-TNPs) for better generalization.
- Comprehensive Ablations: Tests different encoders, transformers, and attention mechanisms.

Weaknesses:
---
- Limited Comparisons: No direct benchmarking against GraphCast, FourCastNet, or Pangu-Weather.
- Hyperparameter Sensitivity: No detailed study on pseudo-token count (M) and grid resolution effects.
- Narrow OOD Testing: Generalization is only tested in weather forecasting, not other spatio-temporal domains.

**Questions For Authors:**

Comparison to State-of-the-Art Weather Forecasting Models:
---
- Why does the paper not compare against transformer-based forecasting models like GraphCast, FourCastNet, or Pangu-Weather?
- If Gridded TNPs were tested against these models, how would their performance compare in terms of accuracy and efficiency?
If the proposed method is competitive with state-of-the-art models, it strengthens its contribution to weather forecasting. If not, it may suggest limitations in its practical applicability.

Pseudo-Token Grid Hyperparameters:
---
- How does the number of pseudo-tokens (M) and grid resolution affect performance?
- Have experiments been conducted to determine optimal scaling laws for different dataset sizes?

**Relation To Broader Scientific Literature:**

The paper extends Neural Processes (CNPs, TNPs) by incorporating transformer-based dependencies for spatio-temporal modeling. It builds on Vision Transformers (ViT, Swin) and adapts them using pseudo-token grid encoding, similar to point cloud transformers. Compared to Graph Neural Networks (GNNs) used in weather models (GraphCast, Aurora), Gridded TNPs focus on scalability and parallelization.

For multi-modal learning, it improves on kernel interpolation (Aardvark Weather, FuXi-DA) by dynamically encoding structured and unstructured data. The translation-equivariant model (TE-TNP) follows Ashman et al. (2024a, 2024b) but improves efficiency using precomputed relative positional encodings. Finally, the work contributes to weather forecasting, aligning with GraphCast, FourCastNet, and Pangu-Weather, but offering a more flexible structured + unstructured data approach.

**Theoretical Claims:**

Most derivations are reasonable and align with prior literature.

---

> ### Author Rebuttal · Authors · 2025-03-31
>
> We thank the reviewer for their detailed review and address their concerns below.
>
> **Claims and evidence**
>
> **Out-of-Distribution (OOD) and Translation Equivariance (TE)** - We perform TE experiments in the station and wind speed experiments. In the former, where we use stations from all regions globally, there are no OOD regions to test on. However, we explicitly study OOD in the multi-modal wind experiment by training on the US and evaluating on both the US (in-distribution, ID) and Europe (OOD). Table 2 shows that plain Swin-TNP fails OOD on Europe (low test log-likelihood), while the strictly TE ($T$) and approximately TE ($\widetilde{T}$) variants generalise better due to appropriate inductive biases.
>
> - “Additional experiments on OOD regions”-This is addressed by the wind experiment, but we welcome suggestions for additional experiments.
> - “More direct comparisons with standard TNPs OOD”-Table 2 provides this comparison, showing that plain Swin-TNP fails on OOD Europe.
>
> **Comparison between strict and approximate TE-TNP** - This comparison is outside this paper's scope, as it is extensively studied in Ashman et al. [1]. In general, approximate TE is helpful when symmetries are present but some symmetry-breaking features exist (e.g. real-world data).
>
> **Benefits of NN-CA** - As noted in our response to reviewer wqS4, we performed a direct ablation study comparing full attention with NN-CA for the Swin-TNP (with PT-GE and KI-GE) in 1) the t2m station experiment (Table 7) and 2) in the synthetic GP experiment (Table 3).
>
> **Weaknesses and Questions**
>
> **Limited Comparisons with GraphCast, FourCastNet, and Pangu-Weather** - Our work focuses on developing a flexible, general, and efficient approach to spatio-temporal modelling rather than an application-specific method. We compare to fundamental techniques from weather modelling that can handle unstructured data: ViT + KI-GE (Aardvark) and Avg-GE (FuXi-DA). For GraphCast, which relies on GNNs, we discuss connections in the Appendix. FourCastNet and Pangu-Weather assume gridded states, making them incompatible with our setup, where models 1) do not require the full state, 2) can handle off-the-grid observations, and 3) can combine modalities that are not observed at the same locations.
>
> That said, a future direction is adapting our method to forecasting conditioned on past observations at arbitrary locations, enabling comparisons. However, models like GraphCast required ~4 weeks on 32 TPUs, whereas ours were trained for at most a few days on a single GPU, making a direct comparison unfair at this stage.
>
> **“Hyperparameter Sensitivity: Pseudo-token Count ($M$) and Grid Resolution”** - In the synthetic GP experiment (Table 3) we vary the number of pseudo-tokens in PT-TNP ($M=128$, $M=256$) and test with different grid sizes. We also analyse this in the t2m experiment for Swin-TNP with PT-GE and ConvCNP (Table 6). In the real-world experiments, PT-TNP used the maximum $M$ allowed by memory constraints, and decreasing it would just increase the performance gap to our models.
>
> **Narrow OOD Testing** - We conducted additional preliminary experiments on the PDE-based Eagle dataset [2]. Due to space constraints, we cannot provide a detailed description of the experiment, but we provide example predictions of the fluid velocity and a table comparing Swin-TNP (PT-GE) with Swin-TNP (KI-GE) for two grid sizes [here](https://anonymous.4open.science/r/ICML_2025-06DE). These preliminary results show how our method can easily be applied to any spatio-temporal task, and that PT-GE outperforms KI-GE in this setting too.
>
> Regarding generalisation of TE models—In [3] the authors prove that if a translation equivariant prediction map is equipped with a receptive field (i.e. if the output stochastic processes have only local dependencies), then the prediction map generalises spatially (Theorem 2.2). Thus, TE models should generalise in any domain where translation equivariance is an appropriate inductive bias.
>
> **Scaling Laws**
>
> Due to computational constraints, we did not perform experiments on varying dataset sizes, but we acknowledge that exploring data efficiency and scaling laws is valuable, building on existing studies like [4].
>
> [1] Ashman, M., Diaconu, C., Weller, A., Bruinsma, W.P., & Turner, R.E. (2024). Approximately Equivariant Neural Processes.
>
> [2] Janny, S., B'eneteau, A., Thome, N., Wolf, M.N., Digne, J., & Wolf, C. (2023). Eagle: Large-Scale Learning of Turbulent Fluid Dynamics with Mesh Transformers.
>
> [3] ​​Ashman, M., Diaconu, C., Kim, J., Sivaraya, L., Markou, S., Requeima, J., Bruinsma, W.P., & Turner, R.E. (2024). Translation Equivariant Transformer Neural Processes.
>
> [4] Bodnar, C., Bruinsma, W.P., Lucic, A., Stanley, M., Vaughan, A., Brandstetter, J., Garvan, P., Riechert, M., Weyn, J.A., Dong, H., Gupta, J.K., Thambiratnam, K., Archibald, A.T., Wu, C., Heider, E., Welling, M., Turner, R.E., & Perdikaris, P. (2024). A Foundation Model for the Earth System.

---

### Official Review · Reviewer_wqS4 · 2025-03-20

**Overall Recommendation:** 3

**Summary:**

The work introduces a novel approach for modeling large-scale spatio-temporal data without being limited to fixed-resolution grids. Instead of traditional methods that constrain inputs, the authors build on transformer neural processes (TNPs) by developing gridded pseudo-token TNPs. These models use specialized encoders and decoders to manage unstructured, heterogeneous data and employ an efficient attention mechanism based on gridded pseudo-tokens. Additionally, they propose an equivariant variant that leverages translation equivariance as an inductive bias, boosting both accuracy and training efficiency.

**Claims And Evidence:**

1. **Novel Attention-Based Grid Encoder**
   This claim is supported by clear empirical evidence. The authors compare the pseudo-token grid encoder (PT-GE) with a kernel-interpolation grid encoder (KI-GE) on both synthetic and real-world tasks. Experimental results (e.g., Figure 3 and Table 1) show that PT-GE consistently yields higher predictive log-likelihoods.

2. **Enabling Efficient Attention via the Pseudo-Token Grid Encoder**
   This claim is supported by clear empirical evidence. By integrating architectures inspired by the ViT and the Swin Transformer, the resulting TNP variants (ViTNP and Swin-TNP) achieve competitive performance with lower computational costs compared to baselines. Experiments reporting prediction quality and forward pass times support this design choice.

3. **Efficient k-Nearest-Neighbour Attention-Based Grid Decoder**
   While the method reduces computational complexity, the claim that it “outperforms full attention” in terms of predictive performance is not consistently supported, making this claim problematic. The authors propose a grid decoder using k-nearest-neighbour cross-attention and claim that it outperforms full cross-attention. However, in the t2m station prediction experiment (Table 1), the variant using nearest-neighbour attention (NN-CA) attains a lower log-likelihood (1.636) compared to the full cross-attention variant (1.819).


4. **Efficient Incorporation of Translation Equivariance**
   There is clear and convincing evidence supporting this claim. The paper adapts methods from Ashman et al. to incorporate translation equivariance, and experiments (see Table 2) show improved log-likelihoods in both t2m and wind speed tasks, as well as enhanced spatial generalisation.

5. **Empirical Evaluation on Large-Scale and Multi-Source Data**
   There is clear and convincing evidence supporting this claim. Extensive experiments on synthetic Gaussian process regression and real-world weather datasets compare gridded TNPs with multiple baselines (including ConvCNP and standard TNPs). Results demonstrate improved predictive performance and computational efficiency, confirming the model’s scalability and its ability to handle unstructured, multi-source data.

**Essential References Not Discussed:**

N/A

**Experimental Designs Or Analyses:**

The experimental designs and analyses are generally sound and well-aligned with the goals of large-scale spatio-temporal modeling.

For the Gaussian Process Regression, The synthetic experiments are set up using Gaussian processes with two different kernel length-scales (ℓ = 0.1 and ℓ = 0.5), allowing the authors to control the level of complexity and the number of fine-grained features (or "wiggles") in the data.

For the Real-World Weather Data Experiments, the experiments include both t2m station prediction and multi-modal wind speed interpolation. These tasks are highly relevant to weather forecasting applications.

**Methods And Evaluation Criteria:**

1. The use of predictive log-likelihood and forward pass time as evaluation metrics provides a comprehensive view of both the model’s accuracy and its computational efficiency.

2. The choice to benchmark on synthetic Gaussian process regression tasks allows for controlled experiments that test the model under varying levels of complexity.

3. The application to real-world weather data, including t2m station prediction and multi-modal wind speed interpolation, ensures that the evaluation reflects practical challenges and the scalability of the proposed approach.

**Other Comments Or Suggestions:**

N/A

**Other Strengths And Weaknesses:**

comparing the effects of different grid sizes, or attention mechanisms independently would strengthen the analysis.

**Questions For Authors:**

N/A

**Relation To Broader Scientific Literature:**

N/A

**Theoretical Claims:**

Checked. No incorrect proofs.

---

> ### Author Rebuttal · Authors · 2025-03-31
>
> Thank you for your review and for describing our experimental design “sound and well-aligned with the goals of large-scale spatio-temporal modelling.” We address your concerns below.
>
> **Claims and Evidence**
>
> 3. We believe there may be some confusion here because, as mentioned in the caption of Table 1, $\cancel{NN}$ indicates that nearest-neighbour cross-attention is **not** employed (hence full-attention is used). In Table 1:
> - **Swin-TNP** uses nearest-neighbour cross-attention (NN-CA) and achieves a log-likelihood of **1.819**.
> - **Swin-TNP $\cancel{NN}$** does *not* use nearest-neighbour cross-attention (i.e., uses full cross-attention) and achieves a lower log-likelihood of 1.636 (worse performance).
>
> This supports our claim that NN-CA outperforms full cross-attention.
>
> We further validate this claim in Appendix G.2.1—**Analysis of influence of nearest-neighbour encoding and decoding** Table 7--- where we test this claim for the kernel interpolation grid encoder (KI-GE) too:
> - Swin-TNP with PT-GE (1.819) vs Swin-TNP $\cancel{NN}$ (i.e. full attention) with PT-GE (1.636)
> - Swin-TNP with KI-GE (1.683) vs Swin-TNP $\cancel{NN}$ (i.e. full attention) with KI-GE (1.544)
>
> Similarly, in Appendix G.1 Table 3, we observe the same pattern in the GP experiment for a grid size of $32 \times 32$:
> 1) $l=0.5$: Swin-TNP with PT-GE (0.844) vs Swin-TNP $\cancel{NN}$ with PT-GE (0.837)
> 2) $l=0.1$: Swin-TNP with PT-GE (0.723) vs Swin-TNP $\cancel{NN}$ with PT-GE (0.709) — we will correct a typo that mistakenly indicates 0.109 currently.
> 3)  $l=0.5$: Swin-TNP with KI-GE (0.844) vs Swin-TNP $\cancel{NN}$ with KI-GE (0.834)
> 4) $l=0.1$: Swin-TNP with KI-GE (0.723) vs Swin-TNP $\cancel{NN}$ with KI-GE (0.716)
>
> **Other Strengths and Weaknesses**---”Comparing attention mechanisms"
>
> This also addresses your comment from **Other Strengths and Weaknesses** regarding the comparison of different attention mechanisms. While we focused this analysis on our best-performing model (Swin-TNP) due to computational constraints, we believe that the reason NN-CA outperforms full attention is task-dependent rather than model-dependent. As noted in Appendix G.2.1, locality provides a good inductive bias for spatio-temporal tasks, which should hold regardless of the model architecture.
>
> Furthermore, instead of comparing k-nearest-neighbour and full attention for all models (rather than just for the best-performing ones), we chose to focus computational resources on exploring other aspects of the attention mechanism—specifically, the **number** of nearest neighbours used in the encoder ($k_{\text{enc}}​$) and decoder ($k_{\text{dec}}$​). The results, presented in Table 8, highlight the importance of the locality inductive bias in our tasks (see the discussion under Table 8 for further details).
>
> **Other Strengths and Weaknesses**---”Comparing grid sizes"
>
> Thank you for your suggestion. Please note we do already include an analysis of the grid size for two models on the t2m experiment: Swin-TNP with PT-GE and ConvCNP for grids of sizes $64 \times 128$ and $192 \times 384$.
>
> Table 6 in Appendix G.2.1 shows how the test log-likelihood increases with increased grid size for both models in the t2m experiment. We believe that this trend should generally hold as the model has access to more fine-grained information. However, this comes at an increased computational cost, with the ideal trade-off being experiment-specific.
>
> We also provide an extensive ablation of grid size for all models in the synthetic GP experiment. Table 3 in Appendix G.1 demonstrates that increasing grid size generally improves performance across models, but the gains are task-dependent. For example, in the synthetic GP experiments, the performance gains with increasing grid size are more pronounced when the task is harder (for $l=0.1$ when the model needs to assimilate finely grained information), which aligns with our expectations.
>
> Moreover, during the rebuttal phase we ran additional experiments for the ConvCNP with increased grid sizes (larger than for Swin-TNP) to address reviewer’s tnKQ comment. The results can be found [here](https://anonymous.4open.science/r/ICML_2025-06DE/swintnp_convcnp_comparison.png).
>
>
> If the reviewer feels that we have sufficiently addressed their concerns, we would greatly appreciate a reconsideration of the score.

---

### Decision · Program_Chairs · 2025-05-01

**Decision:**

Accept (spotlight poster)

**Comment:**

All reviewers agreed that this paper proposes a novel method that improves the performance of Neural Processes (NP) on high dimensional data.
There is a consensus that the paper is well motivated, well presented and contains extensive experimentation that validates the claims.
The contributions include a method to incorporate attention with pseudo-tokens, comparison between different methods, and improvements to prior NP methods.